# Equivariant Polynomial Functional Networks

Thieu N. Vo [* 1]  Viet-Hoang Tran [* 1]  Tho Tran Huu [* 1]  An Nguyen The [2]  Thanh Tran [3]
Minh-Khoi Nguyen-Nhat [2]  Duy-Tung Pham [2]  Tan M. Nguyen [1]

## Abstract

A neural functional network (NFN) is a specialized type of neural network designed to process and learn from entire neural networks as input data. Recent NFNs have been proposed with permutation and scaling equivariance based on either graph-based message-passing mechanisms or parameter-sharing mechanisms. However, the challenge of designing a permutation and scaling equivariant NFN that maintains low memory consumption and running time while preserving expressivity remains unresolved. In this paper, we propose a novel solution with the development of MAGEP-NFN (**M**onomial m**A**trix **G**roup **E**quivariant **P**olynomial **NFN**). Our approach follows the parameter-sharing mechanism but differs from previous works by constructing a nonlinear equivariant layer represented as a polynomial in the input weights. This polynomial formulation enables us to incorporate additional relationships between weights from different input hidden layers, enhancing the model's expressivity while keeping memory consumption and running time low, thereby addressing the aforementioned challenge. We provide empirical evidence demonstrating that MAGEP-NFN achieves competitive performance and efficiency compared to existing baselines.

## 1. Introduction

Neural functional networks (NFNs) serve as specialized architectures that operate on fundamental components of deep neural networks, including weights, gradients, and sparsity masks, by treating them as input data (Zhou et al., 2024b). These networks have been utilized across various domains of machine learning, contributing to applications such as enhancing training efficiency through learnable optimizers (Bengio et al., 2013; Runarsson & Jonsson, 2000; Andrychowicz et al., 2016; Metz et al., 2022), capturing features from implicit data representations (Stanley, 2007; Mildenhall et al., 2021; Runarsson & Jonsson, 2000), modifying network parameters for targeted adjustments (Sinitsin et al., 2020; Cao et al., 2021; Mitchell et al., 2022), assessing policies in reinforcement learning (Harb et al., 2020), and facilitating Bayesian inference by interpreting neural networks as sources of evidence (Sokota et al., 2021).

A fundamental aspect of NFNs is their ability to respect the inherent symmetries present in the weight space of the input neural networks. In the case of a multilayer perceptron (MLP), the weight space exhibits two primary forms of symmetry: permutation symmetry and scaling symmetry. Permutation symmetries arise from the structure of the network itself, since neurons within a hidden layer have no intrinsic ordering. On the other hand, scaling symmetries are induced by the activation functions. For networks with the ReLU activation function, multiplying a neuron's bias and all its incoming weights by the same positive scalar scales its output proportionally, leading to a scaling-type symmetry (Bui Thi Mai & Lampert, 2020; Neyshabur et al., 2015; Badrinarayanan et al., 2015). Similarly, for sine and tanh activations, flipping the sign of both the bias and all incoming weights of a neuron inverts the sign of its output, introducing an alternative form of scaling symmetry (Chen et al., 1993; Fefferman & Markel, 1993; Kurkova & Kainen, 1994). These structural symmetries define equivalence classes in the weight space, and incorporating them into the design of NFNs ensures that learned representations remain consistent and invariant to these transformations.

Recent methods have focused on creating permutation equivariant NFNs, such as (Navon et al., 2023; Zhou et al., 2024b; Kofinas et al., 2024; Zhou et al., 2024c). These methods leverage permutation equivariance to respect symmetries arising from neuron reordering within hidden layers. NFNs that are equivariant to both permutations and scaling or sign-flipping have been introduced in (Kalogeropoulos et al., 2024) using a graph-based message-passing mechanism and in (Tran et al., 2024a) with a parameter sharing mechanism. However, similar to other graph-based neural functional networks, treating the entire input neural network as a graph

---
[*]Equal contribution  [1]National University of Singapore  [2]FPT Software AI Center, Vietnam  [3]VinUniversity, Vietnam. Correspondence to: Viet-Hoang Tran <hoang.tranviet@u.nus.edu >.

*Proceedings of the 42nd International Conference on Machine Learning*, Vancouver, Canada. PMLR 267, 2025. Copyright 2025 by the author(s).

and utilizing graph neural networks causes the graph-based equivariant NFNs in (Kalogeropoulos et al., 2024) to have very high memory consumption and running time. In contrast, the NFNs built upon equivariant linear layers using the parameter sharing mechanism in (Tran et al., 2024a) exhibit much lower memory consumption and running time. Nevertheless, the equivariant linear layers introduced in (Tran et al., 2024a) possess weak expressive properties, as the weights of the input hidden layers are updated solely by the corresponding weights of the same input hidden layers. The challenge of designing an equivariant layer based on the parameter-sharing mechanism that maintains both lower memory consumption and running time while preserving expressivity remains unresolved.

### 1.1. Contribution

This paper aims to develop a novel NFN that is equivariant to both permutations and scaling/sign-flipping symmetries, called MAGEP-NFN (**M**onomial m**A**trix **G**roup **E**quivariant **P**olynomial **NFN**). We follow the parameter-sharing mechanism as described in (Tran et al., 2024a); however, unlike (Tran et al., 2024a), we construct a nonlinear equivariant layer, which is represented as a polynomial in the input weights. This polynomial formulation enables us to incorporate additional relationships between weights from different input hidden layers, thereby addressing the challenges posed in (Tran et al., 2024a) and enhancing the expressivity of MAGEP-NFN.

However, determining equivariant and invariant layers among generic polynomials in the input weights is challenging due to two main factors: the difficulty in identifying polynomial orbits under group actions and the high computational cost of working with generic polynomials. To overcome these issues, we introduce a specialized class of polynomials that remain "stable" under permutations and scaling. Restricting equivariant and invariant layers to linear combinations of these terms ensures computational efficiency and reduced memory consumption.

In particular, our contribution is as follows:

1. We introduce a class of polynomials in the input weights, referred to as *stable polynomial terms*, which remain stable under the group action of the weight space. In addition, we conduct a comprehensive study of the linear independence of stable polynomial terms.

2. We characterize all equivariant and invariant layers among the linear combination of the stable polynomial terms. These layers are polynomials of degree at most $L + 1$ where $L$ is the number of layers of the input neural networks.

3. Build on top of the equivariant and invariant poly-

nomial layers, we design MAGEP-NFN, a family of monomial matrix equivariant NFNs based on the parameter-sharing mechanism that maintains both lower memory consumption and running time while preserving expressivity.

We evaluate MAGEP-NFNs on three tasks: predicting CNN generalization from weights using Small CNN Zoo (Unterthiner et al., 2020), weight space style editing, and classifying INRs using INRs data (Zhou et al., 2024b). Experimental results show that our model achieves competitive performance and efficiency compared to existing baselines.

### 1.2. Notations

Let $n$ be a positive integer. We denote $\mathcal{P}_n$ as the set of all permutation matrices, and $\mathcal{D}_n$ as the set of diagonal matrices in $\mathrm{GL}_n(\mathbb{R})$. We also denote $\mathcal{M}_n$ as the set of monomial matrices in $\mathrm{GL}_n(\mathbb{R})$, where a monomial matrix is a product of a diagonal matrix and a permutation matrix. These sets are subgroups of $\mathrm{GL}_n(\mathbb{R})$. In addition, we use $\mathcal{M}_n^{>0}$ and $\mathcal{M}_n^{\pm}$ to denote the set of monomial matrices whose nonzero entries are positive numbers and $\pm 1$, respectively.

For any permutation matrix $P \in \mathcal{P}_n$, there exists a unique permutation $\pi \in \mathcal{S}_n$ such that $P$ is obtained by permuting the columns of the identity matrix $I_n$ according to $\pi$. We denote this as $P := P_\pi$ and refer to it as the *permutation matrix* associated with $\pi$, where $\mathcal{S}_n$ is the symmetric group of all permutations on $\{1, 2, \ldots, n\}$.

**Organization.** We reformulate the definitions of weight spaces for MLPs and CNNs, as well as the action of monomial matrix groups on these weight spaces. In Section 3, we construct polynomial equivariant and invariant layers, which serve as the main building blocks for our MAGEP-NFNs. Several experiments are conducted in Section 4 to verify the applicability and efficiency of our models in comparison with previous ones in the literature. Some related works will be recalled in Section 5. The paper concludes with a summary in Section 6.

## 2. Background: Weight Spaces and Their Symmetries

Let $\mathcal{U}, \mathcal{V}$ be two sets and assume that a group $G$ acts on them. A function $f : \mathcal{U} \to \mathcal{V}$ is called $G$-equivariant if $f(g \cdot x) = g \cdot f(x)$ for all $x \in \mathcal{U}$ and $g \in G$. In case $G$ acts trivially on $\mathcal{Y}$, the function $f$ is called $G$-invariant. In the context of this paper, $\mathcal{U}$ and $\mathcal{V}$ are weight spaces of a fixed neural network architecture, while $\mathcal{G}$ is a direct product of the groups of monomial matrices.

From now on, we will fix the activation $\sigma$ to be the rectified linear unit $\sigma = \mathrm{ReLU}$. The case when $\sigma$ is another typical

activation, such as semilinear (e.g., LeakyReLU) or odd (e.g., $\sin, \tanh$), can be derived similarly.

Following (Tran et al., 2024a), we write the weight space of an MLP or CNN with $L$ layers and $n_i$ channels at $i$-th layer in the general form $\mathcal{U} = \mathcal{W} \times \mathcal{B}$, where:

$$
\begin{aligned}
\mathcal{W} &= \mathbb{R}^{w_L \times n_L \times n_{L-1}} \times \ldots \times \mathbb{R}^{w_2 \times n_2 \times n_1} \times \mathbb{R}^{w_1 \times n_1 \times n_0}, \\
\mathcal{B} &= \mathbb{R}^{b_L \times n_L \times 1} \quad \times \ldots \times \mathbb{R}^{b_2 \times n_2 \times 1} \times \mathbb{R}^{b_1 \times n_1 \times 1}.
\end{aligned}
\tag{1}
$$

Here, $n_i$ is the number of channels at the $i$-th layer, in particular, $n_0$ and $n_L$ are the number of channels of input and output; $w_i$ is the dimension of weights and $b_i$ is the dimension of the biases in each channel at the $i$-th layer.

Each element $U$ of $\mathcal{U}$ is written as $U = ([W], [b])$, with the weights

$$
[W] = \left( [W]^{(L)}, \ldots, [W]^{(1)} \right) \in \mathcal{W},
\tag{2}
$$

and biases

$$
[b] = \left( [b]^{(L)}, \ldots, [b]^{(1)} \right) \in \mathcal{B}.
\tag{3}
$$

The square brackets will be convenient in the next section when we determine polynomials in the entries of $U$.

The weight space $\mathcal{U}$ exhibits two primary forms of symmetry: permutation symmetry and scaling symmetry. Permutation symmetries arise from the structure of the network itself, since neurons within a hidden layer have no intrinsic ordering. On the other hand, for networks with the ReLU activation function, multiplying a neuron's bias and all its incoming weights by the same positive scalar scales its output proportionally, leading to a scaling-type symmetry. Based on the above observation, we define the group $G$ of the form

$$
G := \{I_{n_L}\} \times \mathcal{M}_{n_{L-1}}^{>0} \times \ldots \times \mathcal{M}_{n_1}^{>0} \times \{I_{n_0}\},
\tag{4}
$$

where $I_n$ is the identity matrix of size $n \times n$, and $\mathcal{M}_n^{>0}$ is the set of monomial matrices whose nonzero entries are positive numbers.

Each element $g \in G$ has the form

$$
g = \left( g^{(L)}, \ldots, g^{(0)} \right),
$$

where $g^{(i)} = D^{(i)} \cdot P_{\pi_i}$ for some diagonal matrix $D^{(i)} = \operatorname{diag}(d_1^{(i)}, \ldots, d_{n_i}^{(i)})$ in $\mathcal{D}_{n_i}$ and permutation $\pi_i \in \mathcal{S}_{n_i}$. The action of $G$ on $\mathcal{U}$ is defined formally as

$$
(g, U) \mapsto gU = ([gW], [gb]),
$$

where:

$$
[gW]^{(i)} := \left( g^{(i)} \right) \cdot [W]^{(i)} \cdot \left( g^{(i-1)} \right)^{-1}
$$

and $[gb]^{(i)} := \left( g^{(i)} \right) \cdot [b]^{(i)},$ (5)

or equivalently,

$$
[gW]_{jk}^{(i)} := \frac{d_j^{(i)}}{d_k^{(i-1)}} \cdot [W]_{\pi_i^{-1}(j) \pi_{i-1}^{-1}(k)}^{(i)}
$$

$$
\text{and} \quad [gb]_j^{(i)} := d_j^{(i)} \cdot [b]_{\pi_i^{-1}(j)}^{(i)}.
\tag{6}
$$

With notation as above, it is well-known that the function $f = f(\,\cdot\,; U, \sigma)$ be an MLP or CNN given in Equation (1) with the weight space $U \in \mathcal{U}$ and an activation $\sigma = \mathrm{ReLU}$ will be $G$-invariant under the action of $G$, i.e.

$$
f(\mathbf{x}\,;\, U, \sigma) = f(\mathbf{x}\,;\, gU, \sigma)
\tag{7}
$$

for all $g \in G$, $U \in \mathcal{U}$ and $\mathbf{x} \in \mathbb{R}^{n_0}$.

# 3. Equivariant and Invariant Polynomial Functional Networks

In this section, we construct our MAGEP-NFNs, a new class of NFNs that exhibit equivariance to the group $G$ of monomial matrices with positive nonzero entries, as described in the previous section. The core components of MAGEP-NFNs are invariant and equivariant polynomial layers, which will be detailed in Subsection 3.2. At the heart of these layers are the stable polynomial terms, which play a crucial role in ensuring low memory consumption and computational efficiency of MAGEP-NFNs. These terms will be formally introduced in Subsection 3.1.

## 3.1. Stable polynomial terms

We follow the parameter-sharing mechanism used in (Tran et al., 2024a) for constructing equivariant and invariant layers from $\mathcal{U}$ to ensure low memory consumption and computational efficiency in our model. Unlike the linear layers utilized in (Tran et al., 2024a), we employ polynomial layers. This choice allows us to capture additional relationships between weights from different hidden layers of the input network, thereby enhancing the expressivity of our model.

However, determining equivariant and invariant layers among generic polynomials in the input weights presents a significant challenge for two key reasons. First, identifying the orbits of a generic polynomial under the group action is difficult, as such polynomials lack an inherent structured compatibility with the symmetry group of the weight space. Second, computations involving equivariant and invariant layers constructed from generic polynomials are highly inefficient in both memory usage and computational cost. To address these challenges, we introduce a specialized class of polynomials, referred to as *stable polynomial terms*. Intuitively, a *stable polynomial term* is a polynomial in the

entries of $U \in \mathcal{U}$ that remains "stable" under the action of $G$ (see Definition 3.1). By restricting equivariant and invariant polynomial layers to linear combinations of these stable polynomial terms, we ensure both computational efficiency and reduced memory consumption.

Consider the case where the weight spaces have the same number of dimensions across all channels, which means $w_i = b_i = d$ for all $i$.

**Definition 3.1** (Stable polynomial terms)**.** Assume that $U = ([W], [b])$ is an element of the weight space $\mathcal{U}$ with weights $[W] = ([W]^{(L)}, \ldots, [W]^{(1)})$ and biases $[b] = ([b]^{(L)}, \ldots, [b]^{(1)})$. For each $L \geqslant s > t \geqslant 0$, we define the following matrices:

$$[W]^{(s,t)} := [W]^{(s)} \cdot [W]^{(s-1)} \cdot \ldots \cdot [W]^{(t+1)},$$
$$[Wb]^{(s,t)(t)} := [W]^{(s,t)} \cdot [b]^{(t)}. \tag{8}$$

In addition, for each indices $s$ and $t$ with $L \geqslant s, t \geqslant 0$, and matrices $\Psi^{(s)(L,t)} \in \mathbb{R}^{1 \times n_L}$ and $\Psi^{(s,0)(L,t)} \in \mathbb{R}^{n_0 \times n_L}$, we also define the following matrices:

$$[bW]^{(s)(L,t)} := [b]^{(s)} \cdot \Psi^{(s)(L,t)} \cdot [W]^{(L,t)},$$
$$[WW]^{(s,0)(L,t)} := [W]^{(s,0)} \cdot \Psi^{(s,0)(L,t)} \cdot [W]^{(L,t)}. \tag{9}$$

The entries of the matrices $[W]^{(s,t)}$, $[Wb]^{(s,t)(t)}$, $[bW]^{(s)(L,t)}$ and $[WW]^{(s,0)(L,t)}$ defined above are called *stable polynomial terms* of $U$ under the action of $G$. Note that we omit the $\Psi^{(-)}$'s in $[bW], [WW]$ for simplicity, they are parameters, and are different for each $s, t$. The denotations are straight-forward and reasonable.

In the above definition, we use the notation $[W]$ and $[WW]$ to denote products of the weight matrices $[W]^{(i)}$ with the appropriate index $i$. The notation $[Wb]$ indicates that this is a product of several weight matrices $[W]^{(i)}$ and a bias vector $[b]^{(j)}$, with appropriate indices $i$ and $j$. For the indices, we use the notation $(s, t)$ to signify that the considered product contains weight matrices with indices ranging from $s$ down to $t + 1$. When the index has two components, for example $[Wb]^{(s,t)(t)}$, the first component $(s, t)$ specifies the range of indices for $[W]$, while the second component $(t)$ indicates the index of the bias vector $[b]$. Specifically, the last two terms $[bW]^{(s)(L,t)}$ and $[WW]^{(s,0)(L,t)}$ contain the matrices $\Psi^-$ to multiply two matrices of different sizes from the left and the right.

**Proposition 3.2** (Stable polynomial terms as generalization of weights and biases)**.** *With notation as above, then for all* $L \geqslant s > t > r \geqslant 0$, *we have*

$$\begin{aligned}
[W]^{(s,s-1)} &= [W]^{(s)} &\in \mathbb{R}^{d \times n_s \times n_{s-1}}, \\
[W]^{(s,t)} \cdot [W]^{(t,r)} &= [W]^{(s,r)} &\in \mathbb{R}^{d \times n_s \times n_r}. \\
[bW]^{(s)(s,t)} \cdot [W]^{(t,r)} &= [bW]^{(s,r)} &\in \mathbb{R}^{d \times n_s \times n_r}, \\
[W]^{(s,t)} \cdot [Wb]^{(t,r)} &= [Wb]^{(s,r)} &\in \mathbb{R}^{d \times n_s \times n_r}.
\end{aligned}$$

The above proposition shows that the stable polynomial terms can be viewed as a generalization of the entries of the weight matrices $[W]^{(i)}$ and bias vectors $[b]^{(i)}$. The stable polynomial terms defined above are actually "stable" under the action of $G$ in the sense presented in the following theorem.

**Theorem 3.3** (Stable polynomial terms are "stable")**.** *With notation as above, let* $gU = ([gW], [gb])$ *be the element of* $\mathcal{U}$ *obtained by acting* $g = (g^{(L)}, \ldots, g^{(0)}) \in G$ *on the element* $U = ([W], [b])$. *Then we have*

$$\begin{aligned}
[gW]^{(s,t)} &= \left(g^{(s)}\right) \cdot [W]^{(s,t)} \cdot \left(g^{(t)}\right)^{-1}, \\
[gb]^{(s)} &= \left(g^{(s)}\right) \cdot [b]^{(s)}, \\
[gWgb]^{(s,t)(t)} &= \left(g^{(s)}\right) \cdot [Wb]^{(s,t)(t)}, \\
[gbgW]^{(s)(L,t)} &= \left(g^{(s)}\right) \cdot [bW]^{(s)(L,t)} \cdot \left(g^{(t)}\right)^{-1}, \\
[gWgW]^{(s,0)(L,t)} &= \left(g^{(s)}\right) \cdot [WW]^{(s,0)(L,t)} \cdot \left(g^{(t)}\right)^{-1}.
\end{aligned}$$

Intuitively, the theorem above asserts that stable polynomials exhibit compatibility with the action of the group $G$. In particular, it provides explicit formulas for efficiently determining the transformation of stable polynomials under group actions. This property plays a crucial role in enabling the efficient computation of equivariant and invariant layers, especially when utilizing the weight-sharing mechanism.

Inherited from Proposition 3.2 and Theorem 3.3, we define the polynomial map $I : \mathcal{U} \to \mathbb{R}^{d'}$ with maps each element $U \in \mathcal{U}$ to the vector $I(U) \in \mathbb{R}^{d'}$ of the following form:

$$\begin{aligned}
I(U) :=& \sum_{L \geqslant s > t \geqslant 0} \sum_{p=1}^{n_s} \sum_{q=1}^{n_t} \Phi_{(s,t):pq} \cdot [W]_{pq}^{(s,t)} \\
&+ \sum_{L \geqslant s > 0} \sum_{p=1}^{n_s} \Phi_{(s):p} \cdot [b]_p^{(s)} \\
&+ \sum_{L \geqslant s > t > 0} \sum_{p=1}^{n_s} \Phi_{(s,t)(t):p} \cdot [Wb]_p^{(s,t)(t)} \\
&+ \sum_{L \geqslant s > 0} \sum_{L > t \geqslant 0} \sum_{p=1}^{n_s} \sum_{q=1}^{n_t} \Phi_{(s)(L,t):pq} \cdot [bW]_{pq}^{(s)(L,t)} \\
&+ \sum_{L \geqslant s > 0} \sum_{L > t \geqslant 0} \sum_{p=1}^{n_s} \sum_{q=1}^{n_t} \Phi_{(s,0)(L,t):pq} \cdot [WW]_{pq}^{(s,0)(L,t)} \\
&+ \Phi_1. \tag{10}
\end{aligned}$$

Intuitively speaking, $I(U)$ is a linear combination of the entries from the input weights $[W]^{(s)}$ (from the first sum) and biases $[b]^{(s)}$ (the second sum), as well as all entries from the stable polynomial terms $[W]^{(s,t)}$ (the first sum), $[Wb]^{(s,t)(t)}$ (the third sum), $[bW]^{(s)(s,t)}$ (the fourth sum), and $[WW]^{(s,L)(0,t)}$ (the last fifth sum) for all appropriate indices $s$ and $t$, with a bias. Here, the coefficients

$\Phi_-$ and the connection matrix $\Psi^-$ (inside $[bW]^{(s)(s,t)}$ and $[WW]^{(s,L)(0,t)}$) are learnable parameters.

It is important to note that $I(U)$ is a polynomial of degree at most $L+1$ in the input weights, encompassing all linear layers as special cases.

To identify invariant layers among polynomials of the form given in $I$, an important step is to determine all relations between elements $U, U' \in \mathcal{U}$ such that $I(U) = I(U')$. When the parameters $\Psi^-$ are fixed, $I$ becomes a linear function in the parameters $\Phi_-$, and the difference satisfies $I(U) - I(U') = I(U - U')$. Thus, the problem of identifying invariant layers within polynomials of the form $I$ reduces to determining the linear dependencies among stable polynomials. The following theorem characterizes these dependencies in $I(U)$, with a formal statement provided in Theorem B.6 in the appendix.

**Theorem 3.4** (Linear dependence of stable polynomials)**.** *For a given pair of coefficients matrix $\Phi_-$ and $\Psi^-$, if $I(U)$ given in Equation (10) is equal to zero for all input weights $U \in \mathcal{U}$, then we have*

$$\sum_{p=1}^{n_L}\sum_{q=1}^{n_0} \Phi_{(L,0):pq} \cdot [W]_{pq}^{(L,0)}$$
$$+ \sum_{L>s>0}\sum_{p=1}^{n_s}\sum_{q=1}^{n_s} \Phi_{(s,0)(L,s):pq} \cdot [WW]_{pq}^{(s,0)(L,s)} = 0,$$

$$(11)$$

*and*

$$\sum_{p=1}^{n_L} \Phi_{(L,t)(t):p} \cdot [Wb]_{p}^{(L,t)(t)}$$
$$+ \sum_{p=1}^{n_t}\sum_{q=1}^{n_t} \Phi_{(t)(L,t):pq} \cdot [bW]_{pq}^{(t)(L,t)} = 0, \quad L > t > 0,$$

$$(12)$$

*and all entries of $\Phi_-$ and $\Psi^-$, except those appear in the above two equations, are equal to zero.*

Intuitively speaking, almost stable polynomial terms are linearly independent over the reals $\mathbb{R}$, except those in Equations (11) and (12). This linear dependence property of the stable polynomials is essential in the computation of equivariant and invariant polynomial layers using weight-sharing mechanism. The proofs of Proposition 3.2, Theorem 3.3, and Theorem 3.4 can be found in Appendix B.

### 3.2. Polynomial Invariant and Equivariant Layers

We now proceed to construct $G$-invariant polynomial layers. The construction of $G$-equivariant polynomial layers is similar and will be derived in detail in Appendix C. These polynomial layers serve as the fundamental building blocks for our MAGEP-NFNs.

We define a polynomial map $I: \mathcal{U} \to \mathbb{R}^{d'}$ with maps each element $U \in \mathcal{U}$ to the vector $I(U) \in \mathbb{R}^{d'}$ of the form given in Equation (10). To make $I$ to be $G$-invariant, the learnable parameters $\Phi_-$ and $\Psi^-$ must satisfy a system of constraints (usually called *parameter sharing*), which are induced from the condition $I(gU) = I(U)$ for all $g \in G$ and $U \in \mathcal{U}$. We show in details what are these constraints and how to derive the concrete formula of $I$ in Appendix C. The formula of $I$ is then determined by

$$I(U) = \sum_{p=1}^{n_L}\sum_{q=1}^{n_0} \Phi_{(L,0)(L,0):pq} \cdot [WW]_{pq}^{(L,0)(L,0)}$$
$$+ \sum_{p=1}^{n_L}\sum_{q=1}^{n_0} \Phi_{(L,0):pq} \cdot [W]_{pq}^{(L,0)}$$
$$+ \sum_{L>s>0}\sum_{p=1}^{n_s} \Phi_{(s,0)(L,s):\bullet\bullet} \cdot [WW]_{pp}^{(s,0)(L,s)}$$
$$+ \sum_{p=1}^{n_L}\sum_{q=1}^{n_0} \Phi_{(L)(L,0):pq} \cdot [bW]_{pq}^{(L)(L,0)}$$
$$+ \sum_{L>t>0}\sum_{p=1}^{n_L} \Phi_{(L,t)(t):p} \cdot [Wb]_{p}^{(L,t)(t)}$$
$$+ \sum_{L>t>0}\sum_{p=1}^{n_t} \Phi_{(t)(L,t):\bullet\bullet} \cdot [bW]_{pp}^{(t)(L,t)}$$
$$+ \sum_{p=1}^{n_L} \Phi_{(L):p} \cdot [b]_{p}^{(L)} + \Phi_1. \tag{13}$$

In the above formula, the bullet symbol $\bullet$ denotes that the corresponding coefficient is independent of the index at that position. The summation terms involving expressions of the form $[W]$ (respectively, $[b], [Wb], [bW], [WW]$) correspond to those present in the summation in Equation (10). The omitted terms are those eliminated during solving the parameter-sharing process to ensure that the resulting formula becomes invariant.

To conclude, we obtain the following:

**Theorem 3.5.** *With the notation given above, the polynomial map $I: \mathcal{U} \to \mathbb{R}^{d'}$ defined by Equation (13) is $G$-invariant. Moreover, if a map takes the form of Equation (10) and is $G$-invariant, then it has the form given in Equation (13).*

*Remark* (Comparison to the invariant/equivariant linear layers in (Tran et al., 2024a)). Equation (13) describes the invariant polynomial layer derived from the parameter-sharing mechanism of our MAGEP-NFNs. In contrast, the invariant equivariant layer proposed in (Tran et al., 2024a) is an ad hoc

Table 1: Classification train and test accuracies (%) for implicit neural representations of MNIST, FashionMNIST, and CIFAR-10. Uncertainties indicate standard error over 5 runs, baseline results are from (Tran et al., 2024a).

|  | MNIST | CIFAR-10 | FashionMNIST |
|---|---|---|---|
| MLP | $10.62 \pm 0.54$ | $10.48 \pm 0.74$ | $9.95 \pm 0.36$ |
| NP (Zhou et al., 2024b) | $\underline{69.82 \pm 0.42}$ | $33.74 \pm 0.26$ | $58.21 \pm 0.31$ |
| HNP (Zhou et al., 2024b) | $66.02 \pm 0.51$ | $31.61 \pm 0.22$ | $57.43 \pm 0.46$ |
| Monomial-NFN (Tran et al., 2024a) | $68.43 \pm 0.51$ | $\underline{34.23 \pm 0.33}$ | $\underline{61.15 \pm 0.55}$ |
| MAGEP-NFNs (ours) | $\mathbf{77.55 \pm 0.68}$ | $\mathbf{37.18 \pm 0.30}$ | $\mathbf{62.83 \pm 0.57}$ |

formulation and does not result from a parameter-sharing mechanism. Consequently, there is no direct relationship between our invariant layer and the invariant layer in (Tran et al., 2024a).

However, the equivariant polynomial layer in our MAGEP-NFNs and the equivariant linear layer from (Tran et al., 2024a) are related. Specifically, the equivariant layer in (Tran et al., 2024a) is exactly the linear component of our equivariant polynomial layer. Due to the lengthy formulation and construction process, we have provided the details of the derived equivariant polynomial layers in Appendix D.4.

## 4. Experimental Results

In this session, we assess the performance of our Monomial Matrix Group Polynomial Equivariant Neural Functionals (MAGEP-NFNs) across a variety of equivariant and invariant tasks. For invariant tasks, we implement our model for classifying Implicit Neural Representations of images and predicting CNN generalization based on weights. The equivariant task focuses on weight space style editing. Our experiments are designed to illustrate that MAGEP-NFNs either outperform or match the performance of other baseline models with a similar number of parameters. We perform five independent runs for each experiment and report the average results. Detailed information on hyperparameter settings, training protocols, memory and runtime analysis can be found in Appendix E. Additionally, we present a supplementary experiment comparing our approach with a GNN-based functional network, which is included in Appendix F.

### 4.1. Classifying Implicit Neural Representations of images

**Experiment setup.** In this experiment, we aim to determine which class each pretrained Implicit Neural Representation (INR) weight was trained on. Following (Tran et al., 2024a), we employ three distinct INR weight datasets (Zhou et al., 2024b), each was trained on a different image dataset: CIFAR-10 (Krizhevsky & Hinton, 2009), Fashion-MNIST (Xiao et al., 2017), and MNIST (LeCun & Cortes, 2005). Each INR weight is trained to encode a single image from its respective class, capturing the image structure by mapping pixel coordinates $(x, y)$ to the corresponding pixel color values—represented as 3-channel RGB values for CIFAR-10 and 1-channel grayscale values for MNIST and FashionMNIST. The varying complexity and diversity of the datasets provide a robust test for evaluating MAGEP-NFN's performance, demonstrating its effectiveness and benchmarking it against existing models.

**Results.** We present the performance of our model alongside several baseline models, including MLP, NP (Zhou et al., 2024b), HNP (Zhou et al., 2024b), and Monomial-NFN (Tran et al., 2024a). As shown in Table 1, our model achieves the highest test accuracies across all INR datasets. Notably, it outperforms the second-best model on the MNIST dataset, by a significant margin of 7.73%. For the CIFAR-10 and FashionMNIST datasets, our model also demonstrates substantial improvements, with accuracy gains of 2.95% and 1.68%, respectively, over the existing baselines. These results indicate that our model leverages the embedded information from the pretrained INRs more effectively than any of the compared baselines. This consistent superior performance across various INR datasets highlights the effectiveness of MAGEP-NFN. It also suggests that our model generalizes well to INR weights embedded with different image structures and complexities.

### 4.2. Predicting CNN generalization from weights

**Experiment setup.** For this experiment, we focus on predicting the generalization performance of pretrained CNNs based solely on their weights, without evaluating them on test data. We utilize the Small CNN Zoo dataset (Unterthiner et al., 2020), which contains various pretrained CNN models trained with different combinations of hyperparameters and activation functions. For our study, we split the Small CNN Zoo into two subsets: one comprising networks using ReLU activations and the other using Tanh activations. These two types of CNNs follow different group actions: $\mathcal{M}_n^{>0}$ for Relu networks (see Equation (4)) and $\mathcal{M}_{n_i}^{\pm 1}$ for Tanh networks.

Table 2: Performance prediction of CNNs on the ReLU subset of Small CNN Zoo with varying scale augmentations. We use Kendall's $\tau$ as the evaluation metric. The uncertainty bars indicate the standard deviation across 5 runs.

| | | Augment settings | | | |
|---|---|---|---|---|---|
| | No augment | $\mathcal{U}[1, 10^1]$ | $\mathcal{U}[1, 10^2]$ | $\mathcal{U}[1, 10^3]$ | $\mathcal{U}[1, 10^4]$ |
| STATNet (Unterthiner et al., 2020) | $0.915 \pm 0.002$ | $0.894 \pm 0.0001$ | $0.853 \pm 0.007$ | $0.523 \pm 0.02$ | $0.516 \pm 0.001$ |
| NP (Zhou et al., 2024b) | $0.920 \pm 0.003$ | $0.900 \pm 0.002$ | $0.898 \pm 0.003$ | $0.884 \pm 0.002$ | $0.884 \pm 0.002$ |
| HNP (Zhou et al., 2024b) | $\underline{0.926 \pm 0.003}$ | $0.913 \pm 0.001$ | $0.903 \pm 0.003$ | $0.891 \pm 0.003$ | $0.601 \pm 0.02$ |
| Monomial-NFN (Tran et al., 2024a) | $0.922 \pm 0.001$ | $\underline{0.920 \pm 0.001}$ | $\underline{0.919 \pm 0.001}$ | $\underline{0.920 \pm 0.002}$ | $\underline{0.920 \pm 0.001}$ |
| MAGEP-NFNs (ours) | $\mathbf{0.933 \pm 0.001}$ | $\mathbf{0.933 \pm 0.001}$ | $\mathbf{0.933 \pm 0.001}$ | $\mathbf{0.932 \pm 0.001}$ | $\mathbf{0.932 \pm 0.001}$ |

Table 3: Performance prediction of CNNs on the Tanh subset of Small CNN Zoo. We use Kendall's $\tau$ as the evaluation metric. The uncertainty bars indicate the standard deviation across 5 runs.

| Model | Kendall's $\tau$ |
|---|---|
| STATNet (Unterthiner et al., 2020) | $0.913 \pm 0.0012$ |
| NP (Zhou et al., 2024b) | $0.925 \pm 0.0013$ |
| HNP (Zhou et al., 2024b) | $0.933 \pm 0.0019$ |
| Monomial-NFN (Tran et al., 2024a) | $\underline{0.939 \pm 0.0004}$ |
| MAGEP-NFNs (ours) | $\mathbf{0.940 \pm 0.001}$ |

To evaluate the robustness of our model to input transformations under group actions, we augment the ReLU dataset by applying randomly sampled group actions $\mathcal{M}_n^{>0}$. Specifically, we randomly sampling the diagonal elements $\mathcal{D}_{n,ii}^{>0}$ of the matrix $\mathcal{D}_n^{>0}$, with each element drawn from uniform distributions over different ranges, defined as $\mathcal{U}[1, 10^i]$ for $i = 1, 2, 3, 4$. To further diversify the transformations, we also randomly sample the permutation matrix $\mathcal{P}_n$.

**Results.** Table 2 illustrates the performance of all models trained on the ReLU subset, where our MAGEP-NFNs model clearly outperforms all other baselines. Notably, it demonstrates robustness to scale and permutation symmetry, similar to Monomial-NFN, while consistently surpassing its performance across both the original and all augmented dataset settings. This suggests that incorporating polynomial layers allows our model to capture more information from the weights across different hidden layers, compared to Monomial-NFN, thereby enhancing expressivity. On the original dataset, our model achieves a Kendall's $\tau$ performance gap of $0.007$ over other baselines, and maintaining at least a $0.012$ advantage in all other augmented settings. Similarly, Table 3 reveals that MAGEP-NFNs achieves the highest Kendall's $\tau$ with Tanh activation, further reinforcing its superior accuracy across different network configurations.

### 4.3. Weight space style editing

**Experiment setup.** In this experiment, we focus on modifying the weights of SIREN (Sitzmann et al., 2020) to modify the image encoded within each model. We utilize the pretrained models from paper (Zhou et al., 2024b), which encode images from the CIFAR-10 and MNIST datasets. Specifically, we address two tasks aimed at modifying the embedded information: enhancing the contrast of CIFAR-10 images and dilating MNIST images encoded in the SIREN models. We report the MSE loss between the images encoded in the modified SIREN network and the ground truth contrast-enhanced CIFAR-10 images or dilated MNIST images.

**Results.** Table 4 demonstrates that our model achieves performance comparable to other baselines. Specifically, MAGEP-NFNs matches the performance of NP and Monomial-NFN in the contrast-enhancing task on the CIFAR-10 dataset. Additionally, our model outperforms Monomial-NFN in the dilation task on the MNIST dataset, while achieving similar results to NP. Interestingly, NP remains a strong candidate in the weight editing tasks, and our model consistently performs on par with NP across both experiments.

### 4.4. Ablation study on the role of higher-order terms

To evaluate the impact of the newly introduced Inter-Layer terms ($[W], [WW], [bW], [Wb]$), we conduct an ablation study focusing on the invariant task of predicting CNN generalization for the ReLU subset, following the same setting outlined in Subsection 4.2. The results presented in Table 5 clearly show that the inclusion of Inter-Layer terms enhances the network's performance. Notably, the performance improves from a Kendall's $\tau$ of $0.929$ with only Non Inter-Layer terms to $0.933$ when both terms are combined, highlighting a significant boost in overall performance attributed to the incorporation of Inter-Layer terms.

Table 4: Test mean squared error (lower is better) between weight-space editing methods and ground-truth image-space transformations. Uncertainties indicate standard error over 5 runs

|  | Contrast (CIFAR-10) | Dilate (MNIST) |
|---|---|---|
| MLP | $0.031 \pm 0.001$ | $0.306 \pm 0.001$ |
| NP (Zhou et al., 2024b) | $\mathbf{0.020 \pm 0.002}$ | $\mathbf{0.068 \pm 0.002}$ |
| HNP (Zhou et al., 2024b) | $\underline{0.021 \pm 0.002}$ | $0.071 \pm 0.001$ |
| Monomial-NFN (Tran et al., 2024a) | $\mathbf{0.020 \pm 0.001}$ | $\underline{0.069 \pm 0.002}$ |
| MAGEP-NFNs (ours) | $\mathbf{0.020 \pm 0.001}$ | $\mathbf{0.068 \pm 0.002}$ |

Table 5: Ablation study assessing the importance of higher-order terms on the task of predicting CNN generalization on the ReLU subset.

| Components | Kendall's $\tau$ |
|---|---|
| Only Non Inter-Layer terms | 0.929 |
| Only Inter-Layer terms | $\underline{0.932}$ |
| Non Inter-Layer terms + $[W]$ | 0.930 |
| Non Inter-Layer terms + $[WW]$ | 0.930 |
| Non Inter-Layer terms + $[Wb]$ | 0.931 |
| Non Inter-Layer terms + $[bW]$ | 0.931 |
| Non Inter-Layer terms + Inter-Layer terms | $\mathbf{0.933}$ |

## 5. Related Work

**Functional Equivalence of Neural Networks.** Hecht-Nielsen was provided a foundational perspective on the relationship between weight symmetries and network functionality in (Hecht-Nielsen, 1990). This work has been extended to different network architectures, such as ReLU networks (Bui Thi Mai & Lampert, 2020; Neyshabur et al., 2015; Badrinarayanan et al., 2015; Albertini & Sontag, 1993), and sin or tanh networks (Chen et al., 1993; Fefferman & Markel, 1993; Kurkova & Kainen, 1994). These studies build on earlier insights into convergence, gradient dynamics, and structural properties of neural networks, as explored in (Allen-Zhu et al., 2019; Du et al., 2019; Frankle & Carbin, 2019; Belkin et al., 2019; Novak et al., 2018).

**Neural Functional Networks.** Early NFNs have been proposed to evaluate their generalization capabilities and uncover insights into neural network dynamics (Baker et al., 2018; Eilertsen et al., 2020; Unterthiner et al., 2020; Schürholt et al., 2021; 2022a;b). These approaches typically involve either flattening the network parameters or deriving parameter statistics for further processing using standard multi-layer perceptrons (MLPs) (Unterthiner et al., 2020; Dupont et al., 2022; Luigi et al., 2023). To involve the symmetric structure of the input neural networks, Schürholt et al. (2021) introduced neuron permutation augmentations to better align model representations with their functional equivalence. Other studies have expanded on these ideas by focusing on encoding and decoding neural network parameters, primarily for reconstruction and generative modeling

(Peebles et al., 2022; Ashkenazi et al., 2023; Knyazev et al., 2021; Erkoç et al., 2023).

**Equivariant Neural Functional Networks.** To achieve NFNs that are equivariant with respect to permutation symmetries of neural networks weight space, several approaches have been used, such as: weight-sharing mechanisms (Navon et al., 2023; Zhou et al., 2024b; Kofinas et al., 2024; Zhou et al., 2024c), set-based (Andreis et al., 2023) or graph-based structures (Lim et al., 2024; Kofinas et al., 2024; Zhou et al., 2024a).

Despite these developments, current approaches often overlook additional symmetries present in neural networks, for instance, weight scaling symmetries in ReLU networks and weight sign-flipping symmetries in sin and tanh networks. Recently, NFNs that are equivariant to both permutations and scaling have been introduced in (Kalogeropoulos et al., 2024; Tran et al., 2024a). These networks leverage advanced techniques such as graph-based message-passing mechanisms (Kalogeropoulos et al., 2024) and parameter-sharing frameworks (Tran et al., 2024a) to extend the scope of equivariant modeling and enhance the expressivity of NFNs. However, the graph-based equivariant NFNs proposed in (Kalogeropoulos et al., 2024) suffer from high memory consumption and significant runtime overhead. While, the Monomial-NFNs constructed using equivariant linear layers and a parameter-sharing mechanism in (Tran et al., 2024a) exhibit limited expressive power.

In contrast, our MAGEP-NFNs are built upon equivariant polynomial layers, leveraging a parameter-sharing mechanism that achieves both lower memory consumption and reduced runtime while preserving strong expressivity.

## 6. Conclusion

We have developed MAGEP-NFN, a novel NFN that is equivariant to both permutations and scaling symmetries. Our approach follows a parameter-sharing mechanism; however, unlike previous works, we construct an equivariant polynomial layer that incorporates stable polynomial terms. This polynomial formulation enables us to capture relationships between weights from different input hidden layers, thereby enhancing the expressivity of MAGEP-NFN while maintaining low memory consumption and efficient running time. Experimental results demonstrate that our model achieves competitive performance and efficiency compared to existing baselines.

One limitation of the equivariant polynomial layers proposed in this paper is that they are applied to a specific architecture. However, since our method is based on a parameter-sharing mechanism, it is applicable to other architectures with additional operators (such as layer normalization, softmax, pooling) and other activation functions, provided that the symmetric group of the weight network is known.

## Impact Statement

This paper presents work whose goal is to advance the field of Machine Learning. There are many potential societal consequences of our work, none which we feel must be specifically highlighted here.

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

# Supplement to "Equivariant Polynomial Functional Networks"

## A. Preliminaries

This section contains notations and basic results on matrices and polynomials that will be used throughout the paper. We will mainly focus on matrices with real entries (real matrices) and polynomials with real coefficients (real polynomials). We will omit almost all of the proofs in this section as they are well-known. These results will be use in proofs in the rest of the paper.

### A.1. Entries of matrices

A real matrix $A$ with $m$ rows and $n$ columns is an element of $\mathbb{R}^{m \times n}$:

$$A = (a_{ij})_{1 \leqslant i \leqslant m, 1 \leqslant j \leqslant n} = \begin{pmatrix} a_{11} & a_{12} & \ldots & a_{1n} \\ a_{21} & a_{22} & \ldots & a_{2n} \\ \vdots & \vdots & \ddots & \vdots \\ a_{m1} & a_{m2} & \ldots & a_{mn} \end{pmatrix} \in \mathbb{R}^{m \times n}.$$

The entry in the $i^{\text{th}}$-row and the $j^{\text{th}}$-column of $A$, or the $(i, j)$ entry of $A$, is denoted by $A_{ij} = a_{ij}$. The $i^{\text{th}}$-row of $A$ and $j^{\text{th}}$-column of $A$ are, respectively, denoted by:

$$\begin{aligned} A_{i*} &= (a_{i1}, a_{i2}, \ldots, a_{in}) & \in \mathbb{R}^{1 \times n}, \\ A_{*j} &= (a_{1j}, a_{2j}, \ldots, a_{mj})^{\top} & \in \mathbb{R}^{m \times 1}. \end{aligned}$$

*Remark.* Sometimes, a comma is added between two subscript indices to make sure there will be no confusion, i.e. $A_{i,j}, a_{i,j}, A_{i,*}, A_{*,j}$.

Let $A^{(L)}, \ldots, A^{(2)}, A^{(1)}$ be $L$ matrices such that the matrix product:

$$A^{(L)} \cdot \ldots \cdot A^{(2)} \cdot A^{(1)},$$

is well-defined.

**Proposition A.1.** *The $(i, j)$ entry of $A^{(L)} \cdot \ldots \cdot A^{(2)} \cdot A^{(1)}$ is equal to:*

$$\begin{aligned} \left( A^{(L)} \cdot \ldots \cdot A^{(2)} \cdot A^{(1)} \right)_{ij} &= A_{i,*}^{(L)} \cdot A^{(L-1)} \cdot \ldots \cdot A^{(2)} \cdot A_{*,j}^{(1)} \\ &= \sum_{k_{L-1}, \ldots, k_2, k_1} a_{i,k_{L-1}}^{(L)} \cdot a_{k_{L-1}, k_{L-2}}^{(L)} \cdot \ldots \cdot a_{k_2, k_1}^{(2)} \cdot a_{k_1, j}^{(1)}. \end{aligned}$$

*In the case where $L = 1$, the above equation is simply $A_{ij}^{(1)} = a_{ij}^{(1)}$.*

We set a denotation for matrices that have only one nonzero entry with value 1. The matrix with the 1 in the $i^{\text{th}}$-row and the $j^{\text{th}}$-column, and the rest are 0, is denoted by $E_{ij}$. Matrix $E_{ij}$ can have any shape, but its shape are usually defined by context, and will be omitted without confusion. The product of matrices of this type is presented as below.

**Proposition A.2.** *Let $E_{i_1, j_1}, E_{i_2, j_2}, \ldots, E_{i_L, j_L}$ be $L$ matrix units such that the product:*

$$E_{i_1, j_1} \cdot E_{i_2, j_2} \cdot \ldots \cdot E_{i_L, j_L},$$

*is well-defined. Then:*

$$E_{i_1, j_1} \cdot E_{i_2, j_2} \cdot \ldots \cdot E_{i_L, j_L} = \left( \delta_{j_1, i_2} \cdot \delta_{j_2, i_3} \cdot \ldots \cdot \delta_{j_{L-1}, i_L} \right) \cdot E_{i_1, j_L},$$

*where $\delta_{ij}$ is the Kronecker delta:*

$$\delta_{ij} = \begin{cases} 0 & \text{if } i \neq j, \\ 1 & \text{if } i = j. \end{cases}$$

We have a direct corollary for $E_{1,1}$'s.

**Corollary A.3.** *We have:*

$$E_{1,1} \cdot E_{1,1} \cdot \ldots \cdot E_{1,1} = E_{1,1}.$$

### A.2. Evaluation of polynomials

Denote $\mathbb{R}[\mathbf{x}_1, \ldots, \mathbf{x}_n]$ be the ring of all polynomials with real coefficients in $n$ indeterminates $\mathbf{x}_1, \ldots, \mathbf{x}_n$.

**Definition A.4.** A *monomial* of $\mathbb{R}[\mathbf{x}_1, \ldots, \mathbf{x}_n]$ is a polynomial of $\mathbb{R}[\mathbf{x}_1, \ldots, \mathbf{x}_n]$ that has one term.

*Remark.* In some contexts, a monomial is defined as a polynomial that has one term with coefficient 1. We will use *both* of these definitions simultaneously.

**Proposition A.5.** $\mathbb{R}[\mathbf{x}_1, \ldots, \mathbf{x}_n]$ *is naturally a vector space over* $\mathbb{R}$. *It is an infinite-dimensional vector space; moreover, the set of all monomials with coefficient* 1 *in* $\mathbb{R}[\mathbf{x}_1, \ldots, \mathbf{x}_n]$ *is a basis for the vector space.*

*Remark.* For $f \in \mathbb{R}[\mathbf{x}_1, \ldots, \mathbf{x}_n]$, by saying monomials in $f$, we refer to all monomials that appeared in the expression of $f$.

Polynomial evaluation is computing of the value of a polynomial when the indeterminates are substituted for some values. We have the well-known result.

**Proposition A.6.** *Let* $f, g$ *be two polynomials of* $\mathbb{R}[\mathbf{x}_1, \ldots, \mathbf{x}_n]$. *If* $f, g$ *are equal at every evaluations, i.e.*

$$f(x_1, \ldots, x_n) = g(x_1, \ldots, x_n) \,, \ \forall (x_1, \ldots, x_n) \in \mathbb{R}^n, \tag{14}$$

*then* $f = g$. *In other words, the only polynomial of* $\mathbb{R}[\mathbf{x}_1, \ldots, \mathbf{x}_n]$, *that has* $\mathbb{R}^n$ *as its zero set, is the polynomial* $0 \in \mathbb{R}[\mathbf{x}_1, \ldots, \mathbf{x}_n]$.

*Remark.* The result still holds if $\mathbb{R}$ is replaced by an arbitrary infinite field, but does not hold if $\mathbb{R}$ is replaced by a finite field.

We have a direct corollary.

**Corollary A.7.** *Let* $f$ *be a nonzero polynomial of* $\mathbb{R}[\mathbf{x}_1, \ldots, \mathbf{x}_n]$. *Then there exists* $(x_1, \ldots, x_n) \in \mathbb{R}^n$ *such that* $f(x_1, \ldots, x_n) \neq 0$.

### A.3. Entries of tensors

**Proposition A.8.** *Let* $a = (a_i)_{1 \leqslant i \leqslant n}$ *and* $b = (b_i)_{1 \leqslant i \leqslant n}$ *be two vectors in* $\mathbb{R}^n$. *If:*

$$a_i \cdot b_j + a_j \cdot b_i = 0, \tag{15}$$

*for all* $1 \leqslant i, j \leqslant n$, *then* $a = 0$ *or* $b = 0$.

*Proof.* Assume that both of $a$ and $b$ are not equal to 0, then there exists $i, j$ such that $a_i$ and $b_j$ are non-zero. From Equation (15), we have:

$$a_i \cdot b_i + a_i \cdot b_i = 0, \tag{16}$$

so $a_i \cdot b_i = 0$. Since $a_i$ is non-zero, then $b_i = 0$. It implies that:

$$a_i \cdot b_j + a_j \cdot b_i = a_i \cdot b_j + 0 = a_i \cdot b_j \neq 0, \tag{17}$$

which contradicts to Equation (15). So at least one of $a$ and $b$ is equal to 0. $\qquad\square$

**Proposition A.9.** *Let* $A = (a_{ij})_{1 \leqslant i \leqslant m, 1 \leqslant j \leqslant n}$ *and* $B = (b_{ij})_{1 \leqslant i \leqslant m, 1 \leqslant j \leqslant n}$ *be two matrices in* $\mathbb{R}^{m \times n}$. *If:*

$$a_{ij} \cdot b_{kl} + a_{kj} \cdot b_{il} + a_{il} \cdot b_{kj} + a_{kl} \cdot b_{ij} = 0, \tag{18}$$

*for all* $1 \leqslant i, k \leqslant m$ *and* $1 \leqslant j, l \leqslant n$, *then* $A = 0$ *or* $B = 0$.

*Proof.* Consider Equation (18) when $1 \leqslant j = l \leqslant n$, we have:

$$0 = a_{ij} \cdot b_{kj} + a_{kj} \cdot b_{ij} + a_{ij} \cdot b_{kj} + a_{kj} \cdot b_{ij} \tag{19}$$
$$= 2 \cdot (a_{ij} \cdot b_{kj} + a_{kj} \cdot b_{ij}), \tag{20}$$

which means:

$$a_{ij} \cdot b_{kj} + a_{kj} \cdot b_{ij} = 0. \tag{21}$$

This holds for all $1 \leqslant i, k \leqslant m$. Apply Proposition A.8, we have $a_{ij} = 0$ for all $1 \leqslant i \leqslant m$, or $b_{ij} = 0$ for all $1 \leqslant i \leqslant m$, which means $A_{*,j} = 0$ or $B_{*,j} = 0$. This holds for all $1 \leqslant j \leqslant n$. Similarly, we have $A_{i,*} = 0$ or $B_{i,*} = 0$ for $1 \leqslant i \leqslant m$. Now, assume that, both of $A$ and $B$ are not equal to $0$, then there exists $i, j$ and $k, l$ such that $a_{ij}$ and $b_{kl}$ are non-zero. By previous observation, we have $B_{i,*} = B_{*,j} = A_{k,*} = A_{*,l} = 0$. It implies that:

$$a_{ij} \cdot b_{kl} + a_{kj} \cdot b_{il} + a_{il} \cdot b_{kj} + a_{kl} \cdot b_{ij} = a_{ij} \cdot b_{kl} + 0 + 0 + 0 = a_{ij} \cdot b_{kl} \neq 0, \tag{22}$$

which contradicts to Equation (18). So at least one of $A$ and $B$ is equal to $0$. $\square$

Proposition A.8 and Proposition A.9 are, respectively, one-dimensional and two-dimensional cases. By using the same arguments, we will obtain the $d$-dimensional version belows.

**Proposition A.10.** *Let $d$ be a positive integer and $n_1, n_2, \ldots, n_d$ be $d$ positive integers. Let:*

$$
\begin{aligned}
A &= (A_{i_1, i_2, \ldots, i_d})_{1 \leqslant i_1 \leqslant n_1, 1 \leqslant i_2 \leqslant n_2, \ldots, 1 \leqslant i_d \leqslant n_d} && \in \mathbb{R}^{n_1 \times n_2 \times \ldots \times n_d}, \\
B &= (B_{i_1, i_2, \ldots, i_d})_{1 \leqslant i_1 \leqslant n_1, 1 \leqslant i_2 \leqslant n_2, \ldots, 1 \leqslant i_d \leqslant n_d} && \in \mathbb{R}^{n_1 \times n_2 \times \ldots \times n_d}.
\end{aligned}
$$

*If for all $1 \leqslant i_1^0, i_1^1 \leqslant n_1, 1 \leqslant i_2^0, i_2^1 \leqslant n_2, \ldots, 1 \leqslant i_d^0, i_d^1 \leqslant n_d$, we have:*

$$\sum_{(\alpha_1, \ldots, \alpha_d) \in \{0,1\}^d} \left( A_{i_1^{\alpha_1}, i_2^{\alpha_2}, \ldots, i_d^{\alpha_d}} \right) \cdot \left( B_{i_1^{1-\alpha_1}, i_2^{1-\alpha_2}, \ldots, i_d^{1-\alpha_d}} \right) = 0, \tag{23}$$

*then $A = 0$ or $B = 0$.*

# B. Stable Polynomial Terms

Intuitively, a *stable polynomial term* is a polynomial in the entries of $U \in \mathcal{U}$ that is "stable" under the action of $G$ (see Definition B.1 below). The equivariant polynomial layers we aim to construct are linear combinations of these stable polynomial terms. In Subsection B.1, we provide a formal definition for stable polynomial terms as well as their properties. We will study the linear dependence of stable polynomial terms in the language of polynomial rings with real coefficients in Subsections B.2 and B.3. These properties play a central role in determining the parameter-sharing computation of equivariant polynomial layers in the next section.

## B.1. Definitions and basic properties

Recall the weight space $\mathcal{U}$ given by:

$$
\begin{aligned}
\mathcal{U} &= \mathcal{W} \times \mathcal{B}, && \text{where:} \\
\mathcal{W} &= \mathbb{R}^{w_L \times n_L \times n_{L-1}} \times \ldots \times \mathbb{R}^{w_2 \times n_2 \times n_1} \times \mathbb{R}^{w_1 \times n_1 \times n_0}, \\
\mathcal{B} &= \mathbb{R}^{b_L \times n_L \times 1} \times \ldots \times \mathbb{R}^{b_2 \times n_2 \times 1} \times \mathbb{R}^{b_1 \times n_1 \times 1}.
\end{aligned}
$$

Let us consider the case where the weight spaces have the same number of dimensions across all channels, which means $w_i = b_i = d$ for all $i$.

**Definition B.1** (Stable polynomial terms)**.** Let $U = ([W], [b])$ be an element of $\mathcal{U}$ with weights $[W] = \left([W]^{(L)}, \ldots, [W]^{(1)}\right)$ and biases $[b] = \left([b]^{(L)}, \ldots, [b]^{(1)}\right)$. For each $L \geqslant s > t \geqslant 0$, we define:

$$
\begin{aligned}
[W]^{(s,t)} &:= [W]^{(s)} \cdot [W]^{(s-1)} \cdot \ldots \cdot [W]^{(t+1)} && \in \mathbb{R}^{d \times n_s \times n_t}, \\
[Wb]^{(s,t)(t)} &:= [W]^{(s,t)} \cdot [b]^{(t)} && \in \mathbb{R}^{d \times n_s \times 1}.
\end{aligned} \tag{24}
$$

In addition, for each $L \geqslant s, t \geqslant 0$, and matrices $\Psi^{(s)(L,t)} \in \mathbb{R}^{1 \times n_L}$ and $\Psi^{(s,0)(L,t)} \in \mathbb{R}^{n_0 \times n_L}$, we also define

$$
\begin{aligned}
[bW]^{(s)(L,t)} &:= [b]^{(s)} \cdot \Psi^{(s)(L,t)} \cdot [W]^{(L,t)} && \in \mathbb{R}^{d \times n_s \times n_t}, \\
[WW]^{(s,0)(L,t)} &:= [W]^{(s,0)} \cdot \Psi^{(s,0)(L,t)} \cdot [W]^{(L,t)} && \in \mathbb{R}^{d \times n_s \times n_t}.
\end{aligned} \tag{25}
$$

The entries of the matrices $[W]^{(s,t)}$, $[Wb]^{(s,t)(t)}$, $[bW]^{(s)(L,t)}$ and $[WW]^{(s,0)(L,t)}$ defined above are called *stable polynomial terms* of $U$ under the action of $G$.

The following observations are direct implications from the definition.

- For all $L \geqslant s > t > r \geqslant 0$:

$$[W]^{(s,s-1)} = [W]^{(s)} \in \mathbb{R}^{d \times n_s \times n_{s-1}}, \tag{26}$$

and

$$[W]^{(s,t)} \cdot [W]^{(t,r)} = [W]^{(s,r)} \in \mathbb{R}^{d \times n_s \times n_r}, \tag{27}$$

by definition. For $g = \left(g^{(L)}, \ldots, g^{(0)}\right) \in \mathcal{G}_{\mathcal{U}}$:

$$[gW]^{(s,t)} = \left(g^{(s)}\right) \cdot [W]^{(s,t)} \cdot \left(g^{(t)}\right)^{-1} \in \mathbb{R}^{d \times n_s \times n_t}. \tag{28}$$

- If $g \in G$, then:

$$[gW]^{(L,t)} = [W]^{(L,t)} \cdot \left(g^{(t)}\right)^{-1} \in \mathbb{R}^{d \times n_L \times n_t} \tag{29}$$

$$[gW]^{(s,0)} = \left(g^{(s)}\right) \cdot [W]^{(s,0)} \in \mathbb{R}^{d \times n_s \times n_0}. \tag{30}$$

- For all $L \geqslant s > t > 0$, we have

$$[gW]^{(s,t)} \cdot [gb]^{(t)} = \left(g^{(s)}\right) \cdot [W]^{(s,t)} \cdot [b]^{(t)} \in \mathbb{R}^{d \times n_s \times 1} \tag{31}$$

- For all $L \geqslant s > 0$, $L > t \geqslant 0$ and $\Psi^{(s,0)(L,t)} \in \mathbb{R}^{d \times n_0 \times n_L}$, we have:

$$[gW]^{(s,0)} \cdot \Psi^{(s,0)(L,t)} \cdot [gW]^{(L,t)}$$
$$= \left(g^{(s)}\right) \cdot [W]^{(s,0)} \cdot \Psi^{(s,0)(L,t)} \cdot [W]^{(L,t)} \cdot \left(g^{(t)}\right)^{-1} \in \mathbb{R}^{d \times n_s \times n_t}. \tag{32}$$

In particular, if $t = s - 1$, we have:

$$[gW]^{(s,0)} \cdot \Psi^{(s,0)(L,s-1)} \cdot [gW]^{(L,s-1)}$$
$$= \left(g^{(s)}\right) \cdot [W]^{(s,0)} \cdot \Psi^{(s,0)(L,s-1)} \cdot [W]^{(L,s-1)} \cdot \left(g^{(s-1)}\right)^{-1} \in \mathbb{R}^{d \times n_s \times n_{s-1}}. \tag{33}$$

- For all $L \geqslant s > 0$ and $\Psi^{(s)(L,t)} \in \mathbb{R}^{d \times 1 \times n_L}$, we have:

$$[gb]^{(s)} \cdot \Psi^{(s)(L,t)} \cdot [gW]^{(L,t)}$$
$$= \left(g^{(s)}\right) \cdot [b]^{(s)} \cdot \Psi^{(s)(L,t)} \cdot [W]^{(L,t)} \cdot \left(g^{(t)}\right)^{-1} \in \mathbb{R}^{d \times n_s \times n_t}. \tag{34}$$

Based on the above observations, we can determine the image of the stable polynomial terms under the action of an element $g \in \mathcal{G}_{\mathcal{U}}$ as follows:

$$
\begin{aligned}
[gW]^{(s,t)} &= \left(g^{(s)}\right) \cdot [W]^{(s,t)} \cdot \left(g^{(t)}\right)^{-1}, \\
[gb]^{(s)} &= \left(g^{(s)}\right) \cdot [b]^{(s)}, \\
[gWgb]^{(s,t)(t)} &= \left(g^{(s)}\right) \cdot [Wb]^{(s,t)(t)}, \\
[gbgW]^{(s)(L,t)} &= \left(g^{(s)}\right) \cdot [bW]^{(s)(L,t)} \cdot \left(g^{(t)}\right)^{-1}, \\
[gWgW]^{(s,0)(L,t)} &= \left(g^{(s)}\right) \cdot [WW]^{(s,0)(L,t)} \cdot \left(g^{(t)}\right)^{-1}.
\end{aligned}
$$

In concrete, we have:

$$
\begin{aligned}
[gW]_{pq}^{(s,t)} &= \frac{d_p^{(s)}}{d_q^{(t)}} \cdot [W]_{\pi_s^{-1}(p),\pi_t^{-1}(q)}^{(s,t)}, \\
[gb]_p^{(s)} &= d_p^{(s)} \cdot [b]_{\pi_s^{-1}(p)}^{(s)}, \\
[gWgb]_p^{(s,t)(t)} &= d_p^{(s)} \cdot [Wb]_{\pi_s^{-1}(p)}^{(s,t)(t)}, \\
[gbgW]_{pq}^{(s)(L,t)} &= \frac{d_p^{(s)}}{d_q^{(t)}} \cdot [bW]_{\pi_s^{-1}(p),\pi_t^{-1}(q)}^{(s)(L,t)}, \\
[gWgW]^{(s,0)(L,t)} &= \frac{d_p^{(s)}}{d_q^{(t)}} \cdot [WW]_{\pi_s^{-1}(p),\pi_t^{-1}(q)}^{(s,0)(L,t)}.
\end{aligned}
$$

## B.2. Input weights as indeterminates

To simplify the technical difficulties, we consider the weight space $\mathcal{U}$ in the case where $d = 1$, i.e.,

$$
\begin{aligned}
\mathcal{U} &= \mathcal{W} \times \mathcal{B}, && \text{where:} \\
\mathcal{W} &= \mathbb{R}^{n_L \times n_{L-1}} & \times \quad \ldots \quad \times & \quad \mathbb{R}^{n_2 \times n_1} \quad \times \quad \mathbb{R}^{n_1 \times n_0}, \\
\mathcal{B} &= \mathbb{R}^{n_L \times 1} & \times \quad \ldots \quad \times & \quad \mathbb{R}^{n_2 \times 1} \quad \times \quad \mathbb{R}^{n_1 \times 1}.
\end{aligned}
$$

We introduce the set $I$ consists of indeterminates defined by:

$$
I := \{\mathbf{x}_{jk}^{(i)} : 1 \leqslant i \leqslant L, 1 \leqslant j \leqslant n_i, 1 \leqslant k \leqslant n_{i-1}\} \cup \{\mathbf{y}_j^{(i)} : 1 \leqslant i \leqslant L, 1 \leqslant j \leqslant n_i\}.
$$

We have $|I| = \dim \mathcal{U}$. Denote $R = \mathbb{R}[I]$, which is the ring of all polynomials with indeterminates are all elements of $I$. For $1 \leqslant i \leqslant L$, we define:

$$
\begin{aligned}
[\mathbf{W}]^{(i)} &:= \left(\mathbf{x}_{jk}^{(i)}\right)_{1 \leqslant j \leqslant n_i, 1 \leqslant k \leqslant n_{i-1}} & \in R^{n_i \times n_{i-1}}, \\
[\mathbf{b}]^{(i)} &:= \left(\mathbf{y}_j^{(i)}\right)_{1 \leqslant j \leqslant n_i} & \in R^{n_i \times 1},
\end{aligned}
$$

and

$$
\begin{aligned}
[\mathbf{W}]^{(s,t)} &:= [\mathbf{W}]^{(s)} \cdot [\mathbf{W}]^{(s-1)} \cdot \ldots \cdot [\mathbf{W}]^{(t+1)} & \in R^{n_s \times n_t}, \\
[\mathbf{Wb}]^{(s,t)(t)} &:= [\mathbf{W}]^{(s,t)} \cdot [\mathbf{b}]^{(t)} & \in R^{n_s \times 1}, \\
[\mathbf{bW}]^{(s)(L,t)} &:= [\mathbf{b}]^{(s)} \cdot \Psi^{(s)(L,t)} \cdot [\mathbf{W}]^{(L,t)} & \in R^{n_s \times n_t}, \\
[\mathbf{WW}]^{(s,0)(L,t)} &:= [\mathbf{W}]^{(s,0)} \cdot \Psi^{(s,0)(L,t)} \cdot [\mathbf{W}]^{(L,t)} & \in R^{n_s \times n_t}.
\end{aligned}
$$

with feasible indices $(s,t)$. The coefficients $\Psi^{(-)}$'s are fixed real matrices and they are omitted from the notations.

Note that the entries of these matrices are stable polynomial terms in which the entries of $U$ are now viewed as indeterminates of the polynomial ring $R$.

## B.3. Linear dependence of stable polynomial terms

In this subsection, we derive a necessary condition for the coefficients $\Phi_-$ and $\Psi^-$ such that the following linear combination of stable polynomial terms are identically zero:

$$
\begin{aligned}
\alpha(\Phi, \Psi) := \sum_{L \geqslant s > t \geqslant 0} \sum_{p=1}^{n_s} \sum_{q=1}^{n_t} \Phi_{(s,t):pq} \cdot [\mathbf{W}]_{pq}^{(s,t)} + \sum_{L \geqslant s > 0} \sum_{p=1}^{n_s} \Phi_{(s):p} \cdot [\mathbf{b}]_p^{(s)} \\
+ \sum_{L \geqslant s > t > 0} \sum_{p=1}^{n_s} \Phi_{(s,t)(t):p} \cdot [\mathbf{Wb}]_p^{(s,t)(t)}
\end{aligned}
$$

$$+ \sum_{L \geqslant s > 0} \sum_{L > t \geqslant 0} \sum_{p=1}^{n_s} \sum_{q=1}^{n_t} \Phi_{(s)(L,t):pq} \cdot [\mathbf{bW}]_{pq}^{(s)(L,t)}$$

$$+ \sum_{L \geqslant s > 0} \sum_{L > t \geqslant 0} \sum_{p=1}^{n_s} \sum_{q=1}^{n_t} \Phi_{(s,0)(L,t):pq} \cdot [\mathbf{WW}]_{pq}^{(s,0)(L,t)} + \Phi_1 \cdot 1. \tag{35}$$

Here, $\alpha$ is parameterized by $\Phi$ and $\Psi$, where $\Phi$ is a collection of real scalars $\Phi_-$'s appeared in the linear combination and $\Psi$ is a collection of real matrices $\Psi^-$'s that be used to define $[\mathbf{bW}]^{(-)}$'s and $[\mathbf{WW}]^{(-)}$'s. The index of each scalar $\Phi_-$ naturally presents its corresponding polynomial in $\alpha(\Phi, \Psi)$. This necessary and sufficient condition enables us to determine the equivariant polynomial map via parameter sharing later.

We first take a look at entries of $[\mathbf{W}]^{(-)}$'s, $[\mathbf{b}]^{(-)}$'s, $[\mathbf{Wb}]^{(-)}$'s, $[\mathbf{bW}]^{(-)}$'s, $[\mathbf{WW}]^{(-)}$'s. It is clear that for one of these matrices, its entries are *homogeneous polynomials with the same degree*. For example:

- $[\mathbf{W}]^{(-)}$: The polynomial $[\mathbf{W}]_{pq}^{(s,t)}$ has degree $s - t$. All of its monomial terms consist of one $\mathbf{x}_-^{(i)}$ for each $s \geqslant i > t$.

- $[\mathbf{b}]^{(-)}$: The polynomial $[\mathbf{b}]_p^{(s)}$ has degree 1. All of its monomial terms consist of one $\mathbf{y}_-^s$.

- $[\mathbf{Wb}]^{(-)}$: The polynomial $[\mathbf{Wb}]_p^{(s,t)(t)}$ has degree $s - t + 1$. All of its monomial terms consist of one $\mathbf{x}_-^{(i)}$ for each $s \geqslant i > t$ and one $\mathbf{y}_-^{(t)}$.

- $[\mathbf{bW}]^{(-)}$: The polynomial $[\mathbf{bW}]_p^{(s)(L,t)}$ has degree $L - t + 1$. All of its monomial terms consist of one $\mathbf{y}_-^{(s)}$ and one $\mathbf{x}_-^{(i)}$ for each $L \geqslant i > t$.

- $[\mathbf{WW}]^{(-)}$: The polynomial $[\mathbf{WW}]_{pq}^{(s,0)(L,t)}$ has degree $L + s - t$. All of its monomial terms consist of one $\mathbf{x}_-^{(i)}$ for each $s \geqslant i > 0$ and one $\mathbf{x}_-^{(i)}$ for each $L \geqslant i > t$.

- 1: The polynomial $1 \in R$.

By these above observations, we have:

- $[\mathbf{W}]^{(-)}, [\mathbf{WW}]^{(-)}$ : Each of the polynomials $[\mathbf{W}]_-^{(-)}$'s and $[\mathbf{WW}]_-^{(-)}$'s is 0 or a non-constant element in $R$, and it is a real polynomial of at least one indeterminate from $\mathbf{x}_-^{(-)}$'s.

- $[\mathbf{b}]^{(-)}$: Each of the polynomials $[\mathbf{b}]_-^{(-)}$'s is a non-constant element in $R$, and it is a real polynomial of one indeterminate from $\mathbf{y}_-^{(-)}$'s.

- $[\mathbf{Wb}]^{(-)}, [\mathbf{bW}]^{(-)}$: Each of the polynomials $[\mathbf{Wb}]_-^{(-)}$'s and $[\mathbf{bW}]_-^{(-)}$'s is 0 or a non-constant element in $R$, and it is a real polynomial of at least one indeterminate from $\mathbf{x}_-^{(-)}$'s and one indeterminate from $\mathbf{y}_-^{(-)}$'s.

Therefore, if $\alpha(\Phi, \Psi) = 0$, we must have

$$0 = \sum_{L \geqslant s > t \geqslant 0} \sum_{p=1}^{n_s} \sum_{q=1}^{n_t} \Phi_{(s,t):pq} \cdot [\mathbf{W}]_{pq}^{(s,t)} + \sum_{L \geqslant s > 0} \sum_{L > t \geqslant 0} \sum_{p=1}^{n_s} \sum_{q=1}^{n_t} \Phi_{(s,0)(L,t):pq} \cdot [\mathbf{WW}]_{pq}^{(s,0)(L,t)}, \tag{36}$$

$$0 = \sum_{L \geqslant s > 0} \sum_{p=1}^{n_s} \Phi_{(s):p} \cdot [\mathbf{b}]_p^{(s)}, \tag{37}$$

$$0 = \sum_{L \geqslant s > t > 0} \sum_{p=1}^{n_s} \Phi_{(s,t)(t):p} \cdot [\mathbf{Wb}]_p^{(s,t)(t)} + \sum_{L \geqslant s > 0} \sum_{L > t \geqslant 0} \sum_{p=1}^{n_s} \sum_{q=1}^{n_t} \Phi_{(s)(L,t):pq} \cdot [\mathbf{bW}]_{pq}^{(s)(L,t)}, \tag{38}$$

$$0 = \Phi_1 \cdot 1. \tag{39}$$

We induce the constraints on $\Phi$ and $\Psi$ in these above equations by using the fact that a set of distinct monomials of $R$ is a linear independent set (see Proposition A.5).

- Equation (36): Observe that

  - If $L \geqslant s > t \geqslant 0$ and $(s, t) \neq (L, 0)$, then the monomials $\mathbf{x}_-^{(s)} \cdot \ldots \cdot \mathbf{x}_-^{(t+1)}$'s only appear in the polynomials $[\mathbf{W}]_-^{(s,t)}$'s. They do not appear in the polynomials $[\mathbf{W}]_-^{(s',t')}$'s for all pairs $(s', t') \neq (s, t)$, and do not appear in the polynomials $[\mathbf{W}\mathbf{W}]_-^{(s',0)(L,t')}$'s for all pairs $(s', t')$.

  - If $L \geqslant s > 0$, $L > t \geqslant 0$, and $s \neq t$, then the monomials $\left(\mathbf{x}_-^{(s)} \cdot \ldots \cdot \mathbf{x}_-^{(1)}\right) \cdot \left(\mathbf{x}_-^{(L)} \cdot \ldots \cdot \mathbf{x}_-^{(t+1)}\right)$'s only appear in the polynomials $[\mathbf{W}\mathbf{W}]_-^{(s,0)(L,t)}$'s. They do not appear in the polynomials $[\mathbf{W}]_-^{(s',t')}$'s for all pairs $(s', t')$, and do not appear in the polynomials $[\mathbf{W}\mathbf{W}]_-^{(s',0)(L,t')}$'s for all pairs $(s', t') \neq (s, t)$.

So from Equation (36), it implies that

$$0 = \sum_{p=1}^{n_s} \sum_{q=1}^{n_t} \Phi_{(s,t):pq} \cdot [\mathbf{W}]_{pq}^{(s,t)}, \tag{40}$$

for all $(s, t) \neq (L, 0)$, and

$$0 = \sum_{p=1}^{n_s} \sum_{q=1}^{n_t} \Phi_{(s,0)(L,t):pq} \cdot [\mathbf{W}\mathbf{W}]_{pq}^{(s,0)(L,t)}, \tag{41}$$

for all $(s, t)$ that $s \neq t$. The rest of the terms in Equation (36) is

$$0 = \sum_{p=1}^{n_L} \sum_{q=1}^{n_0} \Phi_{(L,0):pq} \cdot [\mathbf{W}]_{pq}^{(L,0)} + \sum_{L>s>0} \sum_{p=1}^{n_s} \sum_{q=1}^{n_s} \Phi_{(s,0)(L,s):pq} \cdot [\mathbf{W}\mathbf{W}]_{pq}^{(s,0)(L,s)}. \tag{42}$$

- Equation (37): Since the set of all $[\mathbf{b}]_-^{(-)}$'s, which is the set of all monomials $\mathbf{y}_-^{(-)}$'s, is a linear independent set in $R$, it implies that

$$0 = \Phi_{(s):p}, \tag{43}$$

for all $L \geqslant s > 0$ and $1 \leqslant p \leqslant n_s$.

- Equation (38): Observe that

  - If $L > s > t > 0$, then the monomials $\left(\mathbf{x}_-^{(s)} \cdot \ldots \cdot \mathbf{x}_-^{(t+1)}\right) \cdot \mathbf{y}_-^{(t)}$'s only appear in the polynomials $[\mathbf{W}\mathbf{b}]_-^{(s,t)(t)}$'s. They do not appear in the polynomials $[\mathbf{W}\mathbf{b}]_-^{(s',t')(t')}$'s for all $L \geqslant s' > t' > 0$ that $(s', t') \neq (s, t)$, and do not appear in polynomials $[\mathbf{b}\mathbf{W}]_-^{(s')(L,t')}$'s for all $L \geqslant s' > 0$ and $L > t' \geqslant 0$.

  - If $L \geqslant s > 0$, $L > t \geqslant 0$, and $s \neq t$, then the monomials $\mathbf{y}_-^{(s)} \cdot \left(\mathbf{x}_-^{(L)} \cdot \ldots \cdot \mathbf{x}_-^{(t+1)}\right)$'s only appear in the polynomials $[\mathbf{b}\mathbf{W}]_-^{(s)(L,t)}$'s. They do not appear in the polynomials $[\mathbf{W}\mathbf{b}]_-^{(L,t')(t')}$'s for all $L > t' > 0$, and do not appear in the polynomials $[\mathbf{b}\mathbf{W}]_-^{(s')(L,t')}$'s for all pairs $(s', t') \neq (s, t)$.

  - If $L > t > 0$, then the monomials $\mathbf{y}_-^{(t)} \cdot \left(\mathbf{x}_-^{(L)} \cdot \ldots \cdot \mathbf{x}_-^{(t+1)}\right)$'s only appear in the polynomials $[\mathbf{W}\mathbf{b}]_-^{(L,t)(t)}$'s and appear in the polynomials $[\mathbf{b}\mathbf{W}]_-^{(t)(L,t)}$'s. They do not appear in the polynomials $[\mathbf{W}\mathbf{b}]_-^{(L,t')(t')}$'s for all $L > t' > 0$ that $t' \neq t$, and do not appear in polynomials $[\mathbf{b}\mathbf{W}]_-^{(s')(L,t')}$'s for all pair $(s', t')$ that $t' \neq t$.

So from Equation (38), it implies that

$$0 = \sum_{p=1}^{n_s} \Phi_{(s,t)(t):p} \cdot [\mathbf{W}\mathbf{b}]_p^{(s,t)(t)}, \tag{44}$$

for all $L > s > t > 0$, and

$$0 = \sum_{p=1}^{n_s} \sum_{q=1}^{n_t} \Phi_{(s)(L,t):pq} \cdot [\mathbf{bW}]_{pq}^{(s)(L,t)} \tag{45}$$

for all $(s, t)$ that $s \neq t$, and

$$0 = \sum_{p=1}^{n_L} \Phi_{(L,t)(t):p} \cdot [\mathbf{Wb}]_p^{(L,t)(t)} + \sum_{p=1}^{n_t} \sum_{q=1}^{n_t} \Phi_{(t)(L,t):pq} \cdot [\mathbf{bW}]_{pq}^{(t)(L,t)} \tag{46}$$

for all $L > t > 0$.

- Equation (39): Clearly, it implies that

$$0 = \Phi_1. \tag{47}$$

There are 8 equations, Equations (40)-(47), that are derived. In Equations (43) and (47), the corresponding $\Phi_-$'s are directly characterized, and in Equations (40), (41), (42), (44), (45), (46), the corresponding $\Phi_-$'s are not. We will characterize the $\Phi_-$'s and $\Psi_-$'s in Equations (40), (41), (44) and (45) below by Lemma B.2, Lemma B.3 and Lemma B.5, respectively. These lemma are stated as below (their proofs will be postponed to Section B.4).

**Lemma B.2.** *For a pair $(s, t)$ such that $L \geqslant s > t \geqslant 0$ and $(s, t) \neq (L, 0)$, if*

$$0 = \sum_{p=1}^{n_s} \sum_{q=1}^{n_t} \Phi_{(s,t):pq} \cdot [\mathbf{W}]_{pq}^{(s,t)}, \tag{48}$$

*then $\Phi_{(s,t):pq} = 0$ for all $p, q$.*

**Lemma B.3.** *For a pair $(s, t)$ such that $L \geqslant s > 0$, $L > t \geqslant 0$, if*

$$0 = \sum_{p=1}^{n_s} \sum_{q=1}^{n_t} \Phi_{(s,0)(L,t):pq} \cdot [\mathbf{WW}]_{pq}^{(s,0)(L,t)}, \tag{49}$$

*then $\Phi_{(s,0)(L,t):pq} = 0$ for all $p, q$ or $\Psi^{(s,0)(L,t)} = 0$.*

**Lemma B.4.** *For a pair $(s, t)$ such that $L \geqslant s > t > 0$, if*

$$0 = \sum_{p=1}^{n_s} \Phi_{(s,t)(t):p} \cdot [\mathbf{Wb}]_p^{(s,t)(t)}, \tag{50}$$

*then $\Phi_{(s,t)(t):p} = 0$ for all $p$.*

**Lemma B.5.** *For a pair $(s, t)$ such that $L \geqslant s > 0$, $L > t \geqslant 0$, if*

$$0 = \sum_{p=1}^{n_s} \sum_{q=1}^{n_t} \Phi_{(s)(L,t):pq} \cdot [\mathbf{bW}]_{pq}^{(s)(L,t)}, \tag{51}$$

*then $\Phi_{(s)(L,t):pq} = 0$ for all $p, q$ or $\Psi^{(s)(L,t)} = 0$.*

*Remark.* The reason that we skip the characterizations of $\Phi_-$'s and $\Psi_-$'s in Equations (42) and (46) is they can be concretely characterized. For instance, consider the case when $n_L = \ldots = n_2 = n_1 = n_0 = 1$. From Equation (42), we have

$$0 = \sum_{p=1}^{1} \sum_{q=1}^{1} \Phi_{(L,0):pq} \cdot [\mathbf{W}]_{pq}^{(L,0)} + \sum_{L>s>0} \sum_{p=1}^{1} \sum_{q=1}^{1} \Phi_{(s,0)(L,s):pq} \cdot [\mathbf{WW}]_{pq}^{(s,0)(L,s)}$$
$$= \Phi_{(L,0):1,1} \cdot [\mathbf{W}]_{1,1}^{(L,0)} + \sum_{L>s>0} \Phi_{(s,0)(L,s):1,1} \cdot [\mathbf{WW}]_{1,1}^{(s,0)(L,s)}$$

$$= \Phi_{(L,0):1,1} \cdot \left( \mathbf{x}_{1,1}^{(L)} \cdot \ldots \cdot \mathbf{x}_{1,1}^{(2)} \cdot \mathbf{x}_{1,1}^{(1)} \right)$$
$$+ \sum_{L>s>0} \Phi_{(s,0)(L,s):1,1} \cdot \left( \mathbf{x}_{1,1}^{(s)} \cdot \ldots \cdot \mathbf{x}_{1,1}^{(2)} \cdot \mathbf{x}_{1,1}^{(1)} \right) \cdot \Psi^{(s,0)(L,s)} \cdot \left( \mathbf{x}_{1,1}^{(L)} \cdot \ldots \cdot \mathbf{x}_{1,1}^{(s+2)} \cdot \mathbf{x}_{1,1}^{(s+1)} \right)$$
$$= \Phi_{(L,0):1,1} \cdot \left( \mathbf{x}_{1,1}^{(L)} \cdot \ldots \cdot \mathbf{x}_{1,1}^{(2)} \cdot \mathbf{x}_{1,1}^{(1)} \right)$$
$$+ \sum_{L>s>0} \Phi_{(s,0)(L,s):1,1} \cdot \Psi^{(s,0)(L,s)} \cdot \left( \mathbf{x}_{1,1}^{(s)} \cdot \ldots \cdot \mathbf{x}_{1,1}^{(2)} \cdot \mathbf{x}_{1,1}^{(1)} \right) \cdot \left( \mathbf{x}_{1,1}^{(L)} \cdot \ldots \cdot \mathbf{x}_{1,1}^{(s+2)} \cdot \mathbf{x}_{1,1}^{(s+1)} \right)$$
$$= \Phi_{(L,0):1,1} \cdot \left( \mathbf{x}_{1,1}^{(L)} \cdot \ldots \cdot \mathbf{x}_{1,1}^{(2)} \cdot \mathbf{x}_{1,1}^{(1)} \right)$$
$$+ \sum_{L>s>0} \Phi_{(s,0)(L,s):1,1} \cdot \Psi^{(s,0)(L,s)} \cdot \left( \mathbf{x}_{1,1}^{(L)} \cdot \ldots \cdot \mathbf{x}_{1,1}^{(2)} \cdot \mathbf{x}_{1,1}^{(1)} \right)$$
$$= \left( \Phi_{(L,0):1,1} + \sum_{L>s>0} \Phi_{(s,0)(L,s):1,1} \cdot \Psi^{(s,0)(L,s)} \right) \cdot \left( \mathbf{x}_{1,1}^{(L)} \cdot \ldots \cdot \mathbf{x}_{1,1}^{(2)} \cdot \mathbf{x}_{1,1}^{(1)} \right).$$

It implies that

$$0 = \Phi_{(L,0):1,1} + \sum_{L>s>0} \Phi_{(s,0)(L,s):1,1} \cdot \Psi^{(s,0)(L,s)}. \tag{52}$$

From Equation (52), we can not derive a more concrete relation on the $\Phi_-$'s and $\Psi_-$'s. Similarly, from Equation (46), we have

$$0 = \sum_{p=1}^{1} \Phi_{(L,t)(t):p} \cdot [\mathbf{Wb}]_p^{(L,t)(t)} + \sum_{p=1}^{1} \sum_{q=1}^{1} \Phi_{(t)(L,t):pq} \cdot [\mathbf{bW}]_{pq}^{(t)(L,t)}$$
$$= \Phi_{(L,t)(t):1} \cdot [\mathbf{Wb}]_1^{(L,t)(t)} + \Phi_{(t)(L,t):1,1} \cdot [\mathbf{bW}]_{1,1}^{(t)(L,t)}$$
$$= \Phi_{(L,t)(t):1} \cdot \left( \mathbf{x}_{1,1}^{(L)} \cdot \ldots \cdot \mathbf{x}_{1,1}^{(t+2)} \cdot \mathbf{x}_{1,1}^{(t+1)} \right) \cdot \left( \mathbf{y}_{1,1}^{(t+1)} \right)$$
$$+ \Phi_{(t)(L,t):1,1} \cdot \left( \mathbf{y}_{1,1}^{(t+1)} \right) \cdot \Psi^{(t)(L,t)} \cdot \left( \mathbf{x}_{1,1}^{(L)} \cdot \ldots \cdot \mathbf{x}_{1,1}^{(t+2)} \cdot \mathbf{x}_{1,1}^{(t+1)} \right)$$
$$= \Phi_{(L,t)(t):1} \cdot \left( \mathbf{x}_{1,1}^{(L)} \cdot \ldots \cdot \mathbf{x}_{1,1}^{(t+2)} \cdot \mathbf{x}_{1,1}^{(t+1)} \right) \cdot \left( \mathbf{y}_{1,1}^{(t+1)} \right)$$
$$+ \Phi_{(t)(L,t):1,1} \cdot \Psi^{(t)(L,t)} \cdot \left( \mathbf{y}_{1,1}^{(t+1)} \right) \cdot \left( \mathbf{x}_{1,1}^{(L)} \cdot \ldots \cdot \mathbf{x}_{1,1}^{(t+2)} \cdot \mathbf{x}_{1,1}^{(t+1)} \right)$$
$$= \Phi_{(L,t)(t):1} \cdot \left( \mathbf{x}_{1,1}^{(L)} \cdot \ldots \cdot \mathbf{x}_{1,1}^{(t+2)} \cdot \mathbf{x}_{1,1}^{(t+1)} \right) \cdot \left( \mathbf{y}_{1,1}^{(t+1)} \right)$$
$$+ \Phi_{(t)(L,t):1,1} \cdot \Psi^{(t)(L,t)} \cdot \left( \mathbf{x}_{1,1}^{(L)} \cdot \ldots \cdot \mathbf{x}_{1,1}^{(t+2)} \cdot \mathbf{x}_{1,1}^{(t+1)} \right) \cdot \left( \mathbf{y}_{1,1}^{(t+1)} \right)$$
$$= \left( \Phi_{(L,t)(t):1} + \Phi_{(t)(L,t):1,1} \cdot \Psi^{(t)(L,t)} \right) \cdot \left( \mathbf{x}_{1,1}^{(L)} \cdot \ldots \cdot \mathbf{x}_{1,1}^{(t+2)} \cdot \mathbf{x}_{1,1}^{(t+1)} \right) \cdot \left( \mathbf{y}_{1,1}^{(t+1)} \right)$$

It implies that

$$0 = \Phi_{(L,t)(t):1} + \Phi_{(t)(L,t):1,1} \cdot \Psi^{(t)(L,t)}. \tag{53}$$

From Equation (53), we can not derive a more concrete relation on the $\Phi_-$'s and $\Psi_-$'s.

Combining the discussions above, we obtain the following necessary for the coefficients $\Phi$ and $\Psi$ such that $\alpha(\Phi, \Psi) = 0$.

**Theorem B.6.** *Let $\alpha(\Phi, \Psi)$ be a polynomial given in equation 35. If $\alpha(\Phi, \Psi) = 0$, then the following condition holds:*

1. *For all $L \geqslant s > t \geqslant 0$ with $(s, t) \neq (L, 0)$, and for all $p, q$, we have*

$$\Phi_{(s,t):pq} = 0. \tag{54}$$

2. *For all $L \geqslant s > 0$, $L > t \geqslant 0$ with $s \neq t$, we have*

$$\Phi_{(s,0)(L,t):pq} = 0, \tag{55}$$

   *for all $p, q$, or*

$$\Psi^{(s,0)(L,t)} = 0. \tag{56}$$

3. *We have*

$$0 = \sum_{p=1}^{n_L} \sum_{q=1}^{n_0} \Phi_{(L,0):pq} \cdot [\mathbf{W}]_{pq}^{(L,0)} + \sum_{L>s>0} \sum_{p=1}^{n_s} \sum_{q=1}^{n_s} \Phi_{(s,0)(L,s):pq} \cdot [\mathbf{WW}]_{pq}^{(s,0)(L,s)}. \tag{57}$$

4. *For all $L > s > t > 0$, and for all $p$, we have*

$$\Phi_{(s,t)(t):p} = 0. \tag{58}$$

5. *For all $L \geqslant s > 0$, $L > t \geqslant 0$ with $s \neq t$, we have*

$$\Phi_{(s)(L,t):pq} = 0, \tag{59}$$

   *for all $p, q$, or*

$$\Psi^{(s)(L,t)} = 0. \tag{60}$$

6. *For all $L > t > 0$, we have*

$$0 = \sum_{p=1}^{n_L} \Phi_{(L,t)(t):p} \cdot [\mathbf{Wb}]_p^{(L,t)(t)} + \sum_{p=1}^{n_t} \sum_{q=1}^{n_t} \Phi_{(t)(L,t):pq} \cdot [\mathbf{bW}]_{pq}^{(t)(L,t)}. \tag{61}$$

7. *For all $L \geqslant s > 0$ and for all $p$, we have*

$$\Phi_{(s):p} = 0. \tag{62}$$

8. *We have*

$$\Phi_1 = 0. \tag{63}$$

*Proof.* By the previous observations, $\alpha(\Phi, \Psi) = 0$ implies four Equations (36)-(39). These equations imply Equations (40)-(47). The proof of all parts in Theorem B.6 are as follows.

1. It comes from Equation (40) and Lemma B.2.

2. It comes from Equation (41) and Lemma B.3.

3. It comes from Equation (42).

4. It comes from Equation (44) and Lemma B.4

5. It comes from Equation (45) and Lemma B.5.

6. It comes from Equation (46).

7. It comes from Equation (43).

8. It comes from Equation (47).

The proof is finsihed. □

The following corollary is a direct consequence of Theorem B.6.

**Corollary B.7.** *If for some* $\Phi, \Phi'$, *and* $\Psi$, *we have* $\alpha(\Phi, \Psi) = \alpha(\Phi', \Psi)$, *then:*

1. *For all* $L \geqslant s > t \geqslant 0$ *with* $(s,t) \neq (L,0)$, *and for all* $p, q$, *we have*

$$\Phi_{(s,t):pq} = \Phi'_{(s,t):pq}. \tag{64}$$

2. *For all* $L \geqslant s > 0$, $L > t \geqslant 0$ *with* $s \neq t$, *we have*

$$\Phi_{(s,0)(L,t):pq} = \Phi'_{(s,0)(L,t):pq}, \tag{65}$$

   *for all* $p, q$, *or*

$$\Psi^{(s,0)(L,t)} = 0. \tag{66}$$

3. *We have*

$$\sum_{p=1}^{n_L} \sum_{q=1}^{n_0} \Phi_{(L,0):pq} \cdot [\mathbf{W}]_{pq}^{(L,0)} + \sum_{L>s>0} \sum_{p=1}^{n_s} \sum_{q=1}^{n_s} \Phi_{(s,0)(L,s):pq} \cdot [\mathbf{WW}]_{pq}^{(s,0)(L,s)} \tag{67}$$

$$= \sum_{p=1}^{n_L} \sum_{q=1}^{n_0} \Phi'_{(L,0):pq} \cdot [\mathbf{W}]_{pq}^{(L,0)} + \sum_{L>s>0} \sum_{p=1}^{n_s} \sum_{q=1}^{n_s} \Phi'_{(s,0)(L,s):pq} \cdot [\mathbf{WW}]_{pq}^{(s,0)(L,s)}. \tag{68}$$

4. *For all* $L > s > t > 0$, *and for all* $p$, *we have*

$$\Phi_{(s,t)(t):p} = \Phi'_{(s,t)(t):p}. \tag{69}$$

5. *For all* $L \geqslant s > 0$, $L > t \geqslant 0$ *with* $s \neq t$, *we have*

$$\Phi_{(s)(L,t):pq} = \Phi'_{(s)(L,t):pq}, \tag{70}$$

   *for all* $p, q$, *or*

$$\Psi^{(s)(L,t)} = 0. \tag{71}$$

6. *For all* $L > t > 0$, *we have*

$$\sum_{p=1}^{n_L} \Phi_{(L,t)(t):p} \cdot [\mathbf{Wb}]_p^{(L,t)(t)} + \sum_{p=1}^{n_t} \sum_{q=1}^{n_t} \Phi_{(t)(L,t):pq} \cdot [\mathbf{bW}]_{pq}^{(t)(L,t)} \tag{72}$$

$$= \sum_{p=1}^{n_L} \Phi'_{(L,t)(t):p} \cdot [\mathbf{Wb}]_p^{(L,t)(t)} + \sum_{p=1}^{n_t} \sum_{q=1}^{n_t} \Phi'_{(t)(L,t):pq} \cdot [\mathbf{bW}]_{pq}^{(t)(L,t)}. \tag{73}$$

7. *For all* $L \geqslant s > 0$ *and for all* $p$, *we have*

$$\Phi_{(s):p} = \Phi'_{(s):p}. \tag{74}$$

8. *We have*

$$\Phi_1 = \Phi'_1. \tag{75}$$

*Proof.* Since

$$0 = \alpha(\Phi, \Psi) - \alpha(\Phi', \Psi) = \alpha(\Phi - \Phi', \Psi),$$

so the results come directly from Theorem B.6. □

## B.4. Proofs of Lemmas B.2-B.5

The proofs of Lemmas B.2 through B.5 are directly from the coefficient comparison of two equal polynomials and the following lemma. We omit the proofs of Lemmas B.2 through B.5 and show only the proof for Lemma B.8.

**Lemma B.8.** *For a feasible tuple* $(s, t, p, q)$*, if*

$$[\mathbf{W}\mathbf{W}]_{pq}^{(s,0)(L,t)} = \left( [\mathbf{W}]^{(s,0)} \cdot \Psi^{(s,0)(L,t)} \cdot [\mathbf{W}]^{(L,t)} \right)_{pq} = 0 \in R. \tag{76}$$

*Then* $\Psi^{(s,0)(L,t)} = 0 \in \mathbb{R}^{n_0 \times n_L}$.

*Proof.* We first consider the case where $s = L$ and $t = 0$, then the case $s \leqslant t$, and finally the case $s > t$. Note that, the proof for the last case will be by combining the arguments of the first two cases.

**Case** $s = L$ **and** $t = 0$. From Equation (76), we have

$$[\mathbf{W}\mathbf{W}]_{pq}^{(L,0)(L,0)} = \left( [\mathbf{W}]^{(L,0)} \cdot \Psi^{(L,0)(L,0)} \cdot [\mathbf{W}]^{(L,0)} \right)_{pq} = 0, \tag{77}$$

which means

$$\left( [\mathbf{W}]^{(L)} \cdot \ldots \cdot [\mathbf{W}]^{(1)} \cdot \Psi^{(L,0)(L,0)} \cdot [\mathbf{W}]^{(L)} \cdot \ldots \cdot [\mathbf{W}]^{(1)} \right)_{pq} = 0. \tag{78}$$

Here, we consider three cases, where $L = 1$, $L = 2$ and $L \geqslant 3$.

- Case $L = 1$. Equation (78) becomes

$$\left( [\mathbf{W}]^{(1)} \cdot \Psi^{(1,0)(1,0)} \cdot [\mathbf{W}]^{(1)} \right)_{pq} = 0. \tag{79}$$

By Proposition A.1, this is equivalent to

$$[\mathbf{W}]_{p,*}^{(1)} \cdot \Psi^{(1,0)(1,0)} \cdot [\mathbf{W}]_{*,q}^{(1)} = 0, \tag{80}$$

which is:

$$\sum_{i=1}^{n_0} \sum_{j=0}^{n_1} \Psi_{ij}^{(1,0)(1,0)} \cdot \mathbf{x}_{p,i}^{(1)} \cdot \mathbf{x}_{j,q}^{(1)} = 0. \tag{81}$$

Since the LHS of Equation (81) is a linear combination between distinct monomials

$$\mathbf{x}_{p,i}^{(1)} \cdot \mathbf{x}_{j,q}^{(1)}, \tag{82}$$

for $1 \leqslant i \leqslant n_0$ and $1 \leqslant j \leqslant n_1$, so it implies that

$$\Psi_{ij}^{(1,0)(1,0)} = 0, \tag{83}$$

for all $1 \leqslant i \leqslant n_0$ and $1 \leqslant j \leqslant n_1$, which means

$$\Psi^{(1,0)(1,0)} = 0. \tag{84}$$

- Case $L = 2$. Equation (78) becomes

$$\left( [\mathbf{W}]^{(2)} \cdot [\mathbf{W}]^{(1)} \cdot \Psi^{(2,0)(2,0)} \cdot [\mathbf{W}]^{(2)} \cdot [\mathbf{W}]^{(1)} \right)_{pq} = 0. \tag{85}$$

By Proposition A.1, this is equivalent to

$$[\mathbf{W}]_{p,*}^{(2)} \cdot [\mathbf{W}]^{(1)} \cdot \Psi^{(2,0)(2,0)} \cdot [\mathbf{W}]^{(2)} \cdot [\mathbf{W}]_{*,q}^{(1)} = 0, \tag{86}$$

which is

$$\left( [\mathbf{W}]_{p,*}^{(2)} \cdot [\mathbf{W}]_{*,1}^{(1)} , \ [\mathbf{W}]_{p,*}^{(2)} \cdot [\mathbf{W}]_{*,2}^{(1)} , \ \ldots , \ [\mathbf{W}]_{p,*}^{(2)} \cdot [\mathbf{W}]_{*,n_0}^{(1)} \right) \cdot \Psi^{(2,0)(2,0)} \tag{87}$$

$$\cdot \left( [\mathbf{W}]_{1,*}^{(2)} \cdot [\mathbf{W}]_{*,q}^{(1)} , \ [\mathbf{W}]_{2,*}^{(2)} \cdot [\mathbf{W}]_{*,q}^{(1)} , \ \ldots , \ [\mathbf{W}]_{n_2,*}^{(2)} \cdot [\mathbf{W}]_{*,q}^{(1)} \right)^{\top} = 0, \tag{88}$$

which is

$$\sum_{i=1}^{n_0} \sum_{j=0}^{n_2} \Psi_{ij}^{(2,0)(2,0)} \cdot \left( [\mathbf{W}]_{p,*}^{(2)} \cdot [\mathbf{W}]_{*,i}^{(1)} \right) \cdot \left( [\mathbf{W}]_{j,*}^{(2)} \cdot [\mathbf{W}]_{*,q}^{(1)} \right) = 0. \tag{89}$$

Since the LHS of Equation (89) is a linear combination between elements of a linear independent collection of $R$, which are:

$$\left( [\mathbf{W}]_{p,*}^{(2)} \cdot [\mathbf{W}]_{*,i}^{(1)} \right) \cdot \left( [\mathbf{W}]_{j,*}^{(2)} \cdot [\mathbf{W}]_{*,q}^{(1)} \right), \tag{90}$$

for $1 \leqslant i \leqslant n_0$ and $1 \leqslant j \leqslant n_2$, so it implies that

$$\Psi_{ij}^{(2,0)(2,0)} = 0, \tag{91}$$

for all $1 \leqslant i \leqslant n_0$ and $1 \leqslant j \leqslant n_2$, which means

$$\Psi^{(2,0)(2,0)} = 0. \tag{92}$$

- Case $L \geqslant 3$. Equation (78) becomes

$$\left( [\mathbf{W}]^{(L)} \cdot \ldots \cdot [\mathbf{W}]^{(1)} \cdot \Psi^{(L,0)(L,0)} \cdot [\mathbf{W}]^{(L)} \cdot \ldots \cdot [\mathbf{W}]^{(1)} \right)_{pq} = 0. \tag{93}$$

It is noted that the matrices $[\mathbf{W}]^{(L-1)}, \ldots, [\mathbf{W}]^{(2)}$ can be substituted for some matrix units at $1^{\text{st}}$-row and $1^{\text{st}}$-column such that the product

$$[\mathbf{W}]^{(L-1)} \cdot \ldots \cdot [\mathbf{W}]^{(2)} \tag{94}$$

becomes a matrix unit at $1^{\text{st}}$-row and $1^{\text{st}}$-column. Substitute this in Equation (93), we have

$$\left( [\mathbf{W}]^{(L)} \cdot E_{11} \cdot [\mathbf{W}]^{(1)} \cdot \Psi^{(L,0)(L,0)} \cdot [\mathbf{W}]^{(L)} \cdot E_{11} \cdot [\mathbf{W}]^{(1)} \right)_{pq} = 0. \tag{95}$$

By Proposition A.1, this is equivalent to

$$[\mathbf{W}]_{p,*}^{(L)} \cdot E_{11} \cdot [\mathbf{W}]^{(1)} \cdot \Psi^{(L,0)(L,0)} \cdot [\mathbf{W}]^{(L)} \cdot E_{11} \cdot [\mathbf{W}]_{*,q}^{(1)} = 0. \tag{96}$$

which can be rewritten as

$$[\mathbf{W}]_{p,1}^{(L)} \cdot [\mathbf{W}]_{1,*}^{(1)} \cdot \Psi^{(L,0)(L,0)} \cdot [\mathbf{W}]_{*,1}^{(L)} \cdot [\mathbf{W}]_{1,q}^{(1)} = 0, \tag{97}$$

or

$$\mathbf{x}_{p,1}^{(L)} \cdot [\mathbf{W}]_{1,*}^{(1)} \cdot \Psi^{(L,0)(L,0)} \cdot [\mathbf{W}]_{*,1}^{(L)} \cdot \mathbf{x}_{1,q}^{(1)} = 0. \tag{98}$$

Since the LHS of Equation (98) is a polynomial in $R$, we have

$$[\mathbf{W}]_{1,*}^{(1)} \cdot \Psi^{(L,0)(L,0)} \cdot [\mathbf{W}]_{*,1}^{(L)} = 0, \tag{99}$$

which is

$$\sum_{i=1}^{n_0} \sum_{j=0}^{n_L} \Psi_{ij}^{(L,0)(L,0)} \cdot \mathbf{x}_{1,i}^{(1)} \cdot \mathbf{x}_{j,1}^{(L)} = 0. \tag{100}$$

Since the LHS of Equation (100) is a linear combination between distinct monomials

$$\mathbf{x}_{1,i}^{(1)} \cdot \mathbf{x}_{j,1}^{(L)}, \tag{101}$$

for $1 \leqslant i \leqslant n_0$ and $1 \leqslant j \leqslant n_L$, so it implies that

$$\Psi_{ij}^{(L,0)(L,0)} = 0, \tag{102}$$

for all $1 \leqslant i \leqslant n_0$ and $1 \leqslant j \leqslant n_L$, which means

$$\Psi^{(L,0)(L,0)} = 0. \tag{103}$$

In conclusion, for all cases of $L$, we have

$$\Psi^{(L,0)(L,0)} = 0. \tag{104}$$

We finish the proof of the case where $s = L$ and $t = 0$.

**Case $s \leqslant t$.** From Equation (76), we have

$$[\mathbf{W}\mathbf{W}]_{pq}^{(s,0)(L,t)} = \left( [\mathbf{W}]^{(s,0)} \cdot \Psi^{(s,0)(L,t)} \cdot [\mathbf{W}]^{(L,t)} \right)_{pq} = 0, \tag{105}$$

which means

$$\left( [\mathbf{W}]^{(s)} \cdot \ldots \cdot [\mathbf{W}]^{(1)} \cdot \Psi^{(s,0)(L,t)} \cdot [\mathbf{W}]^{(L)} \cdot \ldots \cdot [\mathbf{W}]^{(t+1)} \right)_{pq} = 0. \tag{106}$$

Since $s \leqslant t$, the $[\mathbf{W}]^-$'s that are in the product $[\mathbf{W}]^{(s)} \cdot \ldots \cdot [\mathbf{W}]^{(1)}$, and the $[\mathbf{W}]^-$'s that are in the product $[\mathbf{W}]^{(L)} \cdot \ldots \cdot [\mathbf{W}]^{(t+1)}$, are distinct. By directly multiplying $[\mathbf{W}]^{(s)} \cdot \ldots \cdot [\mathbf{W}]^{(1)}$ and $[\mathbf{W}]^{(L)} \cdot \ldots \cdot [\mathbf{W}]^{(t+1)}$, we can write these two products in the forms

$$\begin{aligned}
[\mathbf{W}]^{(s)} \cdot \ldots \cdot [\mathbf{W}]^{(1)} &= \left( \mathbf{f}_{ij}^{(s)} \right)_{1 \leqslant i \leqslant n_s, 1 \leqslant j \leqslant n_0} &\in R^{n_s \times n_0}, \\
[\mathbf{W}]^{(L)} \cdot \ldots \cdot [\mathbf{W}]^{(t+1)} &= \left( \mathbf{g}_{ij}^{(t)} \right)_{1 \leqslant i \leqslant n_L, 1 \leqslant j \leqslant n_t} &\in R^{n_L \times n_t},
\end{aligned} \tag{107}$$

where all $\mathbf{f}_-^{(s)}$'s are nonzero and all $\mathbf{g}_-^{(t)}$'s are nonzero. Moreover, $\mathbf{f}_-^{(s)}$'s are real polynomials of indeterminates $\mathbf{x}_-^{(1)}$'s, $\mathbf{x}_-^{(2)}$'s, $\ldots$, $\mathbf{x}_-^{(s)}$'s. Similarly, $\mathbf{g}_-^{(t)}$'s are real polynomials of indeterminates $\mathbf{x}_-^{(L)}$'s, $\mathbf{x}_-^{(L-1)}$'s, $\ldots$, $\mathbf{x}_-^{(t+1)}$'s. Now, Equation (106) is equal to

$$\left( \left( \mathbf{f}_{ij}^{(s)} \right)_{1 \leqslant i \leqslant n_s, 1 \leqslant j \leqslant n_0} \cdot \Psi^{(s,0)(L,t)} \cdot \left( \mathbf{g}_{ij}^{(t)} \right)_{1 \leqslant i \leqslant n_L, 1 \leqslant j \leqslant n_t} \right)_{pq} = 0. \tag{108}$$

By Proposition A.1, this is equivalent to

$$\left( \mathbf{f}_{p,1}^{(s)}, \mathbf{f}_{p,2}^{(s)}, \ldots, \mathbf{f}_{p,n_0}^{(s)} \right) \cdot \Psi^{(s,0)(L,t)} \cdot \left( \mathbf{g}_{1,q}^{(t)}, \mathbf{g}_{2,q}^{(t)}, \ldots, \mathbf{f}_{n_L,q}^{(t)} \right)^\top = 0, \tag{109}$$

which is

$$\sum_{i=1}^{n_0} \sum_{j=0}^{n_L} \Psi_{ij}^{(s,0)(L,t)} \cdot \mathbf{f}_{p,i}^{(s)} \cdot \mathbf{g}_{j,q}^{(t)} = 0. \tag{110}$$

Since the LHS of Equation (110) is a linear combination between elements of a linear independent collection of $R$, which are

$$\mathbf{f}_{p,i}^{(s)} \cdot \mathbf{g}_{j,q}^{(t)}, \tag{111}$$

for $1 \leqslant i \leqslant n_0$ and $1 \leqslant j \leqslant n_L$. The linear dependency comes from the distinction between indeterminates of $\mathbf{f}_-^{(-)}$ and $\mathbf{g}_-^{(-)}$. It implies that

$$\Psi_{ij}^{(s,0)(L,t)} = 0, \tag{112}$$

for all $1 \leqslant i \leqslant n_0$ and $1 \leqslant j \leqslant n_L$, which means

$$\Psi^{(s,0)(L,t)} = 0. \tag{113}$$

We finish the proof of the case where $s \leqslant t$.

**Case $s > t$.** From Equation (76), we have:

$$[\mathbf{W}\mathbf{W}]_{pq}^{(s,0)(L,t)} = \left( [\mathbf{W}]^{(s,0)} \cdot \Psi^{(s,0)(L,t)} \cdot [\mathbf{W}]^{(L,t)} \right)_{pq} = 0, \tag{114}$$

which means

$$\left( [\mathbf{W}]^{(s)} \cdot \ldots \cdot [\mathbf{W}]^{(1)} \cdot \Psi^{(s,0)(L,t)} \cdot [\mathbf{W}]^{(L)} \cdot \ldots \cdot [\mathbf{W}]^{(t+1)} \right)_{pq} = 0. \tag{115}$$

Assume that $\Psi^{(s,0)(L,t)} \neq 0 \in \mathbb{R}^{n_0 \times n_L}$. Observe that

$$[\mathbf{W}]^{(s)} \cdot \ldots \cdot [\mathbf{W}]^{(1)} \cdot \Psi^{(s,0)(L,t)} \cdot [\mathbf{W}]^{(L)} \cdot \ldots \cdot [\mathbf{W}]^{(t+1)} \tag{116}$$

$$= \left( [\mathbf{W}]^{(s)} \cdot \ldots \cdot [\mathbf{W}]^{(t+1)} \cdot [\mathbf{W}]^{(t)} \cdot \ldots \cdot [\mathbf{W}]^{(1)} \right) \tag{117}$$

$$\cdot \Psi^{(s,0)(L,t)} \cdot$$

$$\left( [\mathbf{W}]^{(L)} \cdot \ldots \cdot [\mathbf{W}]^{(s+1)} \cdot [\mathbf{W}]^{(s)} \cdot \ldots \cdot [\mathbf{W}]^{(t+1)} \right)$$

$$= \left( [\mathbf{W}]^{(s)} \cdot \ldots \cdot [\mathbf{W}]^{(t+1)} \right) \tag{118}$$

$$\cdot \left( [\mathbf{W}]^{(t)} \cdot \ldots \cdot [\mathbf{W}]^{(1)} \cdot \Psi^{(s,0)(L,t)} \cdot [\mathbf{W}]^{(L)} \cdot \ldots \cdot [\mathbf{W}]^{(s+1)} \right) \cdot$$

$$\left( [\mathbf{W}]^{(s)} \cdot \ldots \cdot [\mathbf{W}]^{(t+1)} \right).$$

In the second term of Equation (118),

$$[\mathbf{W}]^{(t)} \cdot \ldots \cdot [\mathbf{W}]^{(1)} \cdot \Psi^{(s,0)(L,t)} \cdot [\mathbf{W}]^{(L)} \cdot \ldots \cdot [\mathbf{W}]^{(s+1)}. \tag{119}$$

Since $s > t$, all $[\mathbf{W}]^{(-)}$'s in Equation (119) are distinct. And since $\Psi^{(s,0)(L,t)} \neq 0 \in \mathbb{R}^{n_0 \times n_L}$, with the same argument as in **Case $s \leqslant t$**, we have

$$[\mathbf{W}]^{(t)} \cdot \ldots \cdot [\mathbf{W}]^{(1)} \cdot \Psi^{(s,0)(L,t)} \cdot [\mathbf{W}]^{(L)} \cdot \ldots \cdot [\mathbf{W}]^{(s+1)} \neq 0 \in R^{n_t \times n_s}. \tag{120}$$

Moreover, it can be written in the form

$$[\mathbf{W}]^{(t)} \cdot \ldots \cdot [\mathbf{W}]^{(1)} \cdot \Psi^{(s,0)(L,t)} \cdot [\mathbf{W}]^{(L)} \cdot \ldots \cdot [\mathbf{W}]^{(s+1)} \tag{121}$$

$$= \left( \mathbf{c}_{ij}^{(st)} \right)_{1 \leqslant i \leqslant n_t, 1 \leqslant j \leqslant n_s} \in R^{n_t \times n_s}, \tag{122}$$

where all $\mathbf{c}_-^{(st)}$'s are real polynomials of indeterminates $\mathbf{x}_-^{(1)}$'s, $\mathbf{x}_-^{(2)}$'s, ..., $\mathbf{x}_-^{(t)}$'s and $\mathbf{x}_-^{(L)}$'s, $\mathbf{x}_-^{(L-1)}$'s, ..., $\mathbf{x}_-^{(s+1)}$'s, and at least one of them is a nonzero element in $R$. By Corollary A.7, indeterminates $\mathbf{x}_-^{(1)}$'s, $\mathbf{x}_-^{(2)}$'s, ..., $\mathbf{x}_-^{(t)}$'s and $\mathbf{x}_-^{(L)}$'s, $\mathbf{x}_-^{(L-1)}$'s, ..., $\mathbf{x}_-^{(s+1)}$'s can be substituted for some real values to make

$$[\mathbf{W}]^{(t)} \cdot \ldots \cdot [\mathbf{W}]^{(1)} \cdot \Psi^{(s,0)(L,t)} \cdot [\mathbf{W}]^{(L)} \cdot \ldots \cdot [\mathbf{W}]^{(s+1)}, \tag{123}$$

become a nonzero matrix of $\mathbb{R}^{n_t \times n_s}$. We denote this nonzero matrix by $\overline{\Psi}^{(s,0)(L,t)} \in \mathbb{R}^{n_t \times n_s}$. Note that, since $s > t$, the substitution only applies for indeterminates in $[\mathbf{W}]^{(1)}, [\mathbf{W}]^{(2)}, \ldots, [\mathbf{W}]^{(t)}$ and $[\mathbf{W}]^{(L)}, [\mathbf{W}]^{(L-1)}, \ldots, [\mathbf{W}]^{(s+1)}$. In other words, it does not apply for $[\mathbf{W}]^{(s)}, \ldots, [\mathbf{W}]^{(t+1)}$. So, with the above substitution, Equation (116) becomes

$$[\mathbf{W}]^{(s)} \cdot \ldots \cdot [\mathbf{W}]^{(1)} \cdot \Psi^{(s,0)(L,t)} \cdot [\mathbf{W}]^{(L)} \cdot \ldots \cdot [\mathbf{W}]^{(t+1)}$$

$$= [\mathbf{W}]^{(s)} \cdot \ldots \cdot [\mathbf{W}]^{(t+1)} \cdot \overline{\Psi}^{(s,0)(L,t)} \cdot [\mathbf{W}]^{(s)} \cdot \ldots \cdot [\mathbf{W}]^{(t+1)}. \tag{124}$$

Note that, since $s > t$, there is at least one $[\mathbf{W}]^{(-)}$ in the product $[\mathbf{W}]^{(s)} \cdot \ldots \cdot [\mathbf{W}]^{(t+1)}$. Combine with $\overline{\Psi}^{(s,0)(L,t)} \neq 0$, by applying the argument of **Case** $s = L$ **and** $t = 0$, we have

$$\left( [\mathbf{W}]^{(s)} \cdot \ldots \cdot [\mathbf{W}]^{(1)} \cdot \Psi^{(s,0)(L,t)} \cdot [\mathbf{W}]^{(L)} \cdot \ldots \cdot [\mathbf{W}]^{(t+1)} \right)_{pq}$$

$$= \left( [\mathbf{W}]^{(s)} \cdot \ldots \cdot [\mathbf{W}]^{(t+1)} \cdot \overline{\Psi}^{(s,0)(L,t)} \cdot [\mathbf{W}]^{(s)} \cdot \ldots \cdot [\mathbf{W}]^{(t+1)} \right)_{pq} \neq 0, \tag{125}$$

which contradicts to Equation (115). In conclusion

$$\Psi^{(s,0)(L,t)} = 0. \tag{126}$$

We finish the proof of the case where $s > t$.

In summary, we did consider all possible cases. The proof is finished. $\qquad\square$

## C. Equivariant Polynomial Layers

We now proceed to construct a $G$-equivariant polynomial layer, denoted as $E$. These layers serve as the fundamental building blocks for our MAGEP-NFNs. Our strategy is as follows: we first express $E$ as a polynomial layer that is a linear combination of stable polynomial terms (Subsection C.1). We then find the equivariant maps among these polynomial layers using the parameter sharing mechanism. Equivariance in Machine Learning is used in various context, such as Euclidean equivariance (Tran et al., 2024c; Ruhe et al., 2023), equivariant metanetwork (Tran et al., 2024d; Zhou et al., 2024c;b), Optimal Transport (Tran et al., 2024b; 2025a;c;b), etc.

### C.1. Equivariant layer as a linear combination of stable polynomial terms

For two weight spaces $\mathcal{U}$ and $\mathcal{U}'$ with the same number of layers $L$ as well as the same number of channels at $i^{\text{th}}$ layer $n_i$,

$$
\begin{aligned}
\mathcal{U} &= \mathcal{W} \times \mathcal{B}, && \text{where:}\\
\mathcal{W} &= \mathbb{R}^{w_L \times n_L \times n_{L-1}} & \times \quad \ldots \quad \times \quad \mathbb{R}^{w_2 \times n_2 \times n_1} & \times \quad \mathbb{R}^{w_1 \times n_1 \times n_0},\\
\mathcal{B} &= \mathbb{R}^{b_L \times n_L \times 1} & \times \quad \ldots \quad \times \quad \mathbb{R}^{b_2 \times n_2 \times 1} & \times \quad \mathbb{R}^{b_1 \times n_1 \times 1},
\end{aligned}
$$

and

$$
\begin{aligned}
\mathcal{U}' &= \mathcal{W}' \times \mathcal{B}', && \text{where:}\\
\mathcal{W}' &= \mathbb{R}^{w'_L \times n_L \times n_{L-1}} & \times \quad \ldots \quad \times \quad \mathbb{R}^{w'_2 \times n_2 \times n_1} & \times \quad \mathbb{R}^{w'_1 \times n_1 \times n_0},\\
\mathcal{B}' &= \mathbb{R}^{b'_L \times n_L \times 1} & \times \quad \ldots \quad \times \quad \mathbb{R}^{b'_2 \times n_2 \times 1} & \times \quad \mathbb{R}^{b'_1 \times n_1 \times 1}.
\end{aligned}
$$

We want to build a map $E \colon \mathcal{U} \to \mathcal{U}'$ such that $E$ is $G$-equivariant, where:

$$G = \{\mathrm{id}_{\mathcal{G}_{n_L}}\} \times \mathcal{G}_{n_{L-1}}^{>0} \times \ldots \times \mathcal{G}_{n_1}^{>0} \times \{\mathrm{id}_{\mathcal{G}_{n_0}}\}. \tag{127}$$

Let us consider a polynomial map $E \colon \mathcal{U} \to \mathcal{U}'$ such that, for input $U = ([W], [b])$, each entry of the output $E(U) = ([E(W)], [E(b)])$ is a linear combinations of stable polynomial terms, i.e the entries of $[W]^{(s,t)}, [b]^{(s)}, [Wb]^{(s,t)(t)}$, $[bW]^{(s)(L,t)}, [WW]^{(s,0)(L,t)}$, together with a bias. In concrete:

$$[E(W)]_{jk}^{(i)} := \sum_{L \geqslant s > t \geqslant 0} \sum_{p=1}^{n_s} \sum_{q=1}^{n_t} \Phi_{(s,t):pq}^{(i):jk} \cdot [W]_{pq}^{(s,t)} + \sum_{L \geqslant s > 0} \sum_{p=1}^{n_s} \Phi_{(s):p}^{(i):jk} \cdot [b]_p^{(s)}$$

$$+ \sum_{L \geqslant s > t > 0} \sum_{p=1}^{n_s} \Phi_{(s,t)(t):p}^{(i):jk} \cdot [Wb]_p^{(s,t)(t)}$$

$$+ \sum_{L \geqslant s > 0} \sum_{L > t \geqslant 0} \sum_{p=1}^{n_s} \sum_{q=1}^{n_t} \Phi_{(s)(L,t):pq}^{(i):jk} \cdot [bW]_{pq}^{(s)(L,t)}$$

$$+ \sum_{L \geqslant s > 0} \sum_{L > t \geqslant 0} \sum_{p=1}^{n_s} \sum_{q=1}^{n_t} \Phi_{(s,0)(L,t):pq}^{(i):jk} \cdot [WW]_{pq}^{(s,0)(L,t)} + \Phi_1^{(i):jk},$$

$$[E(b)]_j^{(i)} := \sum_{L \geqslant s > t \geqslant 0} \sum_{p=1}^{n_s} \sum_{q=1}^{n_t} \Phi_{(s,t):pq}^{(i):j} \cdot [W]_{pq}^{(s,t)} + \sum_{L \geqslant s > 0} \sum_{p=1}^{n_s} \Phi_{(s):p}^{(i):j} \cdot [b]_p^{(s)}$$

$$+ \sum_{L \geqslant s > t > 0} \sum_{p=1}^{n_s} \Phi_{(s,t)(t):p}^{(i):j} \cdot [Wb]_p^{(s,t)(t)}$$

$$+ \sum_{L \geqslant s > 0} \sum_{L > t \geqslant 0} \sum_{p=1}^{n_s} \sum_{q=1}^{n_t} \Phi_{(s)(L,t):pq}^{(i):j} \cdot [bW]_{pq}^{(s)(L,t)}$$

$$+ \sum_{L \geqslant s > 0} \sum_{L > t \geqslant 0} \sum_{p=1}^{n_s} \sum_{q=1}^{n_t} \Phi_{(s,0)(L,t):pq}^{(i):j} \cdot [WW]_{pq}^{(s,0)(L,t)} + \Phi_1^{(i):j}.$$

All $\Phi$'s are in $\mathbb{R}^{d' \times d}$, except the biases $\Phi_1^-$'s are in $\mathbb{R}^{d' \times 1}$. In summary, $E$ is parameterized by $\Phi_-$'s and $\Psi_-$'s.

In order to be $G$-equivariant, the polynomial map $E$ must satisfy the condition $E(gU) = gE(U)$ for all $g \in \mathcal{G}_\mathcal{U}$ and $U \in \mathcal{U}$. In the following subsections, we derive the computations of $E(gU)$ and $gE(U)$, and compare them in order to obtain all possible $G$-equivariant polynomial maps among those considered.

**C.2. Compute $E(gU)$**

We have

$$[E(gW)]_{jk}^{(i)} = \sum_{L \geqslant s > t \geqslant 0} \sum_{p=1}^{n_s} \sum_{q=1}^{n_t} \Phi_{(s,t):pq}^{(i):jk} \cdot [gW]_{pq}^{(s,t)}$$

$$+ \sum_{L \geqslant s > 0} \sum_{p=1}^{n_s} \Phi_{(s):p}^{(i):jk} \cdot [gb]_p^{(s)}$$

$$+ \sum_{L \geqslant s > t > 0} \sum_{p=1}^{n_s} \Phi_{(s,t)(t):p}^{(i):jk} \cdot [gWgb]_p^{(s,t)(t)}$$

$$+ \sum_{L \geqslant s > 0} \sum_{L > t \geqslant 0} \sum_{p=1}^{n_s} \sum_{q=1}^{n_t} \Phi_{(s)(L,t):pq}^{(i):jk} \cdot [gbgW]_{pq}^{(s)(L,t)}$$

$$+ \sum_{L \geqslant s > 0} \sum_{L > t \geqslant 0} \sum_{p=1}^{n_s} \sum_{q=1}^{n_t} \Phi_{(s,0)(L,t):pq}^{(i):jk} \cdot [gWgW]_{pq}^{(s,0)(L,t)}$$

$$+ \Phi_1^{(i):jk}$$

$$= \sum_{L \geqslant s > t \geqslant 0} \sum_{p=1}^{n_s} \sum_{q=1}^{n_t} \Phi_{(s,t):pq}^{(i):jk} \cdot \frac{d_p^{(s)}}{d_q^{(t)}} \cdot [W]_{\pi_s^{-1}(p), \pi_t^{-1}(q)}^{(s,t)}$$

$$+ \sum_{L \geqslant s > 0} \sum_{p=1}^{n_s} \Phi_{(s):p}^{(i):jk} \cdot d_p^{(s)} \cdot [b]_{\pi_s^{-1}(p)}^{(s)}$$

$$+ \sum_{L \geqslant s > t > 0} \sum_{p=1}^{n_s} \Phi_{(s,t)(t):p}^{(i):jk} \cdot d_p^{(s)} \cdot [Wb]_{\pi_s^{-1}(p)}^{(s,t)(t)}$$

$$+ \sum_{L \geqslant s > 0} \sum_{L > t \geqslant 0} \sum_{p=1}^{n_s} \sum_{q=1}^{n_t} \Phi_{(s)(L,t):pq}^{(i):jk} \cdot \frac{d_p^{(s)}}{d_q^{(t)}} \cdot [bW]_{\pi_s^{-1}(p),\pi_t^{-1}(q)}^{(s)(L,t)}$$

$$+ \sum_{L \geqslant s > 0} \sum_{L > t \geqslant 0} \sum_{p=1}^{n_s} \sum_{q=1}^{n_t} \Phi_{(s,0)(L,t):pq}^{(i):jk} \cdot \frac{d_p^{(s)}}{d_q^{(t)}} \cdot [WW]_{\pi_s^{-1}(p),\pi_t^{-1}(q)}^{(s,0)(L,t)}$$

$$+ \Phi_1^{(i):jk}$$

$$= \sum_{L \geqslant s > t \geqslant 0} \sum_{p=1}^{n_s} \sum_{q=1}^{n_t} \Phi_{(s,t):\pi_s(p)\pi_t(q)}^{(i):jk} \cdot \frac{d_{\pi_s(p)}^{(s)}}{d_{\pi_t(q)}^{(t)}} \cdot [W]_{pq}^{(s,t)}$$

$$+ \sum_{L \geqslant s > 0} \sum_{p=1}^{n_s} \Phi_{(s):\pi_s(p)}^{(i):jk} \cdot d_{\pi_s(p)}^{(s)} \cdot [b]_p^{(s)}$$

$$+ \sum_{L \geqslant s > t > 0} \sum_{p=1}^{n_s} \Phi_{(s,t)(t):\pi_s(p)}^{(i):jk} \cdot d_{\pi_s(p)}^{(s)} \cdot [Wb]_p^{(s,t)(t)}$$

$$+ \sum_{L \geqslant s > 0} \sum_{L > t \geqslant 0} \sum_{p=1}^{n_s} \sum_{q=1}^{n_t} \Phi_{(s)(L,t):\pi_s(p)\pi_t(q)}^{(i):jk} \cdot \frac{d_{\pi_s(p)}^{(s)}}{d_{\pi_t(q)}^{(t)}} \cdot [bW]_{pq}^{(s)(L,t)}$$

$$+ \sum_{L \geqslant s > 0} \sum_{L > t \geqslant 0} \sum_{p=1}^{n_s} \sum_{q=1}^{n_t} \Phi_{(s,0)(L,t):\pi_s(p)\pi_t(q)}^{(i):jk} \cdot \frac{d_{\pi_s(p)}^{(s)}}{d_{\pi_t(q)}^{(t)}} \cdot [WW]_{pq}^{(s,0)(L,t)}$$

$$+ \Phi_1^{(i):jk}$$

$$[E(gb)]_j^{(i)} = \sum_{L \geqslant s > t \geqslant 0} \sum_{p=1}^{n_s} \sum_{q=1}^{n_t} \Phi_{(s,t):pq}^{(i):j} \cdot [gW]_{pq}^{(s,t)}$$

$$+ \sum_{L \geqslant s > 0} \sum_{p=1}^{n_s} \Phi_{(s):p}^{(i):j} \cdot [gb]_p^{(s)}$$

$$+ \sum_{L \geqslant s > t > 0} \sum_{p=1}^{n_s} \Phi_{(s,t)(t):p}^{(i):j} \cdot [gWgb]_p^{(s,t)(t)}$$

$$+ \sum_{L \geqslant s > 0} \sum_{L > t \geqslant 0} \sum_{p=1}^{n_s} \sum_{q=1}^{n_t} \Phi_{(s)(L,t):pq}^{(i):j} \cdot [gbgW]_{pq}^{(s)(L,t)}$$

$$+ \sum_{L \geqslant s > 0} \sum_{L > t \geqslant 0} \sum_{p=1}^{n_s} \sum_{q=1}^{n_t} \Phi_{(s,0)(L,t):pq}^{(i):j} \cdot [gWgW]_{pq}^{(s,0)(L,t)}$$

$$+ \Phi_1^{(i):j}$$

$$= \sum_{L \geqslant s > t \geqslant 0} \sum_{p=1}^{n_s} \sum_{q=1}^{n_t} \Phi_{(s,t):pq}^{(i):j} \cdot \frac{d_p^{(s)}}{d_q^{(t)}} \cdot [W]_{\pi_s^{-1}(p),\pi_t^{-1}(q)}^{(s,t)}$$

$$+ \sum_{L \geqslant s > 0} \sum_{p=1}^{n_s} \Phi_{(s):p}^{(i):j} \cdot d_p^{(s)} \cdot [b]_{\pi_s^{-1}(p)}^{(s)}$$

$$+ \sum_{L \geqslant s > t > 0} \sum_{p=1}^{n_s} \Phi_{(s,t)(t):p}^{(i):j} \cdot d_p^{(s)} \cdot [Wb]_{\pi_s^{-1}(p)}^{(s,t)(t)}$$

$$
+ \sum_{L \geqslant s > 0} \sum_{L > t \geqslant 0} \sum_{p=1}^{n_s} \sum_{q=1}^{n_t} \Phi^{(i):j}_{(s)(L,t):pq} \cdot \frac{d^{(s)}_p}{d^{(t)}_q} \cdot [bW]^{(s)(L,t)}_{\pi_s^{-1}(p), \pi_t^{-1}(q)}
$$

$$
+ \sum_{L \geqslant s > 0} \sum_{L > t \geqslant 0} \sum_{p=1}^{n_s} \sum_{q=1}^{n_t} \Phi^{(i):j}_{(s,0)(L,t):pq} \cdot \frac{d^{(s)}_p}{d^{(t)}_q} \cdot [WW]^{(s,0)(L,t)}_{\pi_s^{-1}(p), \pi_t^{-1}(q)}
$$

$$
+ \Phi^{(i):j}_1
$$

$$
= \sum_{L \geqslant s > t \geqslant 0} \sum_{p=1}^{n_s} \sum_{q=1}^{n_t} \Phi^{(i):j}_{(s,t):\pi_s(p)\pi_t(q)} \cdot \frac{d^{(s)}_{\pi_s(p)}}{d^{(t)}_{\pi_t(q)}} \cdot [W]^{(s,t)}_{pq}
$$

$$
+ \sum_{L \geqslant s > 0} \sum_{p=1}^{n_s} \Phi^{(i):j}_{(s):\pi_s(p)} \cdot d^{(s)}_{\pi_s(p)} \cdot [b]^{(s)}_p
$$

$$
+ \sum_{L \geqslant s > t > 0} \sum_{p=1}^{n_s} \Phi^{(i):j}_{(s,t)(t):\pi_s(p)} \cdot d^{(s)}_{\pi_s(p)} \cdot [Wb]^{(s,t)(t)}_p
$$

$$
+ \sum_{L \geqslant s > 0} \sum_{L > t \geqslant 0} \sum_{p=1}^{n_s} \sum_{q=1}^{n_t} \Phi^{(i):j}_{(s)(L,t):\pi_s(p)\pi_t(q)} \cdot \frac{d^{(s)}_{\pi_s(p)}}{d^{(t)}_{\pi_t(q)}} \cdot [bW]^{(s)(L,t)}_{pq}
$$

$$
+ \sum_{L \geqslant s > 0} \sum_{L > t \geqslant 0} \sum_{p=1}^{n_s} \sum_{q=1}^{n_t} \Phi^{(i):j}_{(s,0)(L,t):\pi_s(p)\pi_t(q)} \cdot \frac{d^{(s)}_{\pi_s(p)}}{d^{(t)}_{\pi_t(q)}} \cdot [WW]^{(s,0)(L,t)}_{pq}
$$

$$
+ \Phi^{(i):j}_1.
$$

Note that, we can move around the $\pi$'s in above equations since the group $G$ satisfy that: $G \cap \mathcal{P}_i$ is trivial (for $i = 0$ or $i = L$) or the whole $\mathcal{P}_i$ (for $0 < i < L$).

## C.3. Compute $gE(U)$

We have:

$$
[g(E(W))]^{(i)}_{jk} = \frac{d^{(i)}_j}{d^{(i-1)}_k} \cdot [W']^{(i)}_{\pi_i^{-1}(j)\pi_{i-1}^{-1}(k)}
$$

$$
= \sum_{L \geqslant s > t \geqslant 0} \sum_{p=1}^{n_s} \sum_{q=1}^{n_t} \frac{d^{(i)}_j}{d^{(i-1)}_k} \cdot \Phi^{(i):\pi_i^{-1}(j)\pi_{i-1}^{-1}(k)}_{(s,t):pq} \cdot [W]^{(s,t)}_{pq}
$$

$$
+ \sum_{L \geqslant s > 0} \sum_{p=1}^{n_s} \frac{d^{(i)}_j}{d^{(i-1)}_k} \cdot \Phi^{(i):\pi_i^{-1}(j)\pi_{i-1}^{-1}(k)}_{(s):p} \cdot [b]^{(s)}_p
$$

$$
+ \sum_{L \geqslant s > t > 0} \sum_{p=1}^{n_s} \frac{d^{(i)}_j}{d^{(i-1)}_k} \cdot \Phi^{(i):\pi_i^{-1}(j)\pi_{i-1}^{-1}(k)}_{(s,t)(t):p} \cdot [Wb]^{(s,t)(t)}_p
$$

$$
+ \sum_{L \geqslant s > 0} \sum_{L > t \geqslant 0} \sum_{p=1}^{n_s} \sum_{q=1}^{n_t} \frac{d^{(i)}_j}{d^{(i-1)}_k} \cdot \Phi^{(i):\pi_i^{-1}(j)\pi_{i-1}^{-1}(k)}_{(s)(L,t):pq} \cdot [bW]^{(s)(L,t)}_{pq}
$$

$$
+ \sum_{L \geqslant s > 0} \sum_{L > t \geqslant 0} \sum_{p=1}^{n_s} \sum_{q=1}^{n_t} \frac{d^{(i)}_j}{d^{(i-1)}_k} \cdot \Phi^{(i):\pi_i^{-1}(j)\pi_{i-1}^{-1}(k)}_{(s,0)(L,t):pq} \cdot [WW]^{(s,0)(L,t)}_{pq}
$$

$$
+ \frac{d^{(i)}_j}{d^{(i-1)}_k} \cdot \Phi^{(i):\pi_i^{-1}(j)\pi_{i-1}^{-1}(k)}_1
$$

$$[g(E(b))]_j^{(i)} = d_j^{(i)} \cdot [b']_{\pi_i^{-1}(j)}^{(i)}$$

$$= \sum_{L \geqslant s > t \geqslant 0} \sum_{p=1}^{n_s} \sum_{q=1}^{n_t} d_j^{(i)} \cdot \Phi_{(s,t):pq}^{(i):\pi_i^{-1}(j)} \cdot [W]_{pq}^{(s,t)}$$

$$+ \sum_{L \geqslant s > 0} \sum_{p=1}^{n_s} d_j^{(i)} \cdot \Phi_{(s):p}^{(i):\pi_i^{-1}(j)} \cdot [b]_p^{(s)}$$

$$+ \sum_{L \geqslant s > t > 0} \sum_{p=1}^{n_s} d_j^{(i)} \cdot \Phi_{(s,t)(t):p}^{(i):\pi_i^{-1}(j)} \cdot [Wb]_p^{(s,t)(t)}$$

$$+ \sum_{L \geqslant s > 0} \sum_{L > t \geqslant 0} \sum_{p=1}^{n_s} \sum_{q=1}^{n_t} d_j^{(i)} \cdot \Phi_{(s)(L,t):pq}^{(i):\pi_i^{-1}(j)} \cdot [bW]_{pq}^{(s)(L,t)}$$

$$+ \sum_{L \geqslant s > 0} \sum_{L > t \geqslant 0} \sum_{p=1}^{n_s} \sum_{q=1}^{n_t} d_j^{(i)} \cdot \Phi_{(s,0)(L,t):pq}^{(i):\pi_i^{-1}(j)} \cdot [WW]_{pq}^{(s,0)(L,t)}$$

$$+ d_j^{(i)} \cdot \Phi_1^{(i):\pi_i^{-1}(j)}$$

### C.4. Compare $E(gU)$ and $gE(U)$

Since $E(gU) = gE(U)$, from Corollary B.7, the parameters $\Phi_-$'s have to satisfy these following conditions:

1. For all $L \geqslant s > t \geqslant 0$ with $(s,t) \neq (L,0)$, and for all $p, q$, we have

$$\Phi_{(s,t):\pi_s(p)\pi_t(q)}^{(i):jk} \cdot \frac{d_{\pi_s(p)}^{(s)}}{d_{\pi_t(q)}^{(t)}} = \frac{d_j^{(i)}}{d_k^{(i-1)}} \cdot \Phi_{(s,t):pq}^{(i):\pi_i^{-1}(j)\pi_{i-1}^{-1}(k)}$$

$$\Phi_{(s,t):\pi_s(p)\pi_t(q)}^{(i):j} \cdot \frac{d_{\pi_s(p)}^{(s)}}{d_{\pi_t(q)}^{(t)}} = d_j^{(i)} \cdot \Phi_{(s,t):pq}^{(i):\pi_i^{-1}(j)}$$

2. For all $L \geqslant s > 0$, $L > t \geqslant 0$ with $s \neq t$, we have

$$\Phi_{(s,0)(L,t):\pi_s(p)\pi_t(q)}^{(i):jk} \cdot \frac{d_{\pi_s(p)}^{(s)}}{d_{\pi_t(q)}^{(t)}} = \frac{d_j^{(i)}}{d_k^{(i-1)}} \cdot \Phi_{(s,0)(L,t):pq}^{(i):\pi_i^{-1}(j)\pi_{i-1}^{-1}(k)}$$

$$\Phi_{(s,0)(L,t):\pi_s(p)\pi_t(q)}^{(i):j} \cdot \frac{d_{\pi_s(p)}^{(s)}}{d_{\pi_t(q)}^{(t)}} = d_j^{(i)} \cdot \Phi_{(s,0)(L,t):pq}^{(i):\pi_i^{-1}(j)}$$

3. We have

$$\sum_{p=1}^{n_L} \sum_{q=1}^{n_0} \Phi_{(L,0):\pi_L(p)\pi_0(q)}^{(i):jk} \cdot \frac{d_{\pi_L(p)}^{(L)}}{d_{\pi_0(q)}^{(0)}} \cdot [W]_{pq}^{(L,0)}$$

$$+ \sum_{L > s > 0} \sum_{p=1}^{n_s} \sum_{q=1}^{n_s} \Phi_{(s,0)(L,s):\pi_s(p)\pi_s(q)}^{(i):jk} \cdot \frac{d_{\pi_s(p)}^{(s)}}{d_{\pi_s(q)}^{(s)}} \cdot [WW]_{pq}^{(s,0)(L,s)}$$

$$= \sum_{p=1}^{n_L} \sum_{q=1}^{n_0} \frac{d_j^{(i)}}{d_k^{(i-1)}} \cdot \Phi_{(L,0):pq}^{(i):\pi_i^{-1}(j)\pi_{i-1}^{-1}(k)} \cdot [W]_{pq}^{(L,0)}$$

$$+ \sum_{L > s > 0} \sum_{p=1}^{n_s} \sum_{q=1}^{n_s} \frac{d_j^{(i)}}{d_k^{(i-1)}} \cdot \Phi_{(s,0)(L,s):pq}^{(i):\pi_i^{-1}(j)\pi_{i-1}^{-1}(k)} \cdot [WW]_{pq}^{(s,0)(L,s)}$$

$$\sum_{p=1}^{n_L}\sum_{q=1}^{n_0}\Phi^{(i):j}_{(L,0):\pi_L(p)\pi_0(q)}\cdot\frac{d^{(L)}_{\pi_L(p)}}{d^{(0)}_{\pi_0(q)}}\cdot[W]^{(L,0)}_{pq}$$

$$+\sum_{L>s>0}\sum_{p=1}^{n_s}\sum_{q=1}^{n_s}\Phi^{(i):j}_{(s,0)(L,s):\pi_s(p)\pi_s(q)}\cdot\frac{d^{(s)}_{\pi_s(p)}}{d^{(s)}_{\pi_s(q)}}\cdot[WW]^{(s,0)(L,s)}_{pq}$$

$$=\sum_{p=1}^{n_L}\sum_{q=1}^{n_0}d^{(i)}_j\cdot\Phi^{(i):\pi_i^{-1}(j)}_{(L,0):pq}\cdot[W]^{(L,0)}_{pq}$$

$$+\sum_{L>s>0}\sum_{p=1}^{n_s}\sum_{q=1}^{n_s}d^{(i)}_j\cdot\Phi^{(i):\pi_i^{-1}(j)}_{(s,0)(L,s):pq}\cdot[WW]^{(s,0)(L,s)}_{pq}$$

4. For all $L>s>t>0$, and for all $p$, we have

$$\Phi^{(i):jk}_{(s,t)(t):\pi_s(p)}\cdot d^{(s)}_{\pi_s(p)}=\frac{d^{(i)}_j}{d^{(i-1)}_k}\cdot\Phi^{(i):\pi_i^{-1}(j)\pi_{i-1}^{-1}(k)}_{(s,t)(t):p}$$

$$\Phi^{(i):j}_{(s,t)(t):\pi_s(p)}\cdot d^{(s)}_{\pi_s(p)}=d^{(i)}_j\cdot\Phi^{(i):\pi_i^{-1}(j)}_{(s,t)(t):p}.$$

5. For all $L\geqslant s>0$, $L>t\geqslant0$ with $s\neq t$, we have

$$\Phi^{(i):jk}_{(s)(L,t):\pi_s(p)\pi_t(q)}\cdot\frac{d^{(s)}_{\pi_s(p)}}{d^{(t)}_{\pi_t(q)}}=\frac{d^{(i)}_j}{d^{(i-1)}_k}\cdot\Phi^{(i):\pi_i^{-1}(j)\pi_{i-1}^{-1}(k)}_{(s)(L,t):pq}$$

$$\Phi^{(i):j}_{(s)(L,t):\pi_s(p)\pi_t(q)}\cdot\frac{d^{(s)}_{\pi_s(p)}}{d^{(t)}_{\pi_t(q)}}=d^{(i)}_j\cdot\Phi^{(i):\pi_i^{-1}(j)}_{(s)(L,t):pq}$$

6. For all $L>t>0$, we have

$$\sum_{p=1}^{n_L}\Phi^{(i):jk}_{(L,t)(t):\pi_L(p)}\cdot d^{(L)}_{\pi_L(p)}\cdot[Wb]^{(L,t)(t)}_p$$

$$+\sum_{p=1}^{n_t}\sum_{q=1}^{n_t}\Phi^{(i):jk}_{(t)(L,t):\pi_t(p)\pi_t(q)}\cdot\frac{d^{(t)}_{\pi_t(p)}}{d^{(t)}_{\pi_t(q)}}\cdot[bW]^{(t)(L,t)}_{pq}$$

$$=\sum_{p=1}^{n_L}\frac{d^{(i)}_j}{d^{(i-1)}_k}\cdot\Phi^{(i):\pi_i^{-1}(j)\pi_{i-1}^{-1}(k)}_{(L,t)(t):p}\cdot[Wb]^{(L,t)(t)}_p$$

$$+\sum_{p=1}^{n_t}\sum_{q=1}^{n_t}\frac{d^{(i)}_j}{d^{(i-1)}_k}\cdot\Phi^{(i):\pi_i^{-1}(j)\pi_{i-1}^{-1}(k)}_{(t)(L,t):pq}\cdot[bW]^{(t)(L,t)}_{pq}$$

$$\sum_{p=1}^{n_L}\Phi^{(i):j}_{(L,t)(t):\pi_L(p)}\cdot d^{(L)}_{\pi_L(p)}\cdot[Wb]^{(L,t)(t)}_p$$

$$+\sum_{p=1}^{n_t}\sum_{q=1}^{n_t}\Phi^{(i):j}_{(t)(L,t):\pi_t(p)\pi_t(q)}\cdot\frac{d^{(t)}_{\pi_t(p)}}{d^{(t)}_{\pi_t(q)}}\cdot[bW]^{(t)(L,t)}_{pq}$$

$$=\sum_{p=1}^{n_L}d^{(i)}_j\cdot\Phi^{(i):\pi_i^{-1}(j)}_{(L,t)(t):p}\cdot[Wb]^{(L,t)(t)}_p$$

$$+\sum_{p=1}^{n_t}\sum_{q=1}^{n_t}d^{(i)}_j\cdot\Phi^{(i):\pi_i^{-1}(j)}_{(t)(L,t):pq}\cdot[bW]^{(t)(L,t)}_{pq}$$

7. For all $L \geqslant s > 0$ and for all $p$, we have

$$\Phi^{(i):jk}_{(s):\pi_s(p)} \cdot d^{(s)}_{\pi_s(p)} = \frac{d^{(i)}_j}{d^{(i-1)}_k} \cdot \Phi^{(i):\pi_i^{-1}(j)\pi_{i-1}^{-1}(k)}_{(s):p}$$

$$\Phi^{(i):j}_{(s):\pi_s(p)} \cdot d^{(s)}_{\pi_s(p)} = d^{(i)}_j \cdot \Phi^{(i):\pi_i^{-1}(j)}_{(s):p}.$$

8. We have

$$\Phi^{(i):jk}_1 = \frac{d^{(i)}_j}{d^{(i-1)}_k} \cdot \Phi^{(i):\pi_i^{-1}(j)\pi_{i-1}^{-1}(k)}_1$$

$$\Phi^{(i):j}_1 = d^{(i)}_j \cdot \Phi^{(i):\pi_i^{-1}(j)}_1$$

Also, since the group $G$ satisfy that: $G \cap \mathcal{P}_i$ is trivial (for $i = 0$ or $i = L$) or the whole $\mathcal{P}_i$ (for $0 < i < L$), so we can simplify the above conditions by moving some of the permutation $\pi$'s to the left hand sides. We have

1. For all $L \geqslant s > t \geqslant 0$ with $(s,t) \neq (L,0)$, and for all $p, q$, we have

$$\Phi^{(i):\pi_i(j)\pi_{i-1}(k)}_{(s,t):\pi_s(p)\pi_t(q)} \cdot \frac{d^{(s)}_{\pi_s(p)}}{d^{(t)}_{\pi_t(q)}} = \frac{d^{(i)}_{\pi_i(j)}}{d^{(i-1)}_{\pi_{i-1}(k)}} \cdot \Phi^{(i):jk}_{(s,t):pq}$$

$$\Phi^{(i):\pi_i(j)}_{(s,t):\pi_s(p)\pi_t(q)} \cdot \frac{d^{(s)}_{\pi_s(p)}}{d^{(t)}_{\pi_t(q)}} = d^{(i)}_{\pi_i(j)} \cdot \Phi^{(i):j}_{(s,t):pq}$$

2. For all $L \geqslant s > 0$, $L > t \geqslant 0$ with $s \neq t$, we have

$$\Phi^{(i):\pi_i(j)\pi_{i-1}(k)}_{(s,0)(L,t):\pi_s(p)\pi_t(q)} \cdot \frac{d^{(s)}_{\pi_s(p)}}{d^{(t)}_{\pi_t(q)}} = \frac{d^{(i)}_{\pi_i(j)}}{d^{(i-1)}_{\pi_{i-1}(k)}} \cdot \Phi^{(i):jk}_{(s,0)(L,t):pq}$$

$$\Phi^{(i):\pi_i(j)}_{(s,0)(L,t):\pi_s(p)\pi_t(q)} \cdot \frac{d^{(s)}_{\pi_s(p)}}{d^{(t)}_{\pi_t(q)}} = d^{(i)}_{\pi_i(j)} \cdot \Phi^{(i):j}_{(s,0)(L,t):pq}$$

3. We have

$$\sum_{p=1}^{n_L}\sum_{q=1}^{n_0} \Phi^{(i):\pi_i(j)\pi_{i-1}(k)}_{(L,0):\pi_L(p)\pi_0(q)} \cdot \frac{d^{(L)}_{\pi_L(p)}}{d^{(0)}_{\pi_0(q)}} \cdot [W]^{(L,0)}_{pq}$$

$$+ \sum_{L>s>0}\sum_{p=1}^{n_s}\sum_{q=1}^{n_s} \Phi^{(i):\pi_i(j)\pi_{i-1}(k)}_{(s,0)(L,s):\pi_s(p)\pi_s(q)} \cdot \frac{d^{(s)}_{\pi_s(p)}}{d^{(s)}_{\pi_s(q)}} \cdot [WW]^{(s,0)(L,s)}_{pq}$$

$$= \sum_{p=1}^{n_L}\sum_{q=1}^{n_0} \frac{d^{(i)}_{\pi_i(j)}}{d^{(i-1)}_{\pi_{i-1}(k)}} \cdot \Phi^{(i):jk}_{(L,0):pq} \cdot [W]^{(L,0)}_{pq}$$

$$+ \sum_{L>s>0}\sum_{p=1}^{n_s}\sum_{q=1}^{n_s} \frac{d^{(i)}_{\pi_i(j)}}{d^{(i-1)}_{\pi_{i-1}(k)}} \cdot \Phi^{(i):jk}_{(s,0)(L,s):pq} \cdot [WW]^{(s,0)(L,s)}_{pq}$$

$$\sum_{p=1}^{n_L}\sum_{q=1}^{n_0} \Phi^{(i):\pi_i(j)}_{(L,0):\pi_L(p)\pi_0(q)} \cdot \frac{d^{(L)}_{\pi_L(p)}}{d^{(0)}_{\pi_0(q)}} \cdot [W]^{(L,0)}_{pq}$$

$$+ \sum_{L>s>0}\sum_{p=1}^{n_s}\sum_{q=1}^{n_s} \Phi^{(i):\pi_i(j)}_{(s,0)(L,s):\pi_s(p)\pi_s(q)} \cdot \frac{d^{(s)}_{\pi_s(p)}}{d^{(s)}_{\pi_s(q)}} \cdot [WW]^{(s,0)(L,s)}_{pq}$$

$$= \sum_{p=1}^{n_L} \sum_{q=1}^{n_0} d^{(i)}_{\pi_i(j)} \cdot \Phi^{(i):j}_{(L,0):pq} \cdot [W]^{(L,0)}_{pq}$$

$$+ \sum_{L>s>0} \sum_{p=1}^{n_s} \sum_{q=1}^{n_s} d^{(i)}_{\pi_i(j)} \cdot \Phi^{(i):j}_{(s,0)(L,s):pq} \cdot [WW]^{(s,0)(L,s)}_{pq}$$

4. For all $L > s > t > 0$, and for all $p$, we have

$$\Phi^{(i):\pi_i(j)\pi_{i-1}(k)}_{(s,t)(t):\pi_s(p)} \cdot d^{(s)}_{\pi_s(p)} = \frac{d^{(i)}_{\pi_i(j)}}{d^{(i-1)}_{\pi_{i-1}(k)}} \cdot \Phi^{(i):jk}_{(s,t)(t):p}$$

$$\Phi^{(i):\pi_i(j)}_{(s,t)(t):\pi_s(p)} \cdot d^{(s)}_{\pi_s(p)} = d^{(i)}_{\pi_i(j)} \cdot \Phi^{(i):j}_{(s,t)(t):p}.$$

5. For all $L \geqslant s > 0$, $L > t \geqslant 0$ with $s \neq t$, we have

$$\Phi^{(i):\pi_i(j)\pi_{i-1}(k)}_{(s)(L,t):\pi_s(p)\pi_t(q)} \cdot \frac{d^{(s)}_{\pi_s(p)}}{d^{(t)}_{\pi_t(q)}} = \frac{d^{(i)}_{\pi_i(j)}}{d^{(i-1)}_{\pi_{i-1}(k)}} \cdot \Phi^{(i):jk}_{(s)(L,t):pq}$$

$$\Phi^{(i):\pi_i(j)}_{(s)(L,t):\pi_s(p)\pi_t(q)} \cdot \frac{d^{(s)}_{\pi_s(p)}}{d^{(t)}_{\pi_t(q)}} = d^{(i)}_{\pi_i(j)} \cdot \Phi^{(i):j}_{(s)(L,t):pq}$$

6. For all $L > t > 0$, we have

$$\sum_{p=1}^{n_L} \Phi^{(i):\pi_i(j)\pi_{i-1}(k)}_{(L,t)(t):\pi_L(p)} \cdot d^{(L)}_{\pi_L(p)} \cdot [Wb]^{(L,t)(t)}_p$$

$$+ \sum_{p=1}^{n_t} \sum_{q=1}^{n_t} \Phi^{(i):\pi_i(j)\pi_{i-1}(k)}_{(t)(L,t):\pi_t(p)\pi_t(q)} \cdot \frac{d^{(t)}_{\pi_t(p)}}{d^{(t)}_{\pi_t(q)}} \cdot [bW]^{(t)(L,t)}_{pq}$$

$$= \sum_{p=1}^{n_L} \frac{d^{(i)}_{\pi_i(j)}}{d^{(i-1)}_{\pi_{i-1}(k)}} \cdot \Phi^{(i):jk}_{(L,t)(t):p} \cdot [Wb]^{(L,t)(t)}_p$$

$$+ \sum_{p=1}^{n_t} \sum_{q=1}^{n_t} \frac{d^{(i)}_{\pi_i(j)}}{d^{(i-1)}_{\pi_{i-1}(k)}} \cdot \Phi^{(i):jk}_{(t)(L,t):pq} \cdot [bW]^{(t)(L,t)}_{pq}$$

$$\sum_{p=1}^{n_L} \Phi^{(i):\pi_i(j)}_{(L,t)(t):\pi_L(p)} \cdot d^{(L)}_{\pi_L(p)} \cdot [Wb]^{(L,t)(t)}_p$$

$$+ \sum_{p=1}^{n_t} \sum_{q=1}^{n_t} \Phi^{(i):\pi_i(j)}_{(t)(L,t):\pi_t(p)\pi_t(q)} \cdot \frac{d^{(t)}_{\pi_t(p)}}{d^{(t)}_{\pi_t(q)}} \cdot [bW]^{(t)(L,t)}_{pq}$$

$$= \sum_{p=1}^{n_L} d^{(i)}_{\pi_i(j)} \cdot \Phi^{(i):j}_{(L,t)(t):p} \cdot [Wb]^{(L,t)(t)}_p$$

$$+ \sum_{p=1}^{n_t} \sum_{q=1}^{n_t} d^{(i)}_{\pi_i(j)} \cdot \Phi^{(i):j}_{(t)(L,t):pq} \cdot [bW]^{(t)(L,t)}_{pq}$$

7. For all $L \geqslant s > 0$ and for all $p$, we have

$$\Phi^{(i):\pi_i(j)\pi_{i-1}(k)}_{(s):\pi_s(p)} \cdot d^{(s)}_{\pi_s(p)} = \frac{d^{(i)}_{\pi_i(j)}}{d^{(i-1)}_{\pi_{i-1}(k)}} \cdot \Phi^{(i):jk}_{(s):p}$$

$$\Phi^{(i):\pi_i(j)}_{(s):\pi_s(p)} \cdot d^{(s)}_{\pi_s(p)} = d^{(i)}_{\pi_i(j)} \cdot \Phi^{(i):j}_{(s):p}.$$

8. We have

$$\Phi_1^{(i):\pi_i(j)\pi_{i-1}(k)} = \frac{d_{\pi_i(j)}^{(i)}}{d_{\pi_{i-1}(k)}^{(i-1)}} \cdot \Phi_1^{(i):jk}$$

$$\Phi_1^{(i):\pi_i(j)} = d_{\pi_i(j)}^{(i)} \cdot \Phi_1^{(i):j}$$

From the above equalities, we can determine all constraints for the coefficients according to the entries of $[E(W)]$ as follows:

1. • If $(s,t) = (i, i-1) \notin \{(L, L-1), (1, 0)\}$, $p = j$, $q = k$,

$$\Phi_{(i,i-1):\pi(j)\pi'(k)}^{(i):\pi(j)\pi'(k)} = \Phi_{(i,i-1):jk}^{(i):jk}$$

• If $(s,t) = (i, i-1) = (L, L-1)$, $q = k$,

$$\Phi_{(L,L-1):p\pi_{L-1}(k)}^{(L):j\pi_{L-1}(k)} = \Phi_{(L,L-1):pk}^{(L):jk}$$

• If $(s,t) = (i, i-1) = (1, 0)$, $p = j$,

$$\Phi_{(1,0):\pi_1(j)q}^{(1):\pi_1(j)k} = \Phi_{(1,0):jq}^{(1):jk}$$

2. • If $(s,t) = (i, i-1) \notin \{(L, L-1), (1, 0)\}$, $p = j$, $q = k$,

$$\Phi_{(i,0)(L,i-1):\pi(j)\pi'(k)}^{(i):\pi(j)\pi'(k)} = \Phi_{(i,0)(L,i-1):jk}^{(i):jk}$$

• If $(s,t) = (i, i-1) = (L, L-1)$, $q = k$,

$$\Phi_{(L,0)(L,L-1):p\pi_{L-1}(k)}^{(L):j\pi_{L-1}(k)} = \Phi_{(L,0)(L,L-1):pk}^{(L):jk}$$

• If $(s,t) = (i, i-1) = (1, 0)$, $p = j$,

$$\Phi_{(1,0)(L,0):\pi_1(j)q}^{(1):\pi_1(j)k} = \Phi_{(1,0)(L,0):jq}^{(1):jk}$$

3.

$$\sum_{p=1}^{n_L}\sum_{q=1}^{n_0} \Phi_{(L,0):pq}^{(i):\pi_i(j)\pi_{i-1}(k)} \cdot [W]_{pq}^{(L,0)}$$

$$+ \sum_{L>s>0}\sum_{p=1}^{n_s}\sum_{q=1}^{n_s} \Phi_{(s,0)(L,s):\pi_s(p)\pi_s(q)}^{(i):\pi_i(j)\pi_{i-1}(k)} \cdot \frac{d_{\pi_s(p)}^{(s)}}{d_{\pi_s(q)}^{(s)}} \cdot [WW]_{pq}^{(s,0)(L,s)}$$

$$= \sum_{p=1}^{n_L}\sum_{q=1}^{n_0} \frac{d_{\pi_i(j)}^{(i)}}{d_{\pi_{i-1}(k)}^{(i-1)}} \cdot \Phi_{(L,0):pq}^{(i):jk} \cdot [W]_{pq}^{(L,0)}$$

$$+ \sum_{L>s>0}\sum_{p=1}^{n_s}\sum_{q=1}^{n_s} \frac{d_{\pi_i(j)}^{(i)}}{d_{\pi_{i-1}(k)}^{(i-1)}} \cdot \Phi_{(s,0)(L,s):pq}^{(i):jk} \cdot [WW]_{pq}^{(s,0)(L,s)}$$

For each $L > l > 0$, by scaling all the $d_{-}^{(l)}$'s by the same scalar, we have

$$0 = \sum_{p=1}^{n_L}\sum_{q=1}^{n_0} \Phi_{(L,0):pq}^{(i):\pi_i(j)\pi_{i-1}(k)} \cdot [W]_{pq}^{(L,0)}$$

$$+ \sum_{L>s>0} \sum_{p=1}^{n_s} \sum_{q=1}^{n_s} \Phi^{(i):\pi_i(j)\pi_{i-1}(k)}_{(s,0)(L,s):\pi_s(p)\pi_s(q)} \cdot \frac{d^{(s)}_{\pi_s(p)}}{d^{(s)}_{\pi_s(q)}} \cdot [WW]^{(s,0)(L,s)}_{pq}$$

$$= \sum_{p=1}^{n_L} \sum_{q=1}^{n_0} \Phi^{(i):jk}_{(L,0):pq} \cdot [W]^{(L,0)}_{pq}$$

$$+ \sum_{L>s>0} \sum_{p=1}^{n_s} \sum_{q=1}^{n_s} \Phi^{(i):jk}_{(s,0)(L,s):pq} \cdot [WW]^{(s,0)(L,s)}_{pq}$$

4. For all $L > s > t > 0$, and for all $p$, we have

$$\Phi^{(i):\pi_i(j)\pi_{i-1}(k)}_{(s,t)(t):\pi_s(p)} \cdot d^{(s)}_{\pi_s(p)} = \frac{d^{(i)}_{\pi_i(j)}}{d^{(i-1)}_{\pi_{i-1}(k)}} \cdot \Phi^{(i):jk}_{(s,t)(t):p}.$$

We have

$$\Phi^{(i):jk}_{(s,t)(t):p} = 0.$$

5. For all $L \geqslant s > 0$, $L > t \geqslant 0$ with $s \neq t$, we have

$$\Phi^{(i):\pi_i(j)\pi_{i-1}(k)}_{(s)(L,t):\pi_s(p)\pi_t(q)} \cdot \frac{d^{(s)}_{\pi_s(p)}}{d^{(t)}_{\pi_t(q)}} = \frac{d^{(i)}_{\pi_i(j)}}{d^{(i-1)}_{\pi_{i-1}(k)}} \cdot \Phi^{(i):jk}_{(s)(L,t):pq}.$$

- If $(s,t) = (i, i-1) \notin \{(L, L-1), (1, 0)\}$, $p = j$, $q = k$,

$$\Phi^{(i):\pi(j)\pi'(k)}_{(i)(L,i-1):\pi(j)\pi'(k)} = \Phi^{(i):jk}_{(i)(L,i-1):jk}.$$

- If $i = L$, $(s,t) = (L, L-1)$, $q = k$,

$$\Phi^{(L):j\pi(k)}_{(L)(L,L-1):p\pi(k)} = \Phi^{(L):jk}_{(L)(L,L-1):pk}$$

- If $i = 1$, $(s,t) = (1, 0)$, $p = j$,

$$\Phi^{(1):\pi(j)k}_{(1)(L,0):\pi(j)q} = \Phi^{(1):jk}_{(1)(L,0):jq}.$$

6. For all $L > t > 0$, we have

$$\sum_{p=1}^{n_L} \Phi^{(i):\pi_i(j)\pi_{i-1}(k)}_{(L,t)(t):p} \cdot [Wb]^{(L,t)(t)}_p$$

$$+ \sum_{p=1}^{n_t} \sum_{q=1}^{n_t} \Phi^{(i):\pi_i(j)\pi_{i-1}(k)}_{(t)(L,t):\pi_t(p)\pi_t(q)} \cdot \frac{d^{(t)}_{\pi_t(p)}}{d^{(t)}_{\pi_t(q)}} \cdot [bW]^{(t)(L,t)}_{pq}$$

$$= \sum_{p=1}^{n_L} \frac{d^{(i)}_{\pi_i(j)}}{d^{(i-1)}_{\pi_{i-1}(k)}} \cdot \Phi^{(i):jk}_{(L,t)(t):p} \cdot [Wb]^{(L,t)(t)}_p$$

$$+ \sum_{p=1}^{n_t} \sum_{q=1}^{n_t} \frac{d^{(i)}_{\pi_i(j)}}{d^{(i-1)}_{\pi_{i-1}(k)}} \cdot \Phi^{(i):jk}_{(t)(L,t):pq} \cdot [bW]^{(t)(L,t)}_{pq}$$

For each $L > l > 0$, by scaling all the $d^{(l)}_-$'s by the same scalar, we have

$$0 = \sum_{p=1}^{n_L} \Phi^{(i):\pi_i(j)\pi_{i-1}(k)}_{(L,t)(t):p} \cdot [Wb]^{(L,t)(t)}_p$$

$$+ \sum_{p=1}^{n_t} \sum_{q=1}^{n_t} \Phi_{(t)(L,t):\pi_t(p)\pi_t(q)}^{(i):\pi_i(j)\pi_{i-1}(k)} \cdot \frac{d_{\pi_t(p)}^{(t)}}{d_{\pi_t(q)}^{(t)}} \cdot [bW]_{pq}^{(t)(L,t)}$$

$$= \sum_{p=1}^{n_L} \frac{d_{\pi_i(j)}^{(i)}}{d_{\pi_{i-1}(k)}^{(i-1)}} \cdot \Phi_{(L,t)(t):p}^{(i):jk} \cdot [Wb]_p^{(L,t)(t)}$$

$$+ \sum_{p=1}^{n_t} \sum_{q=1}^{n_t} \frac{d_{\pi_i(j)}^{(i)}}{d_{\pi_{i-1}(k)}^{(i-1)}} \cdot \Phi_{(t)(L,t):pq}^{(i):jk} \cdot [bW]_{pq}^{(t)(L,t)}$$

7. If $i = s = 1, p = j$,

$$\Phi_{(1):\pi(j)}^{(1):\pi(j)k} = \Phi_{(1):j}^{(1):jk}.$$

8. We have

$$\Phi_1^{(i):jk} = 0.$$

Similarly, we can also determine all constraints for the coefficients according to the entries of $[E(b)]$ as follows:

1. For all $L \geqslant s > t \geqslant 0$ with $(s,t) \neq (L,0)$, and for all $p, q$, we have

$$\Phi_{(s,t):\pi_s(p)\pi_t(q)}^{(i):\pi_i(j)} \cdot \frac{d_{\pi_s(p)}^{(s)}}{d_{\pi_t(q)}^{(t)}} = d_{\pi_i(j)}^{(i)} \cdot \Phi_{(s,t):pq}^{(i):j}$$

$(s,t) = (i,0), p = j$,

$$\Phi_{(i,0):\pi(j)q}^{(i):\pi(j)} = \Phi_{(i,0):jq}^{(i):j}$$

2. For all $L \geqslant s > 0, L > t \geqslant 0$ with $s \neq t$, we have

$$\Phi_{(s,0)(L,t):\pi_s(p)\pi_t(q)}^{(i):\pi_i(j)} \cdot \frac{d_{\pi_s(p)}^{(s)}}{d_{\pi_t(q)}^{(t)}} = d_{\pi_i(j)}^{(i)} \cdot \Phi_{(s,0)(L,t):pq}^{(i):j}$$

- $t = 0, s = i < L, p = j$,

$$\Phi_{(i,0)(L,0):\pi(j)q}^{(i):\pi(j)} \cdot = \Phi_{(i,0)(L,0):jq}^{(i):j}$$

- $t = 0, s = i = L$,

$$\Phi_{(L,0)(L,0):pq}^{(L):j} = \Phi_{(L,0)(L,0):pq}^{(L):j}$$

3. We have

$$\sum_{p=1}^{n_L} \sum_{q=1}^{n_0} \Phi_{(L,0):pq}^{(i):\pi_i(j)} \cdot [W]_{pq}^{(L,0)}$$

$$+ \sum_{L>s>0} \sum_{p=1}^{n_s} \sum_{q=1}^{n_s} \Phi_{(s,0)(L,s):\pi_s(p)\pi_s(q)}^{(i):\pi_i(j)} \cdot \frac{d_{\pi_s(p)}^{(s)}}{d_{\pi_s(q)}^{(s)}} \cdot [WW]_{pq}^{(s,0)(L,s)}$$

$$= \sum_{p=1}^{n_L} \sum_{q=1}^{n_0} d_{\pi_i(j)}^{(i)} \cdot \Phi_{(L,0):pq}^{(i):j} \cdot [W]_{pq}^{(L,0)}$$

$$+ \sum_{L>s>0} \sum_{p=1}^{n_s} \sum_{q=1}^{n_s} d_{\pi_i(j)}^{(i)} \cdot \Phi_{(s,0)(L,s):pq}^{(i):j} \cdot [WW]_{pq}^{(s,0)(L,s)}$$

- If $L > i > 0$. For each $L > l > 0$, by scaling all the $d_-^{(l)}$'s by the same scalar, we have

$$
0 = \sum_{p=1}^{n_L} \sum_{q=1}^{n_0} \Phi_{(L,0):pq}^{(i):\pi_i(j)} \cdot [W]_{pq}^{(L,0)}
$$
$$
+ \sum_{L>s>0} \sum_{p=1}^{n_s} \sum_{q=1}^{n_s} \Phi_{(s,0)(L,s):\pi_s(p)\pi_s(q)}^{(i):\pi_i(j)} \cdot \frac{d_{\pi_s(p)}^{(s)}}{d_{\pi_s(q)}^{(s)}} \cdot [WW]_{pq}^{(s,0)(L,s)}
$$
$$
= \sum_{p=1}^{n_L} \sum_{q=1}^{n_0} d_{\pi_i(j)}^{(i)} \cdot \Phi_{(L,0):pq}^{(i):j} \cdot [W]_{pq}^{(L,0)}
$$
$$
+ \sum_{L>s>0} \sum_{p=1}^{n_s} \sum_{q=1}^{n_s} d_{\pi_i(j)}^{(i)} \cdot \Phi_{(s,0)(L,s):pq}^{(i):j} \cdot [WW]_{pq}^{(s,0)(L,s)}
$$

- If $i = L$. We have

$$
\sum_{p=1}^{n_L} \sum_{q=1}^{n_0} \Phi_{(L,0):pq}^{(L):j} \cdot [W]_{pq}^{(L,0)}
$$
$$
+ \sum_{L>s>0} \sum_{p=1}^{n_s} \sum_{q=1}^{n_s} \Phi_{(s,0)(L,s):\pi_s(p)\pi_s(q)}^{(L):j} \cdot \frac{d_{\pi_s(p)}^{(s)}}{d_{\pi_s(q)}^{(s)}} \cdot [WW]_{pq}^{(s,0)(L,s)}
$$
$$
= \sum_{p=1}^{n_L} \sum_{q=1}^{n_0} \Phi_{(L,0):pq}^{(L):j} \cdot [W]_{pq}^{(L,0)}
$$
$$
+ \sum_{L>s>0} \sum_{p=1}^{n_s} \sum_{q=1}^{n_s} \Phi_{(s,0)(L,s):pq}^{(L):j} \cdot [WW]_{pq}^{(s,0)(L,s)}
$$

which means $\Phi_{(L,0):pq}^{(L):j}$ can be arbitrary. The rest is

$$
\sum_{L>s>0} \sum_{p=1}^{n_s} \sum_{q=1}^{n_s} \Phi_{(s,0)(L,s):\pi_s(p)\pi_s(q)}^{(L):j} \cdot \frac{d_{\pi_s(p)}^{(s)}}{d_{\pi_s(q)}^{(s)}} \cdot [WW]_{pq}^{(s,0)(L,s)}
$$
$$
= \sum_{L>s>0} \sum_{p=1}^{n_s} \sum_{q=1}^{n_s} \Phi_{(s,0)(L,s):pq}^{(L):j} \cdot [WW]_{pq}^{(s,0)(L,s)}
$$

For an $L > r > 0$, by letting $\pi_r$ be the identity, and $d_p^{(r)}$ be 1 for all $p$, we have

$$
\sum_{L>s>0,s\neq r} \sum_{p=1}^{n_s} \sum_{q=1}^{n_s} \Phi_{(s,0)(L,s):\pi_s(p)\pi_s(q)}^{(L):j} \cdot \frac{d_{\pi_s(p)}^{(s)}}{d_{\pi_s(q)}^{(s)}} \cdot [WW]_{pq}^{(s,0)(L,s)}
$$
$$
= \sum_{L>s>0,s\neq r} \sum_{p=1}^{n_s} \sum_{q=1}^{n_s} \Phi_{(s,0)(L,s):pq}^{(L):j} \cdot [WW]_{pq}^{(s,0)(L,s)},
$$

so

$$
\sum_{p=1}^{n_r} \sum_{q=1}^{n_r} \Phi_{(r,0)(L,r):\pi_r(p)\pi_r(q)}^{(L):j} \cdot \frac{d_{\pi_r(p)}^{(r)}}{d_{\pi_r(q)}^{(r)}} \cdot [WW]_{pq}^{(r,0)(L,r)}
$$
$$
= \sum_{p=1}^{n_r} \sum_{q=1}^{n_r} \Phi_{(r,0)(L,r):pq}^{(L):j} \cdot [WW]_{pq}^{(r,0)(L,r)}
$$

By Lemma B.3 and Corollary A.7, we have

$$\Phi^{(L):j}_{(r,0)(L,r):\pi_r(p)\pi_r(q)} \cdot \frac{d^{(r)}_{\pi_r(p)}}{d^{(r)}_{\pi_r(q)}} = \Phi^{(L):j}_{(r,0)(L,r):pq}.$$

So

$$\Phi^{(L):j}_{(r,0)(L,r):pq} = 0$$

for $p \neq q$, and

$$\Phi^{(L):j}_{(r,0)(L,r):\pi_r(p)\pi_r(p)} = \Phi^{(L):j}_{(r,0)(L,r):pp}.$$

In conclusion, we have $\Phi^{(L):j}_{(L,0):pq}$ is arbitrary, and for $L > s > 0$,

$$\Phi^{(L):j}_{(s,0)(L,s):\pi(p)\pi(p)} = \Phi^{(L):j}_{(s,0)(L,s):pp}.$$

4. For all $L > s > t > 0$, and for all $p$, we have

$$\Phi^{(i):\pi_i(j)}_{(s,t)(t):\pi_s(p)} \cdot d^{(s)}_{\pi_s(p)} = d^{(i)}_{\pi_i(j)} \cdot \Phi^{(i):j}_{(s,t)(t):p}.$$

If $i = s, p = j$,

$$\Phi^{(i):\pi(j)}_{(i,t)(t):\pi(j)} = \Phi^{(i):j}_{(i,t)(t):j}.$$

5. For all $L \geqslant s > 0$, $L > t \geqslant 0$ with $s \neq t$, we have

$$\Phi^{(i):\pi_i(j)}_{(s)(L,t):\pi_s(p)\pi_t(q)} \cdot \frac{d^{(s)}_{\pi_s(p)}}{d^{(t)}_{\pi_t(q)}} = d^{(i)}_{\pi_i(j)} \cdot \Phi^{(i):j}_{(s)(L,t):pq}$$

- $s = i < L, t = 0, p = j$,

$$\Phi^{(i):\pi(j)}_{(i)(L,0):\pi(j)q} = \Phi^{(i):j}_{(i)(L,0):jq}$$

- $s = i = L, t = 0$

$$\Phi^{(L):j}_{(L)(L,0):pq} = \Phi^{(L):j}_{(L)(L,0):pq}$$

6. For all $L > t > 0$, we have

$$\sum_{p=1}^{n_L} \Phi^{(i):\pi_i(j)}_{(L,t)(t):p} \cdot [Wb]^{(L,t)(t)}_p$$

$$+ \sum_{p=1}^{n_t} \sum_{q=1}^{n_t} \Phi^{(i):\pi_i(j)}_{(t)(L,t):\pi_t(p)\pi_t(q)} \cdot \frac{d^{(t)}_{\pi_t(p)}}{d^{(t)}_{\pi_t(q)}} \cdot [bW]^{(t)(L,t)}_{pq}$$

$$= \sum_{p=1}^{n_L} d^{(i)}_{\pi_i(j)} \cdot \Phi^{(i):j}_{(L,t)(t):p} \cdot [Wb]^{(L,t)(t)}_p$$

$$+ \sum_{p=1}^{n_t} \sum_{q=1}^{n_t} d^{(i)}_{\pi_i(j)} \cdot \Phi^{(i):j}_{(t)(L,t):pq} \cdot [bW]^{(t)(L,t)}_{pq}$$

- If $L > i > 0$. For each $L > l > 0$, by scaling all the $d^{(l)}_-$'s by the same scalar, we have

$$0 = \sum_{p=1}^{n_L} \Phi_{(L,t)(t):\pi_L(p)}^{(i):\pi_i(j)} \cdot d_{\pi_L(p)}^{(L)} \cdot [Wb]_p^{(L,t)(t)}$$

$$+ \sum_{p=1}^{n_t} \sum_{q=1}^{n_t} \Phi_{(t)(L,t):\pi_t(p)\pi_t(q)}^{(i):\pi_i(j)} \cdot \frac{d_{\pi_t(p)}^{(t)}}{d_{\pi_t(q)}^{(t)}} \cdot [bW]_{pq}^{(t)(L,t)}$$

$$= \sum_{p=1}^{n_L} d_{\pi_i(j)}^{(i)} \cdot \Phi_{(L,t)(t):p}^{(i):j} \cdot [Wb]_p^{(L,t)(t)}$$

$$+ \sum_{p=1}^{n_t} \sum_{q=1}^{n_t} d_{\pi_i(j)}^{(i)} \cdot \Phi_{(t)(L,t):pq}^{(i):j} \cdot [bW]_{pq}^{(t)(L,t)}$$

- If $i = L$. We have

$$\sum_{p=1}^{n_L} \Phi_{(L,t)(t):p}^{(L):j} \cdot [Wb]_p^{(L,t)(t)}$$

$$+ \sum_{p=1}^{n_t} \sum_{q=1}^{n_t} \Phi_{(t)(L,t):\pi_t(p)\pi_t(q)}^{(L):j} \cdot \frac{d_{\pi_t(p)}^{(t)}}{d_{\pi_t(q)}^{(t)}} \cdot [bW]_{pq}^{(t)(L,t)}$$

$$= \sum_{p=1}^{n_L} \Phi_{(L,t)(t):p}^{(L):j} \cdot [Wb]_p^{(L,t)(t)}$$

$$+ \sum_{p=1}^{n_t} \sum_{q=1}^{n_t} \cdot \Phi_{(t)(L,t):pq}^{(L):j} \cdot [bW]_{pq}^{(t)(L,t)}$$

which means $\Phi_{(L,t)(t):p}^{(L):j}$ can be arbitrary. The rest is

$$\sum_{p=1}^{n_t} \sum_{q=1}^{n_t} \Phi_{(t)(L,t):\pi_t(p)\pi_t(q)}^{(L):j} \cdot \frac{d_{\pi_t(p)}^{(t)}}{d_{\pi_t(q)}^{(t)}} \cdot [bW]_{pq}^{(t)(L,t)}$$

$$= \sum_{p=1}^{n_t} \sum_{q=1}^{n_t} \Phi_{(t)(L,t):pq}^{(L):j} \cdot [bW]_{pq}^{(t)(L,t)}$$

By Lemma B.5 and Corollary A.7, we have

$$\Phi_{(t)(L,t):\pi_t(p)\pi_t(q)}^{(L):j} \cdot \frac{d_{\pi_t(p)}^{(t)}}{d_{\pi_t(q)}^{(t)}} = \Phi_{(t)(L,t):pq}^{(L):j}$$

So

$$\Phi_{(t)(L,t):pq}^{(L):j} = 0$$

for $p \neq q$, and

$$\Phi_{(t)(L,t):\pi_t(p)\pi_t(p)}^{(L):j} = \Phi_{(t)(L,t):pp}^{(L):j}$$

In conclusion, for all $L > t > 0$, we have $\Phi_{(L,t)(t):p}^{(L):j}$ is arbitrary, and

$$\Phi_{(t)(L,t):\pi(p)\pi(p)}^{(L):j} = \Phi_{(t)(L,t):pp}^{(L):j}.$$

7. For all $L \geqslant s > 0$ and for all $p$, we have

$$\Phi_{(s):\pi_s(p)}^{(i):\pi_i(j)} \cdot d_{\pi_s(p)}^{(s)} = d_{\pi_i(j)}^{(i)} \cdot \Phi_{(s):p}^{(i):j}.$$

- If $i = s < L$, $p = j$,

$$\Phi^{(i):\pi(j)}_{(i):\pi(j)} = \Phi^{(i):j}_{(i):j}.$$

- If $i = s = L$,

$$\Phi^{(L):j}_{(L):p} = \Phi^{(L):j}_{(L):p}.$$

8. We have

$$\Phi^{(i):\pi_i(j)}_1 = d^{(i)}_{\pi_i(j)} \cdot \Phi^{(i):j}_1$$

- If $i = L$, we have

$$\Phi^{(L):j}_1 = \Phi^{(L):j}_1.$$

## C.5. $G$-Equivariant polynomial layers

Based on the above discussions, we conclude that every $G$-equivariant polynomial layer , which is defined as a linear combination of stable polynomial terms, is given as $E(U) = ([E(W)], [E(b)])$, where the entries of $[E(W)]$ and $[E(b)]$ are given case by case as folows:

- For $i = L$, we have

$$[E(W)]^{(L)}_{jk} = \sum_{p=1}^{n_L} \Phi^{(L):j\bullet}_{(L,L-1):p\bullet} \cdot [W]^{(L,L-1)}_{pk} + \sum_{p=1}^{n_L} \Phi^{(L):j\bullet}_{(L,0)(L,L-1):p\bullet} \cdot [WW]^{(L,0)(L,L-1)}_{pk}$$

$$+ \sum_{p=1}^{n_L} \Phi^{(L):j\bullet}_{(L)(L,L-1):p\bullet} \cdot [bW]^{(L)(L,L-1)}_{pk}$$

$$[E(b)]^{(L)}_j = \sum_{p=1}^{n_L} \sum_{q=1}^{n_0} \Phi^{(L):j}_{(L,0)(L,0):pq} \cdot [WW]^{(L,0)(L,0)}_{pq} + \sum_{p=1}^{n_L} \sum_{q=1}^{n_0} \Phi^{(L):j}_{(L,0):pq} \cdot [W]^{(L,0)}_{pq}$$

$$+ \sum_{L>s>0} \sum_{p=1}^{n_s} \Phi^{(L):j}_{(s,0)(L,s):\bullet\bullet} \cdot [WW]^{(s,0)(L,s)}_{pp} + \sum_{p=1}^{n_L} \sum_{q=1}^{n_0} \Phi^{(L):j}_{(L)(L,0):pq} \cdot [bW]^{(L)(L,0)}_{pq}$$

$$+ \sum_{L>t>0} \sum_{p=1}^{n_L} \Phi^{(L):j}_{(L,t)(t):p} \cdot [Wb]^{(L,t)(t)}_p + \sum_{L>t>0} \sum_{p=1}^{n_t} \Phi^{(L):j}_{(t)(L,t):\bullet\bullet} \cdot [bW]^{(t)(L,t)}_{pp}$$

$$+ \sum_{p=1}^{n_L} \Phi^{(L):j}_{(L):p} \cdot [b]^{(L)}_p + \Phi^{(L):j}_1$$

$$[E(b)]^{(L)}_j = \sum_{p=1}^{n_L} \sum_{q=1}^{n_0} \Phi^{(L):j}_{(L,0)(L,0):pq} \cdot [WW]^{(L,0)(L,0)}_{pq} + \sum_{p=1}^{n_L} \sum_{q=1}^{n_0} \Phi^{(L):j}_{(L,0):pq} \cdot [W]^{(L,0)}_{pq}$$

$$+ \sum_{L>s>0} \Phi^{(L):j}_{(s,0)(L,s):\bullet\bullet} \cdot \sum_{p=1}^{n_s} [WW]^{(s,0)(L,s)}_{pp} + \sum_{p=1}^{n_L} \sum_{q=1}^{n_0} \Phi^{(L):j}_{(L)(L,0):pq} \cdot [bW]^{(L)(L,0)}_{pq}$$

$$+ \sum_{L>t>0} \sum_{p=1}^{n_L} \Phi^{(L):j}_{(L,t)(t):p} \cdot [Wb]^{(L,t)(t)}_p + \sum_{L>t>0} \Phi^{(L):j}_{(t)(L,t):\bullet\bullet} \cdot \sum_{p=1}^{n_t} [bW]^{(t)(L,t)}_{pp}$$

$$+ \sum_{p=1}^{n_L} \Phi^{(L):j}_{(L):p} \cdot [b]^{(L)}_p + \Phi^{(L):j}_1$$

- For $i = 1$, we have

$$[E(W)]^{(1)}_{jk} = \sum_{q=1}^{n_0} \Phi^{(1):\bullet k}_{(1,0):\bullet q} \cdot [W]^{(1,0)}_{jq} + \sum_{q=1}^{n_0} \Phi^{(1):\bullet k}_{(1,0)(L,0):\bullet q} \cdot [WW]^{(1,0)(L,0)}_{jq}$$

$$+ \sum_{q=1}^{n_0} \Phi_{(1)(L,0):\bullet q}^{(1):\bullet k} \cdot [bW]_{jq}^{(1)(L,0)} + \Phi_{(1):\bullet}^{(1):\bullet k} \cdot [b]_j^{(1)}$$

$$[E(b)]_j^{(1)} = \sum_{q=1}^{n_0} \Phi_{(1,0):\bullet q}^{(1):\bullet} \cdot [W]_{jq}^{(1,0)} + \sum_{q=1}^{n_0} \Phi_{(1,0)(L,0):\bullet q}^{(1):\bullet} \cdot [WW]_{jq}^{(1,0)(L,0)}$$

$$+ \sum_{q=1}^{n_0} \Phi_{(1)(L,0):\bullet q}^{(1):\bullet} \cdot [bW]_{jq}^{(1)(L,0)} + \Phi_{(1):\bullet}^{(1):\bullet} \cdot [b]_j^{(1)}$$

- For $L > i > 1$, we have

$$[E(W)]_{jk}^{(i)} = \Phi_{(i,i-1):\bullet\bullet}^{(i):\bullet\bullet} \cdot [W]_{jk}^{(i,i-1)} + \Phi_{(i,0)(L,i-1):\bullet\bullet}^{(i):\bullet\bullet} \cdot [WW]_{jk}^{(i,0)(L,i-1)}$$

$$+ \Phi_{(i)(L,i-1):\bullet\bullet}^{(i):\bullet\bullet} \cdot [bW]_{jk}^{(i)(L,i-1)}$$

$$[E(b)]_j^{(i)} = \sum_{q=1}^{n_0} \Phi_{(i,0):\bullet q}^{(i):\bullet} \cdot [W]_{jq}^{(i,0)} + \sum_{q=1}^{n_0} \Phi_{(i,0)(L,0):\bullet q}^{(i):\bullet} \cdot [WW]_{jq}^{(t,0)(L,0)}$$

$$+ \sum_{i>t>0} \sum_{p=1}^{n_i} \Phi_{(i,t)(t):\bullet}^{(i):\bullet} \cdot [Wb]_j^{(i,t)(t)} + \sum_{q=1}^{n_t} \Phi_{(i)(L,0):\bullet q}^{(i):\bullet} \cdot [bW]_{jq}^{(i)(L,0)}$$

$$+ \Phi_{(i):\bullet}^{(i):\bullet} \cdot [b]_j^{(i)}$$

In the above formulas, the bullet $\bullet$ indicates that the corresponding coefficient is independent of the indices at the bullet.

## D. Invariant Polynomial Layers

In this section, we construct a polynomial map $I: \mathcal{U} \to \mathbb{R}^{d'}$ that is $G$-invariant, i.e., $I(gU) = I(U)$ for all $g \in \mathcal{G}_{\mathcal{U}}$ and $U \in \mathcal{U}$.

### D.1. Invariant layer as a linear combination of stable polynomial terms

Similar to the equivariant maps, we seek the invariant map among polynomial maps that is a linear combination of stable polynomial terms, specifically the entries of $[W]^{(s,t)}$, $[b]^{(s)}$, $[Wb]^{(s,t)(t)}$, $[bW]^{(s)(L,t)}$, and $[WW]^{(s,0)(L,t)}$, along with a bias. In concrete:

$$I(U) := \sum_{L \geqslant s > t \geqslant 0} \sum_{p=1}^{n_s} \sum_{q=1}^{n_t} \Phi_{(s,t):pq} \cdot [W]_{pq}^{(s,t)} + \sum_{L \geqslant s > 0} \sum_{p=1}^{n_s} \Phi_{(s):p} \cdot [b]_p^{(s)}$$

$$+ \sum_{L \geqslant s > t > 0} \sum_{p=1}^{n_s} \Phi_{(s,t)(t):p} \cdot [Wb]_p^{(s,t)(t)}$$

$$+ \sum_{L \geqslant s > 0} \sum_{L > t \geqslant 0} \sum_{p=1}^{n_s} \sum_{q=1}^{n_t} \Phi_{(s)(L,t):pq} \cdot [bW]_{pq}^{(s)(L,t)}$$

$$+ \sum_{L \geqslant s > 0} \sum_{L > t \geqslant 0} \sum_{p=1}^{n_s} \sum_{q=1}^{n_t} \Phi_{(s,0)(L,t):pq} \cdot [WW]_{pq}^{(s,0)(L,t)} + \Phi_1.$$

All $\Phi_-$'s are in $\mathbb{R}^{d' \times d}$, except the bias $\Phi_1$ is in $\mathbb{R}^{d' \times 1}$. In summary, $I$ is parameterized by $\Phi_-$'s and $\Psi_-$'s. We need $I$ to be $G$-invariant, which means $I(gU) = I(U)$.

### D.2. Compute $I(gU)$

From the definition of $I(U)$, we have:

$$
I(gU) = \sum_{L \geqslant s > t \geqslant 0} \sum_{p=1}^{n_s} \sum_{q=1}^{n_t} \Phi_{(s,t):pq} \cdot [gW]_{pq}^{(s,t)}
$$

$$
+ \sum_{L \geqslant s > 0} \sum_{p=1}^{n_s} \Phi_{(s):p} \cdot [gb]_p^{(s)}
$$

$$
+ \sum_{L \geqslant s > t > 0} \sum_{p=1}^{n_s} \Phi_{(s,t)(t):p} \cdot [gWgb]_p^{(s,t)(t)}
$$

$$
+ \sum_{L \geqslant s > 0} \sum_{L > t \geqslant 0} \sum_{p=1}^{n_s} \sum_{q=1}^{n_t} \Phi_{(s)(L,t):pq} \cdot [gbgW]_{pq}^{(s)(L,t)}
$$

$$
+ \sum_{L \geqslant s > 0} \sum_{L > t \geqslant 0} \sum_{p=1}^{n_s} \sum_{q=1}^{n_t} \Phi_{(s,0)(L,t):pq} \cdot [gWgW]_{pq}^{(s,0)(L,t)}
$$

$$
+ \Phi_1
$$

$$
= \sum_{L \geqslant s > t \geqslant 0} \sum_{p=1}^{n_s} \sum_{q=1}^{n_t} \Phi_{(s,t):pq} \cdot \frac{d_p^{(s)}}{d_q^{(t)}} \cdot [W]_{\pi_s^{-1}(p), \pi_t^{-1}(q)}^{(s,t)}
$$

$$
+ \sum_{L \geqslant s > 0} \sum_{p=1}^{n_s} \Phi_{(s):p} \cdot d_p^{(s)} \cdot [b]_{\pi_s^{-1}(p)}^{(s)}
$$

$$
+ \sum_{L \geqslant s > t > 0} \sum_{p=1}^{n_s} \Phi_{(s,t)(t):p} \cdot d_p^{(s)} \cdot [Wb]_{\pi_s^{-1}(p)}^{(s,t)(t)}
$$

$$
+ \sum_{L \geqslant s > 0} \sum_{L > t \geqslant 0} \sum_{p=1}^{n_s} \sum_{q=1}^{n_t} \Phi_{(s)(L,t):pq} \cdot \frac{d_p^{(s)}}{d_q^{(t)}} \cdot [bW]_{\pi_s^{-1}(p), \pi_t^{-1}(q)}^{(s)(L,t)}
$$

$$
+ \sum_{L \geqslant s > 0} \sum_{L > t \geqslant 0} \sum_{p=1}^{n_s} \sum_{q=1}^{n_t} \Phi_{(s,0)(L,t):pq} \cdot \frac{d_p^{(s)}}{d_q^{(t)}} \cdot [WW]_{\pi_s^{-1}(p), \pi_t^{-1}(q)}^{(s,0)(L,t)}
$$

$$
+ \Phi_1
$$

$$
= \sum_{L \geqslant s > t \geqslant 0} \sum_{p=1}^{n_s} \sum_{q=1}^{n_t} \Phi_{(s,t):\pi_s(p)\pi_t(q)} \cdot \frac{d_{\pi_s(p)}^{(s)}}{d_{\pi_t(q)}^{(t)}} \cdot [W]_{pq}^{(s,t)}
$$

$$
+ \sum_{L \geqslant s > 0} \sum_{p=1}^{n_s} \Phi_{(s):\pi_s(p)} \cdot d_{\pi_s(p)}^{(s)} \cdot [b]_p^{(s)}
$$

$$
+ \sum_{L \geqslant s > t > 0} \sum_{p=1}^{n_s} \Phi_{(s,t)(t):\pi_s(p)} \cdot d_{\pi_s(p)}^{(s)} \cdot [Wb]_p^{(s,t)(t)}
$$

$$
+ \sum_{L \geqslant s > 0} \sum_{L > t \geqslant 0} \sum_{p=1}^{n_s} \sum_{q=1}^{n_t} \Phi_{(s)(L,t):\pi_s(p)\pi_t(q)} \cdot \frac{d_{\pi_s(p)}^{(s)}}{d_{\pi_t(q)}^{(t)}} \cdot [bW]_{pq}^{(s)(L,t)}
$$

$$
+ \sum_{L \geqslant s > 0} \sum_{L > t \geqslant 0} \sum_{p=1}^{n_s} \sum_{q=1}^{n_t} \Phi_{(s,0)(L,t):\pi_s(p)\pi_t(q)} \cdot \frac{d_{\pi_s(p)}^{(s)}}{d_{\pi_t(q)}^{(t)}} \cdot [WW]_{pq}^{(s,0)(L,t)}
$$

$$
+ \Phi_1.
$$

**D.3. Compare $I(gU)$ and $I(U)$**

Since $I(gU) = I(U)$, from Corollary B.7, the parameters $\Phi_-$'s have to satisfy these following conditions:

1. For all $L \geqslant s > t \geqslant 0$ with $(s, t) \neq (L, 0)$, and for all $p, q$, we have

$$\Phi_{(s,t):\pi_s(p)\pi_t(q)} \cdot \frac{d^{(s)}_{\pi_s(p)}}{d^{(t)}_{\pi_t(q)}} = \Phi_{(s,t):pq}$$

2. For all $L \geqslant s > 0$, $L > t \geqslant 0$ with $s \neq t$, we have

$$\Phi_{(s,0)(L,t):\pi_s(p)\pi_t(q)} \cdot \frac{d^{(s)}_{\pi_s(p)}}{d^{(t)}_{\pi_t(q)}} = \Phi_{(s,0)(L,t):pq}$$

3. We have

$$
\sum_{p=1}^{n_L} \sum_{q=1}^{n_0} \Phi_{(L,0):\pi_L(p)\pi_0(q)} \cdot \frac{d^{(L)}_{\pi_L(p)}}{d^{(0)}_{\pi_0(q)}} \cdot [W]^{(L,0)}_{pq}
$$
$$
+ \sum_{L>s>0} \sum_{p=1}^{n_s} \sum_{q=1}^{n_s} \Phi_{(s,0)(L,s):\pi_s(p)\pi_s(q)} \cdot \frac{d^{(s)}_{\pi_s(p)}}{d^{(s)}_{\pi_s(q)}} \cdot [WW]^{(s,0)(L,s)}_{pq}
$$
$$
= \sum_{p=1}^{n_L} \sum_{q=1}^{n_0} \Phi_{(L,0):pq} \cdot [W]^{(L,0)}_{pq}
$$
$$
+ \sum_{L>s>0} \sum_{p=1}^{n_s} \sum_{q=1}^{n_s} \Phi_{(s,0)(L,s):pq} \cdot [WW]^{(s,0)(L,s)}_{pq}
$$

4. For all $L > s > t > 0$, and for all $p$, we have

$$\Phi_{(s,t)(t):\pi_s(p)} \cdot d^{(s)}_{\pi_s(p)} = \Phi_{(s,t)(t):p}.$$

5. For all $L \geqslant s > 0$, $L > t \geqslant 0$ with $s \neq t$, we have

$$\Phi_{(s)(L,t):\pi_s(p)\pi_t(q)} \cdot \frac{d^{(s)}_{\pi_s(p)}}{d^{(t)}_{\pi_t(q)}} = \Phi_{(s)(L,t):pq}$$

6. For all $L > t > 0$, we have

$$
\sum_{p=1}^{n_L} \Phi_{(L,t)(t):\pi_L(p)} \cdot d^{(L)}_{\pi_L(p)} \cdot [Wb]^{(L,t)(t)}_p
$$
$$
+ \sum_{p=1}^{n_t} \sum_{q=1}^{n_t} \Phi_{(t)(L,t):\pi_t(p)\pi_t(q)} \cdot \frac{d^{(t)}_{\pi_t(p)}}{d^{(t)}_{\pi_t(q)}} \cdot [bW]^{(t)(L,t)}_{pq}
$$
$$
= \sum_{p=1}^{n_L} \Phi_{(L,t)(t):p} \cdot [Wb]^{(L,t)(t)}_p
$$
$$
+ \sum_{p=1}^{n_t} \sum_{q=1}^{n_t} \Phi_{(t)(L,t):pq} \cdot [bW]^{(t)(L,t)}_{pq}
$$

7. For all $L \geqslant s > 0$ and for all $p$, we have

$$\Phi_{(s):\pi_s(p)} \cdot d^{(s)}_{\pi_s(p)} = \Phi_{(s):p}.$$

8. We have

$$\Phi_1 = \Phi_1.$$

Solve these equations, we have

1. For all $L \geqslant s > t \geqslant 0$ with $(s, t) \neq (L, 0)$, and for all $p, q$, we have

$$\Phi_{(s,t):pq} = 0$$

2. For all $L \geqslant s > 0, L > t \geqslant 0$ with $s \neq t$, we have

$$\Phi_{(s,0)(L,t):\pi_s(p)\pi_t(q)} \cdot \frac{d^{(s)}_{\pi_s(p)}}{d^{(t)}_{\pi_t(q)}} = \Phi_{(s,0)(L,t):pq}$$

   If $(s, t) = (L, 0$, we have

$$\Phi_{(L,0)(L,0):pq} = \Phi_{(L,0)(L,0):pq}.$$

3. We have

$$\sum_{p=1}^{n_L} \sum_{q=1}^{n_0} \Phi_{(L,0):pq} \cdot [W]^{(L,0)}_{pq}$$

$$+ \sum_{L>s>0} \sum_{p=1}^{n_s} \sum_{q=1}^{n_s} \Phi_{(s,0)(L,s):\pi_s(p)\pi_s(q)} \cdot \frac{d^{(s)}_{\pi_s(p)}}{d^{(s)}_{\pi_s(q)}} \cdot [WW]^{(s,0)(L,s)}_{pq}$$

$$= \sum_{p=1}^{n_L} \sum_{q=1}^{n_0} \Phi_{(L,0):pq} \cdot [W]^{(L,0)}_{pq}$$

$$+ \sum_{L>s>0} \sum_{p=1}^{n_s} \sum_{q=1}^{n_s} \Phi_{(s,0)(L,s):pq} \cdot [WW]^{(s,0)(L,s)}_{pq}$$

which means $\Phi_{(L,0):pq}$ can be arbitrary. The rest is

$$\sum_{L>s>0} \sum_{p=1}^{n_s} \sum_{q=1}^{n_s} \Phi_{(s,0)(L,s):\pi_s(p)\pi_s(q)} \cdot \frac{d^{(s)}_{\pi_s(p)}}{d^{(s)}_{\pi_s(q)}} \cdot [WW]^{(s,0)(L,s)}_{pq}$$

$$= \sum_{L>s>0} \sum_{p=1}^{n_s} \sum_{q=1}^{n_s} \Phi_{(s,0)(L,s):pq} \cdot [WW]^{(s,0)(L,s)}_{pq}$$

For an $L > r > 0$, by letting $\pi_r$ be the identity, and $d^{(r)}_p$ be 1 for all $p$, we have

$$\sum_{L>s>0, s \neq r} \sum_{p=1}^{n_s} \sum_{q=1}^{n_s} \Phi_{(s,0)(L,s):\pi_s(p)\pi_s(q)} \cdot \frac{d^{(s)}_{\pi_s(p)}}{d^{(s)}_{\pi_s(q)}} \cdot [WW]^{(s,0)(L,s)}_{pq}$$

$$= \sum_{L>s>0, s \neq r} \sum_{p=1}^{n_s} \sum_{q=1}^{n_s} \Phi_{(s,0)(L,s):pq} \cdot [WW]^{(s,0)(L,s)}_{pq},$$

so

$$\sum_{p=1}^{n_r} \sum_{q=1}^{n_r} \Phi_{(r,0)(L,r):\pi_r(p)\pi_r(q)} \cdot \frac{d^{(r)}_{\pi_r(p)}}{d^{(r)}_{\pi_r(q)}} \cdot [WW]^{(r,0)(L,r)}_{pq}$$

$$= \sum_{p=1}^{n_r} \sum_{q=1}^{n_r} \Phi_{(r,0)(L,r):pq} \cdot [WW]_{pq}^{(r,0)(L,r)}$$

By Lemma B.3 and Corollary A.7, we have

$$\Phi_{(r,0)(L,r):\pi_r(p)\pi_r(q)} \cdot \frac{d_{\pi_r(p)}^{(r)}}{d_{\pi_r(q)}^{(r)}} = \Phi_{(r,0)(L,r):pq}.$$

So

$$\Phi_{(r,0)(L,r):pq} = 0,$$

for $p \neq q$, and

$$\Phi_{(r,0)(L,r):\pi_r(p)\pi_r(p)} = \Phi_{(r,0)(L,r):pp}.$$

In conclusion, we have $\Phi_{(L,0):pq}$ is arbitrary, and for $L > s > 0$,

$$\Phi_{(s,0)(L,s):\pi(p)\pi(p)} = \Phi_{(s,0)(L,s):pp}.$$

4. For all $L > s > t > 0$, and for all $p$, we have

$$\Phi_{(s,t)(t):p} = 0.$$

5. For all $L \geqslant s > 0$, $L > t \geqslant 0$ with $s \neq t$, we have

$$\Phi_{(s)(L,t):\pi_s(p)\pi_t(q)} \cdot \frac{d_{\pi_s(p)}^{(s)}}{d_{\pi_t(q)}^{(t)}} = \Phi_{(s)(L,t):pq}$$

If $(s,t) = (L,0)$, we have

$$\Phi_{(L)(L,0):pq} = \Phi_{(L)(L,0):pq}.$$

6. For all $L > t > 0$, we have

$$\sum_{p=1}^{n_L} \Phi_{(L,t)(t):p} \cdot [Wb]_p^{(L,t)(t)} + \sum_{p=1}^{n_t} \sum_{q=1}^{n_t} \Phi_{(t)(L,t):\pi_t(p)\pi_t(q)} \cdot \frac{d_{\pi_t(p)}^{(t)}}{d_{\pi_t(q)}^{(t)}} \cdot [bW]_{pq}^{(t)(L,t)}$$

$$= \sum_{p=1}^{n_L} \Phi_{(L,t)(t):p} \cdot [Wb]_p^{(L,t)(t)} + \sum_{p=1}^{n_t} \sum_{q=1}^{n_t} \Phi_{(t)(L,t):pq} \cdot [bW]_{pq}^{(t)(L,t)}$$

which means $\Phi_{(L,t)(t):p}$ can be arbitrary. The rest is

$$\sum_{p=1}^{n_t} \sum_{q=1}^{n_t} \Phi_{(t)(L,t):\pi_t(p)\pi_t(q)} \cdot \frac{d_{\pi_t(p)}^{(t)}}{d_{\pi_t(q)}^{(t)}} \cdot [bW]_{pq}^{(t)(L,t)} = \sum_{p=1}^{n_t} \sum_{q=1}^{n_t} \Phi_{(t)(L,t):pq} \cdot [bW]_{pq}^{(t)(L,t)}$$

By Lemma B.5 and Lemma A.7, we have

$$\Phi_{(t)(L,t):\pi_t(p)\pi_t(q)} \cdot \frac{d_{\pi_t(p)}^{(t)}}{d_{\pi_t(q)}^{(t)}} = \Phi_{(t)(L,t):pq}.$$

So

$$\Phi_{(t)(L,t):pq} = 0$$

for $p \neq q$, and

$$\Phi_{(t)(L,t):\pi_t(p)\pi_t(p)} = \Phi_{(t)(L,t):pp}.$$

In conclusion, for all $L > t > 0$, we have $\Phi_{(L,t)(t):p}$ is arbitrary, and

$$\Phi_{(t)(L,t):\pi(p)\pi(p)} = \Phi_{(t)(L,t):pp}$$

7. For all $L \geqslant s > 0$ and for all $p$, we have

$$\Phi_{(s):\pi_s(p)} \cdot d^{(s)}_{\pi_s(p)} = \Phi_{(s):p}.$$

If $s = L$, we have

$$\Phi_{(L):p} = \Phi_{(L):p}.$$

8. We have

$$\Phi_1 = \Phi_1.$$

## D.4. $G$-Invariant polynomial layers

Based on the above discussions, we conclude that every $G$-invariant polynomial layer, which is defined as a linear combination of stable polynomial terms, is given as:

$$
\begin{aligned}
I(U) = &\sum_{p=1}^{n_L}\sum_{q=1}^{n_0} \Phi_{(L,0)(L,0):pq} \cdot [WW]^{(L,0)(L,0)}_{pq} + \sum_{p=1}^{n_L}\sum_{q=1}^{n_0} \Phi_{(L,0):pq} \cdot [W]^{(L,0)}_{pq} \\
&+ \sum_{L>s>0}\sum_{p=1}^{n_s} \Phi_{(s,0)(L,s):\bullet\bullet} \cdot [WW]^{(s,0)(L,s)}_{pp} + \sum_{p=1}^{n_L}\sum_{q=1}^{n_0} \Phi_{(L)(L,0):pq} \cdot [bW]^{(L)(L,0)}_{pq} \\
&+ \sum_{L>t>0}\sum_{p=1}^{n_L} \Phi_{(L,t)(t):p} \cdot [Wb]^{(L,t)(t)}_{p} + \sum_{L>t>0}\sum_{p=1}^{n_t} \Phi_{(t)(L,t):\bullet\bullet} \cdot [bW]^{(t)(L,t)}_{pp} \\
&+ \sum_{p=1}^{n_L} \Phi_{(L):p} \cdot [b]^{(L)}_{p} + \Phi_1
\end{aligned}
$$

In the above formula, the bullet $\bullet$ indicates that the corresponding coefficient is independent of the index at the bullet.

# E. Additional Experimental Details

## E.1. Predicting generalization from weights

**Dataset.** The $\mathrm{Tanh}$ subset from the CNN Zoo dataset has 5,949 training instances and 1,488 testing instances, while the original $\mathrm{ReLU}$ subset consists of 6,050 training instances and 1,513 testing instances. We do the augmentation for ReLU subset, with an augmentation factor of 2, effectively doubling the size of the dataset by adding one augmented version of each original instance. The overall dataset sizes, including both the original and augmented data, are summarized in Table 6.

**Baselines** In this experiment, we compare our model with five other baselines:

- **STATNN** (Unterthiner et al., 2020): utilizes statistical features of the weights and biases

- **Graph-NN** (Kofinas et al., 2024): represents input network parameters as graphs and processes using Graph Neural Networks.

- **NP and HNP** (Zhou et al., 2024b): incoporates the permutation symmetries of neurons into neural functional networks.

- **Monomial-NFN** (Tran et al., 2024a): extends the group action on weights from group of permutation matrices to the group of monomial matrices by incoporates scaling/sign-flipping symmetries.

Table 6: Datasets information for predicting generalization task.

| Dataset | Train size | Val size |
|---|---|---|
| Original ReLU | 6050 | 1513 |
| Augment ReLU | 12100 | 3026 |
| Tanh | 5949 | 1488 |

Table 7: Number of parameters of all models for prediciting generalization task.

| Model | ReLU dataset | Tanh dataset |
|---|---|---|
| STATNN | 1.06M | 1.06M |
| NP | 2.03M | 2.03M |
| HNP | 2.81M | 2.81M |
| Monomial-NFN | 0.25M | 1.41M |
| MAGEP-NFN (ours) | 0.99M | 0.99M |

Table 8: Hyperparameters for MAGEP-NFN on prediciting generalization task.

| MLP hidden | Loss | Optimizer | Learning rate | Batch size | Epoch |
|---|---|---|---|---|---|
| 500 | Binary cross-entropy | Adam | 0.001 | 8 | 50 |

Table 9: Dataset size for Classifying INRs task.

| | Train | Validation | Test |
|---|---|---|---|
| CIFAR-10 | 45000 | 5000 | 10000 |
| MNIST size | 45000 | 5000 | 10000 |
| Fashion-MNIST | 45000 | 5000 | 20000 |

**Implementation Details.** Our architecture begins with a Monomial-NFN layer featuring 20 channels, aligning the dimensions of the weights and biases. This is followed by two equivariant MAGEP-NFN layers, each with 20 channels, using either a ReLU activation for the ReLU dataset or a Tanh activation for the Tanh dataset. Next, the output is processed by a MAGEP-NFN Invariant layer. The final output from this layer is flattened and mapped to $\mathbb{R}^{500}$. This vector is further processed by a fully connected MLP with two hidden layers, both activated by ReLU. The final output undergoes a linear projection to a scalar, followed by a sigmoid function. The model is trained using Binary Cross Entropy (BCE) loss over 50 epochs, with early stopping determined by a threshold $\tau$ on the validation set. The entire training process on an A100 GPU takes 30 minutes. A summary of hyperparameters can be found in Table 8.

For the baseline models, we adhere to the original implementations as outlined in (Zhou et al., 2024b), utilizing the official code (available at: https://github.com/AllanYangZhou/nfn), and (Tran et al., 2024a). In the HNP, NP, and Monomial-NFN models, we employ three equivariant layers with channel configurations of 16, 16, and 5, respectively. The extracted features are then passed through an average pooling layer, followed by three MLP layers with hidden dimensions of 200 (Monomial-NFN ReLU case) and 1000 neurons (Monomial-NFN Tanh case and other models). The hyperparameters for our model, along with the parameter counts for all models involved in this task, are detailed in Table 7.

### E.2. Classifying implicit neural representations of images

**Dataset.** We use the original INRs dataset, which contained three dataset: CIFAR-10, MNIST and Fashion-MNIST, obtained by applying a single SIREN model to every image. The detailed information about dataset is described in (Zhou et al., 2024b). We use the datasett without any data augmentation as in the settings of (Tran et al., 2024a). The breakdown of training, validation, and test sample sizes for each dataset is detailed in Table 9.

Table 10: Hyperparameters of MAGEP-NFN for each dataset in Classify INRs task.

| | MNIST | Fashion-MNIST | CIFAR-10 |
|---|---|---|---|
| MAGEP-NFN hidden dimension | 128x3 | 64 | 64 x 3 |
| Base model | MAGEP-Inv | NP | MAGEP-Inv |
| Base model hidden dimension | 128 | 128x3 | 64 |
| MLP hidden neurons | 1000 | 500 | 1000 |
| Dropout | 0.1 | 0.1 | 0.1 |
| Learning rate | 0.0001 | 0.0001 | 0.0001 |
| Batch size | 32 | 32 | 32 |
| Step | 200000 | 200000 | 200000 |
| Loss | Binary cross-entropy | Binary cross-entropy | Binary cross-entropy |

Table 11: Number of parameters of all models for classifying INRs task.

| | CIFAR-10 | MNIST | Fashion-MNIST |
|---|---|---|---|
| MLP | 2M | 2M | 2M |
| NP | 16M | 15M | 15M |
| HNP | 42M | 22M | 22M |
| Monomial-NFN | 16M | 22M | 20M |
| MAGEP-NFN (ours) | 3.4M | 4.1M | 4.9M |

Table 12: Number of parameters of all models for Weight space style editing task.

| Model | Number of parameters |
|---|---|
| MLP | 4.5M |
| NP | 4.1M |
| HNP | 12.8M |
| Monomial-NFN | 4.1M |
| MAGEP-NFN (ours) | 4.1M |

Table 13: Hyperparameters for MAGEP-NFN on weight space style editing task.

| Name | Value |
|---|---|
| MAGEP-NFN hidden dimension | 16 |
| NP dimension | 128 |
| Optimizer | Adam |
| Learning rate | 0.001 |
| Batch size | 32 |
| Steps | 50000 |

**Implementation Details.** In these experiments, we use two different architectures. For both the MNIST and CIFAR datasets, the architecture begins with a Monomial-NFN layer to adjust the weight dimensions, followed by three MAGEP-NFN layers, each utilizing sine activation. The resulting weight features are then passed through a MAGEP Invariant layer. Finally, the output is flattened and processed by an MLP with two hidden layers, each containing 1,000 units and using $\mathrm{ReLU}$ activations.

For the Fashion-MNIST dataset, we begin with a Monomial-NFN layer with sine activation, followed by a MAGEP-NFN

layer also utilizing sine activation, and then a Monomial-NFN layer with absolute activation. The architecture then aligns with the design of the NP model from (Zhou et al., 2024b). Specifically, a Gaussian Fourier Transformation is applied to encode the input into sine and cosine components, mapping from 1 dimension to 256 dimensions. The encoded features are processed through IOSinusoidalEncoding, a positional encoding tailored for the NP layer, which uses a maximum frequency of 10 and 6 frequency bands. Following this, the features pass through three NP layers with $\mathrm{ReLU}$ activations. An average pooling is applied, after which the output is flattened, and the resulting vector is processed by an MLP with two hidden layers, each containing 1,000 units and using $\mathrm{ReLU}$ activations. Finally, the output is linearly projected to a scalar. We employ the Binary Cross Entropy (BCE) loss function and train the model for 200,000 steps, taking approximately 2 hours on an A100 GPU. The parameter counts for all models are presented in Table 11

### E.3. Weight space style editing

**Dataset.** We utilize the same INRs dataset as employed in the classification task, with the sizes of the training, validation, and test sets provided in Table 9.

**Implementation Details.** In these experiments, our architecture begins with two MAGEP-NFN layers, each with 16 hidden dimensions. The rest of the design follows the NP model outlined in (Zhou et al., 2024b). Specifically, we apply a Gaussian Fourier Transformation with a mapping size of 256, followed by IOSinusoidalEncoding. The features are then processed through three NP layers, each with 128 hidden dimensions and $\mathrm{ReLU}$ activation. The final output is passed through an NP layer for scalar projection and a LearnedScale layer as described in the Appendix of (Zhou et al., 2024b). We use the Binary Cross Entropy (BCE) loss function and train the model for 50,000 steps, which takes approximately 35 minutes on an A100 GPU.

For the baseline models, we maintain the same settings as the official implementation. Specifically, the HNP or NP model consists of three layers, each with 128 hidden dimensions, followed by $\mathrm{ReLU}$ activations. An NFN of the same type is applied to map the output to one dimension, after which it is processed by a LearnedScale layer. The number of parameters for all models is detailed in Table 12, and the hyperparameters for our model are presented in Table 13.

## F. Performance comparison with graph-based NFNs

**Experiment Setup**: Following the same experiment setup in Appendix E.1, we compare the predictive performance of our model and two graph-based baselines: GNN (Kofinas et al., 2024) and ScaleGMN (Kalogeropoulos et al., 2024), using HNP (Zhou et al., 2024b) as a reference.

**Results**: The results are presented in Table 14. The GNN model exhibits a noticeable performance decline when tested on separate activation subsets. Although ScaleGMN significantly improves performance on the Tanh subset, its enhancements on the ReLU subset are comparatively modest. In contrast, our model demonstrates substantial overall improvements across both datasets, highlighting its effectiveness.

Table 14: Performance comparison with Graph-based NFNs on Small CNN Zoo task.

|  | ReLU subset | Tanh subset |
| --- | --- | --- |
| HNP (Zhou et al., 2024b) | 0.897 | 0.934 |
| GNN (Kofinas et al., 2024) | 0.897 | 0.893 |
| ScaleGMN (Kalogeropoulos et al., 2024) | 0.928 | **0.942** |
| MAGEP-NFNs (ours) | **0.933** | 0.940 |

## G. Memory and Runtime.

Table 15 and 16 provide runtime and memory consumption data for our model and other baselines in predicting CNN generalization task. For graph-based architectures, we compare with two recent works: GNN (Kofinas et al., 2024) and ScaleGMN (Kalogeropoulos et al., 2024). Our model runs significantly faster and uses much less memory than these graph-based networks and NP/HNP (Zhou et al., 2024b). Introducing additional polynomial terms slightly increases our model's runtime and memory usage compared to Monomial-NFN (Tran et al., 2024a). However, this trade-off results in considerably enhanced expressivity, which is evident across many tasks like Predict CNN Generalization or INRs

Table 15: Runtime of models on Small CNN Zoo task.

|  | ReLU subset | Tanh subset |
| --- | --- | --- |
| NP (Zhou et al., 2024b) | 36m40s | 35m34s |
| HNP (Zhou et al., 2024b) | 30m06s | 29m37s |
| GNN (Kofinas et al., 2024) | 4h27m29s | 4h25m17s |
| ScaledGMN (Kalogeropoulos et al., 2024) | 1h20m | 1h20m |
| Monomial-NFN (Tran et al., 2024a) | **23m47s** | **18m23s** |
| MAGEP-NFNs (ours) | 28m43s | 28m12s |

Table 16: Memory consumption of models on Small CNN Zoo task.

|  | ReLU subset | Tanh subset |
| --- | --- | --- |
| NP (Zhou et al., 2024b) | 838MB | 838MB |
| HNP (Zhou et al., 2024b) | 856MB | 856MB |
| GNN (Kofinas et al., 2024) | 6390MB | 6390MB |
| ScaledGMN (Kalogeropoulos et al., 2024) | 2918MB | 2918MB |
| Monomial-NFN (Tran et al., 2024a) | **560MB** | **582MB** |
| MAGEP-NFNs (ours) | 584MB | 584MB |

Classification.

# H. Implementation of Equivariant and Invariant Layers

We provide the multi-channel implementations of the $\mathcal{G}_{\mathcal{U}}$-equivariant map $E\colon \mathcal{U}^d \to \mathcal{U}^e$ and the $\mathcal{G}_{\mathcal{U}}$-invariant map $I\colon \mathcal{U}^d \to \mathbb{R}^{e \times d'}$. For uniformity in implementing Equivariant and Invariant layers from Appendix C.5 and Appendix D.4, we employ einops-style pseudocode as a consistent framework.

We summarize the key dimensions in Table 17 and outline the shapes of the input terms in Table 18.

## H.1. Equivariant Layers Pseudocode

### H.1.1. PSEUDOCODE FOR CASE $i = L$

From the formula for $[E(W)]_{jk}^{(L)}$:

$$[E(W)]_{jk}^{(L)} = \sum_{p=1}^{n_L} \Phi_{(L,L-1):p}^{(L):j} \cdot [W]_{pk}^{(L,L-1)} + \sum_{p=1}^{n_L} \Phi_{(L,0)(L,L-1):p}^{(L):j} \cdot [WW]_{pk}^{(L,0)(L,L-1)}$$

Table 17: Summary of key dimensions involved in the implementation

| Symbol | Description |
| --- | --- |
| $d$ | Number of input channels for the equivariant and invariant layer |
| $e$ | Number of output channels for the equivariant and invariant layer |
| $b$ | Batch size |
| $n_i$ | Number of channels at the $i_t h$ layer |
| $d'$ | Embedding dimension of the invariant layer's output |

Table 18: Shapes of input terms used in the implementation

| Term | Shape |
|------|-------|
| $[W]^{(s,t)}$ | $[b, d, n_s, n_t]$ |
| $[Wb]^{(s,t)(t)}$ | $[b, d, n_s]$ |
| $[bW]^{(s)(L,t)}$ | $[b, d, n_s, n_t]$ |
| $[WW]^{(s,0)(L,t)}$ | $[b, d, n_s, n_t]$ |

$$+ \sum_{p=1}^{n_L} \Phi_{(L)(L,L-1):p}^{(L):j} \cdot [bW]_{pk}^{(L)(L,L-1)}$$

We define the pseudocode for each term:

For $\Phi_{(L,L-1):p}^{(L):j} \cdot [W]_{pk}^{(L,L-1)}$,

    with $[W]_{pk}^{(L,L-1)}$ of shape $[b, d, n_L, n_{L-1}]$ and $\Phi_{(L,L-1):p}^{(L):j}$ of shape $[e, d, n_L, n_L]$,

    Corresponding pseudocode: $\texttt{einsum}(edpj, bdpk \rightarrow bejk)$

For $\Phi_{(L,0)(L,L-1):p}^{(L):j} \cdot [WW]_{pk}^{(L,0)(L,L-1)}$,

    with $[WW]_{pk}^{(L,0)(L,L-1)}$ of shape $[b, d, n_L, n_{L-1}]$ and $\Phi_{(L,0)(L,L-1):p}^{(L):j}$ of shape $[e, d, n_L, n_L]$,

    Corresponding pseudocode: $\texttt{einsum}(edpj, bdpk \rightarrow bejk)$

For $\Phi_{(L)(L,L-1):p}^{(L):j} \cdot [bW]_{pk}^{(L)(L,L-1)}$,

    with $[bW]_{pk}^{(L)(L,L-1)}$ of shape $[b, d, n_L, n_{L-1}]$ and $\Phi_{(L)(L,L-1):p}^{(L):j}$ of shape $[e, d, n_L, n_L]$,

    Corresponding pseudocode: $\texttt{einsum}(edpj, bdpk \rightarrow bejk)$

From the formula for $[E(b)]_j^{(L)}$:

$$[E(b)]_j^{(L)} = \sum_{p=1}^{n_L}\sum_{q=1}^{n_0} \Phi_{(L,0)(L,0):pq}^{(L):j} \cdot [WW]_{pq}^{(L,0)(L,0)} + \sum_{p=1}^{n_L}\sum_{q=1}^{n_0} \Phi_{(L,0):pq}^{(L):j} \cdot [W]_{pq}^{(L,0)}$$

$$+ \sum_{L>s>0}\sum_{p=1}^{n_s} \Phi_{(s,0)(L,s):}^{(L):j} \cdot [WW]_{pp}^{(s,0)(L,s)} + \sum_{p=1}^{n_L}\sum_{q=1}^{n_0} \Phi_{(L)(L,0):pq}^{(L):j} \cdot [bW]_{pq}^{(L)(L,0)}$$

$$+ \sum_{L>t>0}\sum_{p=1}^{n_L} \Phi_{(L,t)(t):p}^{(L):j} \cdot [Wb]_p^{(L,t)(t)} + \sum_{L>t>0}\sum_{p=1}^{n_t} \Phi_{(L)(L,t):}^{(L):j} \cdot [bW]_{pp}^{(t)(L,t)}$$

$$+ \sum_{p=1}^{n_L} \Phi_{(L):p}^{(L):j} \cdot [b]_p^{(L)} + \Phi_1^{(L):j}$$

We define the pseudocode for each term:

For $\Phi_{(L,0)(L,0):pq}^{(L):j} \cdot [WW]_{pq}^{(L,0)(L,0)}$,

    with $[WW]_{pq}^{(L,0)(L,0)}$ of shape $[b, d, n_L, n_0]$ and $\Phi_{(L,0)(L,0):pq}^{(L):j}$ of shape $[e, d, n_L, n_0, n_L]$,

    Corresponding pseudocode: $\texttt{einsum}(edpqj, bdpq \rightarrow bej)$

For $\Phi_{(L,0):pq}^{(L):j} \cdot [W]_{pq}^{(L,0)}$, with $[W]_{pq}^{(L,0)}$ of shape $[b, d, n_L, n_0]$ and $\Phi_{(L,0):pq}^{(L):j}$ of shape $[e, d, n_L, n_0, n_L]$,

    Corresponding pseudocode: $\texttt{einsum}(edpqj, bdpq \rightarrow bej)$

For $\Phi^{(L):j}_{(s,0)(L,s):} \cdot [WW]^{(s,0)(L,s)}_{pp}$,

    with $[WW]^{(s,0)(L,s)}_{pp}$ of shape $[b,d,n_s,n_s]$ and $\Phi^{(L):j}_{(s,0)(L,s):}$ of shape $[e,d,n_s,n_s,n_L]$,

    Corresponding pseudocode: $\texttt{einsum}(edppj, bdpp \to bej)$

For $\Phi^{(L):j}_{(L)(L,0):pq} \cdot [bW]^{(L)(L,0)}_{pq}$,

    with $[bW]^{(L)(L,0)}_{pq}$ of shape $[b,d,n_L,n_0]$ and $\Phi^{(L):j}_{(L)(L,0):pq}$ of shape $[e,d,n_L,n_0,n_L]$,

    Corresponding pseudocode: $\texttt{einsum}(edpqj, bdpq \to bej)$

For $\Phi^{(L):j}_{(L,t)(t):p} \cdot [Wb]^{(L,t)(t)}_{p}$, with $[Wb]^{(L,t)(t)}_{p}$ of shape $[b,d,n_L]$ and $\Phi^{(L):j}_{(L,t)(t):p}$ of shape $[e,d,n_L,n_L]$,

    Corresponding pseudocode: $\texttt{einsum}(edpj, bdp \to bej)$

For $\Phi^{(L):j}_{(t)(L,t):} \cdot [bW]^{(t)(L,t)}_{pp}$, with $[bW]^{(t)(L,t)}_{pp}$ of shape $[b,d,n_t,n_t]$ and $\Phi^{(L):j}_{(t)(L,t):}$ of shape $[e,d,n_t,n_t,n_L]$,

    Corresponding pseudocode: $\texttt{einsum}(edppj, bdpp \to bej)$

For $\Phi^{(L):j}_{(L):p} \cdot [b]^{(L)}_{p}$, with $[b]^{(L)}_{p}$ of shape $[b,d,n_L]$ and $\Phi^{(L):j}_{(L):p}$ of shape $[e,d,n_L,n_L]$,

    Corresponding pseudocode: $\texttt{einsum}(edpj, bdp \to bej)$

For $\Phi^{(L):j}_{1}$ of shape $[e,n_L]$,

    Corresponding pseudocode: $\texttt{einsum}(ej, \to ej).unsqueeze(0)$

### H.1.2. PSEUDOCODE FOR CASE $i = 1$

From the formula for $[E(W)]^{(1)}_{jk}$:

$$[E(W)]^{(1)}_{jk} = \sum_{q=1}^{n_0} \Phi^{(1):\bullet k}_{(1,0):\bullet q} \cdot [W]^{(1,0)}_{jq} + \sum_{q=1}^{n_0} \Phi^{(1):\bullet k}_{(1,0)(L,0):\bullet q} \cdot [WW]^{(1,0)(L,0)}_{jq}$$

$$+ \sum_{q=1}^{n_0} \Phi^{(1):\bullet k}_{(1)(L,0):\bullet q} \cdot [bW]^{(1)(L,0)}_{jq} + \Phi^{(1):\bullet k}_{(1):\bullet} \cdot [b]^{(1)}_{j}$$

We define the pseudocode for each term:

    For $\Phi^{(1):\bullet k}_{(1,0):\bullet q} \cdot [W]^{(1,0)}_{jq}$, with $[W]^{(1,0)}_{jq}$ of shape $[b,d,n_1,n_0]$ and $\Phi^{(1):\bullet k}_{(1,0):\bullet q}$ of shape $[d,e,n_0,n_0]$,

        Corresponding pseudocode: $\texttt{einsum}(bdjq, deqk \to bejk)$

    For $\Phi^{(1):\bullet k}_{(1,0)(L,0):\bullet q} \cdot [WW]^{(1,0)(L,0)}_{jq}$,

        with $[WW]^{(1,0)(L,0)}_{jq}$ of shape $[b,d,n_1,n_0]$ and $\Phi^{(1):\bullet k}_{(1,0)(L,0):\bullet q}$ of shape $[d,e,n_0,n_0]$,

        Corresponding pseudocode: $\texttt{einsum}(bdjq, deqk \to bejk)$

    For $\Phi^{(1):\bullet k}_{(1)(L,0):\bullet q} \cdot [bW]^{(1)(L,0)}_{jq}$,

        with $[bW]^{(1)(L,0)}_{jq}$ of shape $[b,d,n_1,n_0]$ and $\Phi^{(1):\bullet k}_{(1)(L,0):\bullet q}$ of shape $[d,e,n_0,n_0]$,

        Corresponding pseudocode: $\texttt{einsum}(bdjq, deqk \to bejk)$

    For $\Phi^{(1):\bullet k}_{(1):\bullet} \cdot [b]^{(1)}_{j}$, with $[b]^{(1)}_{j}$ of shape $[b,d,n_1]$ and $\Phi^{(1):\bullet k}_{(1):\bullet}$ of shape $[d,e,n_0]$,

        Corresponding pseudocode: $\texttt{einsum}(bdj, dek \to bejk)$

From the formula for $[E(b)]^{(1)}_{j}$:

$$[E(b)]^{(1)}_{j} = \sum_{q=1}^{n_0} \Phi^{(1):\bullet}_{(1,0):\bullet q} \cdot [W]^{(1,0)}_{jq} + \sum_{q=1}^{n_0} \Phi^{(1):\bullet}_{(1,0)(L,0):\bullet q} \cdot [WW]^{(1,0)(L,0)}_{jq}$$

$$+ \sum_{q=1}^{n_0} \Phi_{(1)(L,0):\bullet q}^{(1):\bullet} \cdot [bW]_{jq}^{(1)(L,0)} + \Phi_{(1):\bullet}^{(1):\bullet} \cdot [b]_j^{(1)}$$

We define the pseudocode for each term:

For $\Phi_{(1,0):\bullet q}^{(1):\bullet} \cdot [W]_{jq}^{(1,0)}$, with $[W]_{jq}^{(1,0)}$ of shape $[b, d, n_1, n_0]$ and $\Phi_{(1,0):\bullet q}^{(1):\bullet}$ of shape $[d, e, n_0]$,

Corresponding pseudocode: `einsum`($bdjq, deq \rightarrow bej$)

For $\Phi_{(1,0)(L,0):\bullet q}^{(1):\bullet} \cdot [WW]_{jq}^{(1,0)(L,0)}$,

with $[WW]_{jq}^{(1,0)(L,0)}$ of shape $[b, d, n_1, n_0]$ and $\Phi_{(1,0)(L,0):\bullet q}^{(1):\bullet}$ of shape $[d, e, n_0]$,

Corresponding pseudocode: `einsum`($bdjq, deq \rightarrow bej$)

For $\Phi_{(1)(L,0):\bullet q}^{(1):\bullet} \cdot [bW]_{jq}^{(1)(L,0)}$, with $[bW]_{jq}^{(1)(L,0)}$ of shape $[b, d, n_1, n_0]$ and $\Phi_{(1)(L,0):\bullet q}^{(1):\bullet}$ of shape $[d, e, n_0]$,

Corresponding pseudocode: `einsum`($bdjq, deq \rightarrow bej$)

For $\Phi_{(1):\bullet}^{(1):\bullet} \cdot [b]_j^{(1)}$, with $[b]_j^{(1)}$ of shape $[b, d, n_1]$ and $\Phi_{(1):\bullet}^{(1):\bullet}$ of shape $[d, e]$,

Corresponding pseudocode: `einsum`($bdj, de \rightarrow bej$)

### H.1.3. PSEUDOCODE FOR CASE $1 < i < L$

From the formula for $[E(W)]_{jk}^{(i)}$:

$$[E(W)]_{jk}^{(i)} = \left( \Phi_{(i,i-1):\bullet\bullet}^{(i):\bullet\bullet} \right) \cdot [W]_{jk}^{(i,i-1)} + \left( \Phi_{(i,0)(L,i-1):\bullet\bullet}^{(i):\bullet\bullet} \right) \cdot [WW]_{jk}^{(i,0)(L,i-1)}$$
$$+ \left( \Phi_{(i)(L,i-1):\bullet\bullet}^{(i):\bullet\bullet} \right) \cdot [bW]_{jk}^{(i)(L,i-1)}$$

We define the pseudocode for each term:

For $\Phi_{(i,i-1):\bullet\bullet}^{(i):\bullet\bullet} \cdot [W]_{jk}^{(i,i-1)}$, with $[W]_{jk}^{(i,i-1)}$ of shape $[b, d, n_i, n_{i-1}]$ and $\Phi_{(i,i-1):\bullet\bullet}^{(i):\bullet\bullet}$ of shape $[d, e]$,

Corresponding pseudocode: `einsum`($bdjk, de \rightarrow bejk$)

For $\Phi_{(i,0)(L,i-1):\bullet\bullet}^{(i):\bullet\bullet} \cdot [WW]_{jk}^{(i,0)(L,i-1)}$,

with $[WW]_{jk}^{(i,0)(L,i-1)}$ of shape $[b, d, n_i, n_{i-1}]$ and $\Phi_{(i,0)(L,i-1):\bullet\bullet}^{(i):\bullet\bullet}$ of shape $[d, e]$,

Corresponding pseudocode: `einsum`($bdjk, de \rightarrow bejk$)

For $\Phi_{(i)(L,i-1):\bullet\bullet}^{(i):\bullet\bullet} \cdot [bW]_{jk}^{(i)(L,i-1)}$,

with $[bW]_{jk}^{(i)(L,i-1)}$ of shape $[b, d, n_i, n_{i-1}]$ and $\Phi_{(i)(L,i-1):\bullet\bullet}^{(i):\bullet\bullet}$ of shape $[d, e]$,

Corresponding pseudocode: `einsum`($bdjk, de \rightarrow bejk$)

From the formula for $[E(b)]_j^{(i)}$:

$$[E(b)]_j^{(i)} = \sum_{q=1}^{n_0} \left( \Phi_{(i,0):\bullet q}^{(i):\bullet} \right) \cdot [W]_{jq}^{(i,0)} + \sum_{q=1}^{n_0} \left( \Phi_{(i,0)(L,0):\bullet q}^{(i):\bullet} \right) \cdot [WW]_{jq}^{(i,0)(L,0)}$$
$$+ \sum_{i > t > 0} \left( \Phi_{(i,t)(t):\bullet}^{(i):\bullet} \right) \cdot [Wb]_j^{(i,t)(t)} + \sum_{q=1}^{n_t} \left( \Phi_{(i)(L,0):\bullet q}^{(i):\bullet} \right) \cdot [bW]_{jq}^{(i)(L,0)}$$
$$+ \left( \Phi_{(i):\bullet}^{(i):\bullet} \right) \cdot [b]_j^{(i)}$$

We define the pseudocode for each term:

For $\Phi_{(i,0):\bullet q}^{(i):\bullet} \cdot [W]_{jq}^{(i,0)}$, with $[W]_{jq}^{(i,0)}$ of shape $[b, d, n_i, n_0]$ and $\Phi_{(i,0):\bullet q}^{(i):\bullet}$ of shape $[d, e, n_0]$,

Corresponding pseudocode: $\texttt{einsum}(bdjq, deq \rightarrow bej)$

For $\Phi_{(i,0)(L,0):\bullet q}^{(i):\bullet} \cdot [WW]_{jq}^{(i,0)(L,0)}$,

with $[WW]_{jq}^{(i,0)(L,0)}$ of shape $[b, d, n_i, n_0]$ and $\Phi_{(i,0)(L,0):\bullet q}^{(i):\bullet}$ of shape $[d, e, n_0]$,

Corresponding pseudocode: $\texttt{einsum}(bdjq, deq \rightarrow bej)$

For $\Phi_{(i,t)(t):\bullet}^{(i):\bullet} \cdot [Wb]_{j}^{(i,t)(t)}$, with $[Wb]_{j}^{(i,t)(t)}$ of shape $[b, d, n_i]$ and $\Phi_{(i,t)(t):\bullet}^{(i):\bullet}$ of shape $[d, e]$,

Corresponding pseudocode: $\texttt{einsum}(bdj, de \rightarrow bej)$

For $\Phi_{(i)(L,0):\bullet q}^{(i):\bullet} \cdot [bW]_{jq}^{(i)(L,0)}$, with $[bW]_{jq}^{(i)(L,0)}$ of shape $[b, d, n_i, n_0]$ and $\Phi_{(i)(L,0):\bullet q}^{(i):\bullet}$ of shape $[d, e, n_0]$,

Corresponding pseudocode: $\texttt{einsum}(bdjq, deq \rightarrow bej)$

For $\Phi_{(i):\bullet}^{(i):\bullet} \cdot [b]_{j}^{(i)}$, with $[b]_{j}^{(i)}$ of shape $[b, d, n_i]$ and $\Phi_{(i):\bullet}^{(i):\bullet}$ of shape $[d, e]$,

Corresponding pseudocode: $\texttt{einsum}(bdj, de \rightarrow bej)$

## H.2. Invariant Layers Pseudocode

From the formula for the Invariant layer $I(U)$:

$$
\begin{aligned}
I(U) = & \sum_{p=1}^{n_L} \sum_{q=1}^{n_0} \Phi_{(L,0)(L,0):pq} \cdot [WW]_{pq}^{(L,0)(L,0)} + \sum_{p=1}^{n_L} \sum_{q=1}^{n_0} \Phi_{(L,0):pq} \cdot [W]_{pq}^{(L,0)} \\
& + \sum_{L>s>0} \sum_{p=1}^{n_s} \Phi_{(s,0)(L,s):\bullet\bullet} \cdot [WW]_{pp}^{(s,0)(L,s)} + \sum_{p=1}^{n_L} \sum_{q=1}^{n_0} \Phi_{(L)(L,0):pq} \cdot [bW]_{pq}^{(L)(L,0)} \\
& + \sum_{L>t>0} \sum_{p=1}^{n_L} \Phi_{(L,t)(t):p} \cdot [Wb]_{p}^{(L,t)(t)} + \sum_{L>t>0} \sum_{p=1}^{n_t} \Phi_{(t)(L,t):\bullet\bullet} \cdot [bW]_{pp}^{(t)(L,t)} \\
& + \sum_{p=1}^{n_L} \Phi_{(L):p} \cdot [b]_{p}^{(L)} + \Phi_1
\end{aligned}
$$

We define the pseudocode for each term:

For $\Phi_{(L,0)(L,0):pq} \cdot [WW]_{pq}^{(L,0)(L,0)}$,

with $[WW]_{pq}^{(L,0)(L,0)}$ of shape $[b, d, n_L, n_0]$ and $\Phi_{(L,0)(L,0):pq}$ of shape $[d, e, n_L, n_0, d']$,

Corresponding pseudocode: $\texttt{einsum}(bdpq, depqk \rightarrow bek)$

For $\Phi_{(L,0):pq} \cdot [W]_{pq}^{(L,0)}$, with $[W]_{pq}^{(L,0)}$ of shape $[b, d, n_L, n_0]$ and $\Phi_{(L,0):pq}$ of shape $[d, e, n_L, n_0, d']$,

Corresponding pseudocode: $\texttt{einsum}(bdpqk, depqk \rightarrow bek)$

For $\Phi_{(s,0)(L,s):\bullet\bullet} \cdot [WW]_{pp}^{(s,0)(L,s)}$,

with $[WW]_{pp}^{(s,0)(L,s)}$ of shape $[b, d, n_s]$ and $\Phi_{(s,0)(L,s):\bullet\bullet}$ of shape $[d, e, d']$,

Corresponding pseudocode: $\texttt{einsum}(bdpk, dek \rightarrow bek)$

For $\Phi_{(L)(L,0):pq} \cdot [bW]_{pq}^{(L)(L,0)}$,

with $[bW]_{pq}^{(L)(L,0)}$ of shape $[b, d, n_L, n_0]$ and $\Phi_{(L)(L,0):pq}$ of shape $[d, e, n_L, n_0, d']$,

Corresponding pseudocode: $\texttt{einsum}(bdpqk, depqk \rightarrow bek)$

For $\Phi_{(L,t)(t):p} \cdot [Wb]_{p}^{(L,t)(t)}$, with $[Wb]_{p}^{(L,t)(t)}$ of shape $[b, d, n_L]$ and $\Phi_{(L,t)(t):p}$ of shape $[d, e, n_L, d']$,

Corresponding pseudocode: $\texttt{einsum}(bdpk, deijk \rightarrow bek)$

For $\Phi_{(t)(L,t):\bullet\bullet} \cdot [bW]_{pp}^{(t)(L,t)}$, with $[bW]_{pp}^{(t)(L,t)}$ of shape $[b, d, n_t]$ and $\Phi_{(t)(L,t):\bullet\bullet}$ of shape $[d, e, d']$,

Corresponding pseudocode: $\texttt{einsum}(bdpk, dek \rightarrow bek)$

For $\Phi_{(L):p} \cdot [b]_p^{(L)}$, with $[b]_p^{(L)}$ of shape $[b, d, n_L]$ and $\Phi_{(L):p}$ of shape $[d, e, n_L, d']$,

    Corresponding pseudocode: $\texttt{einsum}(bdpk, depk \rightarrow bek)$

For $\Phi_1$ of shape $[e, d']$,

    Corresponding pseudocode: $\texttt{einsum}(ek \rightarrow ek).unsqueeze(0)$

