# OpenReview forum: "Equivariant Polynomial Functional Networks"
_ICML.cc/2025/Conference — ICML 2025 poster_

### Official Review · Reviewer_WSKW · 2025-03-15

**Overall Recommendation:** 4

**Summary:**

A neural functional network takes (the weights and biases of) a neural net as input and predicts some related quantity, such as its expected performance. A central problem is that standard architectures exhibit a variety of symmetries, such as permuting the neurons in fully connected in each layer, or scaling their outputs.

Earlier work proposed functional architectures that are invariant or equivariant to these symmetries, but in the most closely related work [Tran et al. 2024] the constructed invariant/equivariant layers are *linear* in the weights of the input network. The present paper builds on [Tran et al. 2024] to construct higher order equivariant layers. This increases the expressivity of the functional networks.

**Claims And Evidence:**

The theoretical claims are well laid out and proved in theorems. The empirical evaluation also appears sound.

**Essential References Not Discussed:**

n/a

**Experimental Designs Or Analyses:**

Experiments seem okay.

**Methods And Evaluation Criteria:**

Yes.

**Other Comments Or Suggestions:**

- if you refer to the hyperbolic tangent as "tanh" I suggest you refer to "sine" as you "sin"
- Overall I like it that the notions used in the paper are relatively compact. However, the "Wb" "bW" "WW" notation might be taking things a bit too far because they can be confused with the actual product of W and b, etc.. As far as I understand, "Wb", "bW" (indexed by some other parameters) are individual matrices. It gets worse when the group action comes in and the authors use notation such as "gWgW" which really suggests that the two "W"'s here are separate, specific matrices.
- I think that in lines 194 and 222 in the right hand column "with" should be "which"
- the sentence on line 118 in the right hand column seems a bit garbled

**Other Strengths And Weaknesses:**

The main question is how much of a difference it makes to have higher order layers rather than a potentially larger number of linear layers. The authors do present an ablation study, but the difference seems a little incremental. This does not diminish the theoretical significance of the mathematical results presented in the paper, performance differences between competing neural architectures are often very small.

**Questions For Authors:**

- The main question in my mind, as mentioned above, is what the practical benefit of higher order polynomial layers are to a larger number of linear layers.
- How many of these layers is it reasonable to use an actual functional net?
- When you mention that the framework is applicable to CNNs as well, do you mean that neglecting the x-y dimensions in the image plane, the way that that the CNN mixes channels is actually just like a fully connected network and you can apply the same formalism to it for this reason?
- [Tran et al., 2024] also talks about sign-flipping symmetry. Does that have to be sacrificed in the polynomial case?

**Relation To Broader Scientific Literature:**

The authors do a fairly good job of situation their contribution in the wider literature. [Tran et al., 2024] is obviously very closely related and they are upfront about that. Both the setup and the proof strategies appear similar, but the present paper is obviously more general because it considers higher order polynomial invariant/equivariant layers too, not just linear ones. Following [Tran et al., 2024] the present paper's surprise value is limited, but it is still a nice contribution because it carefully derives the form of higher order polynomial layers.

**Theoretical Claims:**

The proofs appear to be correct but I did not check them line by line.

---

> ### Author Rebuttal · Authors · 2025-03-31
>
> **Q1: If you refer to the hyperbolic tangent as "tanh" I suggest you refer to "sine" as you "sin"**
>
> **Q3: I think that in lines 194 and 222 in the right hand column "with" should be "which"**
>
> **Q4: The sentence on line 118 in the right hand column seems a bit garbled**
>
> **Answer Q1+Q3+Q4**. We are grateful to the Reviewer for highlighting these points and will incorporate the appropriate edits into the manuscript.
>
> **Q2: Overall I like it that the notions used in the paper are relatively compact. However, the "Wb" "bW" "WW" ... here are separate, specific matrices.**
>
> **Answer Q2.** We are pleased that the Reviewer appreciated the choice of notation used in the paper. For example, the notation $[WW]^{(s,t)}$ refers to the product
>
> $[W]^{(s)} \cdot \ldots \cdot [W]^{(t+1)}$ (line 178).
>
> After applying a group element $g$, this term becomes
>
> $[gWgW]^{(s,t)} = [gW]^{(s)} \cdot \ldots \cdot [gW]^{(t+1)}$.
>
> We did consider using $[W]^{(s,t)}$ instead of $[WW]^{(s,t)}$, but ultimately chose the latter for consistency and clarity—particularly in cases like $[Wb]$, where a one-character notation would be ambiguous or insufficient.
>
> Terms after group action, such as $[gWgW]$, $[gWgb]$, etc., can indeed be expressed more concisely as $[gWW]$, $[gWb]$, and so on. Given the notational complexity—though necessary—we will continue refining the notation to improve the overall clarity of the paper
>
> **W1: The main question is how much ... but the difference seems a little incremental.**
>
> **Q5: The main question in my mind, as mentioned above, ... are to a larger number of linear layers.**
>
> **Answer W1+Q5.** The motivation for introducing polynomial terms into the layers is to address limitations caused by parameter sharing in the construction of equivariant functionals. Specifically, enforcing equivariance through parameter sharing often forces many parameters to be zero or identical, which can significantly limit the model’s expressiveness.
>
> Theoretically, the representational capacity of the proposed MAGEP-NFN is greater than that of the Monomial-NFN layers introduced in [Tran et al., 2024]. Empirically, MAGEP-NFN also shows slightly improved performance over Monomial-NFN. However, we have not yet explored whether using a small number of higher-order polynomial layers is more effective than using a larger number of linear layers. From the perspective of representational capacity, this remains an interesting theoretical question, and we plan to investigate it in future work.
>
> **Q6: How many of these layers is it reasonable to use an actual functional net?**
>
> **Answer Q6.** We believe the appropriate number of layers depends on the specific task being considered. For example, in the CNN generalization prediction task, we found that two layers of $E(U)$ followed by one layer of $I(U)$ are sufficient. In contrast, for the INR classification task, we use three layers of $E(U)$ followed by one layer of $I(U)$. Across all experiments, we ensure a fair comparison by keeping the number of parameters comparable across all baselines. Further details can be found in Appendix E.
>
> **Q7: When you mention that the framework is applicable to CNNs as well, do you mean that neglecting the x-y dimensions in the image plane, the way that the CNN mixes channels is actually just like a fully connected network and you can apply the same formalism to it for this reason?**
>
> **Answer Q7.** For CNNs, the weight space described in Eq. (1) includes distinct $w_i$'s and $b_i$'s, in contrast to MLPs where all $w_i$'s and $b_i$'s are set to 1. To normalize these differences, we first applied a Monomial-NFN model from [Tran et al., 2024]—which shares the same structure as our proposed layers but without the polynomial terms—to rescale the values. This was followed by applying the MAGEP-NFN layers. This is mentioned in the implementation details (lines 2684–2690).
>
> **Q8: [Tran et al., 2024] also talks about sign-flipping symmetry. Does that have to be sacrificed in the polynomial case?**
>
> **Answer Q8.** Our method is applicable to sign-flipping symmetries. In fact, the layers computed in the paper are themselves sign-flipping equivariant or invariant. However, to ensure that the overall functional—composed of these layers—remains equivariant or invariant under sign-flipping, it is necessary to use odd activation functions. This requirement is consistent with the approach in [Tran et al., 2024].
>
> ---
>
> We thank the Reviewer for the constructive feedback. If the Reviewer finds our clarifications satisfactory, we kindly ask you to consider raising the score. We would be happy to address any further concerns during the next stage of the discussion.

---

### Official Review · Reviewer_wEUN · 2025-03-15

**Overall Recommendation:** 3

**Summary:**

This article proposes the design of specific equivariant polynomial functional network.
The authors introduce their notations and some definitions about polynomial terms, that are transformations of the input weights. Their main result states that for some choice of this polynomial map, it is $G$-invariant where $G$ is a group of permutation and scaling of the weights.

**Claims And Evidence:**

The authors present intermediate claims on stability. Although the statement seem to be technically sound, the authors do not explicit the insights they provide or how actionable they can be for the design of Neural Networks, or Functional Neural Networks thereof.

Also, the notation is particularly heavy, making the evaluation of this article particularly difficult. I would like to enter a detailed discussion phase with the authors to clarify this.

**Essential References Not Discussed:**

NA

**Experimental Designs Or Analyses:**

Yes, the design of the experiments is sound (see additional comment in Methods And Evaluation Criteria).

**Methods And Evaluation Criteria:**

The benchmark datasets are relevant for the evaluation, although the authors do not confirm experimentally (for example with synthetic datasets) that their Functional Neural Networks are, say, equivariant.

**Other Comments Or Suggestions:**

Page 3 first column, there is an entire paragraph that is the same as another one page 1 second column.

**Other Strengths And Weaknesses:**

Other weakness: The presentation of the problem that the article is addressing, and the overall writing of the article could be improved. Some references are very redundant.

**Questions For Authors:**

- In my opinion, for readers who are not very familiar with functional neural networks, the article may gain in clarity if the results were presented only as the design of equivariant neural networks for certain group actions: it does not make a difference, to my understanding, that the inputs are weights of neural networks, except to define the symmetries that are being investigated. The results presented (at least, the theoretical ones, are not specific to $\textit{functional neural networks}$, they are properties of neural networks themselves.

- How does one go from the expressions obtained for $I(U)$ to a neural network?

- Notations for Theorem 3.3 are unclear. What is $g^{(s)}$ ? each component of $g$ s composed $s$ times?

**Relation To Broader Scientific Literature:**

The article relates to many references studying Functional Neural Networks, and the design of Equivariant Functional Neural Networks.
One comment: The reference [Tran et al., 2024] is cited at least 15 times in the article.

**Theoretical Claims:**

I could only review the beginning of these proofs (the first 3 pages), and they are correct to me. The supplementary material contains a very large set of proofs (about 40 pages) that are difficult to entirely review for a venue such as ICML. I would recommend this aspect to be taken in account for a final decision in the paper's acceptance.

Given the above issues above the significance of the results, I would strongly suggest the authors to include proof sketches as a section of the main text, at least for a conference version of the article.

---

> ### Author Rebuttal · Authors · 2025-03-31
>
> **W1: The authors present intermediate claims on stability ... Functional Neural Networks thereof.**
>
> **Answer W1.** We kindly refer the Reviewer to our response to **W1+W3** in Reviewer FUc7’s review.
>
> **W2: Also, the notation is particularly heavy, ... to clarify this.**
>
> **Answer W2.** We kindly refer the Reviewer to our response to **Q1, part d** in Reviewer HSsC’s review.
>
> **W3: The supplementary material contains ... such as ICML.**
>
> **Q1: I would strongly suggest the authors to include proof sketches as a section of the main text.**
>
> **W4: The presentation of the problem ... at least 15 times in the article.**
>
> **Answer W3+W4+Q1**. While the proofs are extensive, we believe they are essential to ensure the rigor and validity of our method. We thank the Reviewer for the helpful suggestion and will include a proof sketch in the revised version.
>
> In broad terms, the derivation of the equivariant and invariant layers follows a two-fold structure:
>
> - Computational Derivation: Starting from the general formulation provided in Equation (10), the specific layer structures in Equation (13) are derived through explicit computation, as detailed in Appendices C and D.
>
> - Theoretical Justification: During this computational process, we address several theoretical challenges in Appendices B.3 and B.4, focusing on key properties like linear independence and basis completeness essential for capturing all equivariant and invariant formulations.
>
> In the revision, we'll include a summary of these steps in the main text to guide readers through the logical structure, and direct them to the appendices for the complete technical details. We'll also refine the writing and references for clarity and readability. We refer to [3] multiple times throughout the paper, as the authors address the same core problem: constructing linear-based functionals with the same equivariance/invariance properties. In contrast, other related works often focus on different types of symmetries or employ alternative methods, such as graph-based approaches. Additionally, their system of notation aligns well with our approach, making it particularly suitable for introducing polynomial terms in our framework.
>
> **Q2: The authors do not confirm experimentally ... equivariant.**
>
> **Answer Q2.** Our equivariant layer is theoretically guaranteed to be equivariant by design, as it is derived through a deterministic and principled framework presented in the paper. Therefore, experimental verification of this property is not strictly necessary.
>
> However, for readers who prefer empirical validation over detailed proofs, we provide a simple method to do so. For the layer $E$, we randomly sample multiple input weights $U$ and group actions $g$, and check whether the equivariance condition $E(g(U))=g(E(U))$ holds. This condition will be generally satisfied, though minor deviations may occur due to rounding errors in computation. A verification code is provided in https://sites.google.com/view/polynomialfunctional-rebuttal.
>
> **Q3: Page 3 first column, ... page 1 second column.**
>
> **Answer Q3.** We appreciate the Reviewer’s observation and will revise the relevant part accordingly to address the issue.
>
> **Q4: In my opinion, for readers ... neural networks themselves.**
>
> **Answer Q4.** In the literature on neural functional networks—also known as hypernetworks or metanetworks—it is important to use precise terminology when describing models that operate on data consisting of neural network weights. Since these models are themselves neural network architectures, referring to them explicitly as neural functional networks (or hypernetworks, metanetworks) helps prevent confusion between the model and its input. We believe it is essential to distinguish these concepts through clear and consistent naming. This practice is also reflected in prior works, including but not limited to: [1], [2], [3], [4], [5], [6], [7].
>
> **Q5: How does one go from ... a neural network?**
>
> **Answer Q5.** The implementation details of the functional modules used for various tasks are provided in Appendix E. In summary, to construct equivariant functionals, we stack multiple equivariant functional layers with ReLU activations in between. For invariant functionals used in regression or classification tasks, we append an invariant functional layer to the end of the equivariant stack, followed by a final MLP.
>
> **Q6: Notations for Theorem 3.3 ... composed $s$ times?**
>
> **Answer Q6.** $g^{(s)}$ is the $s$-th component in the group element $g$. The group and its action on weight spaces are defined in Section 2 (see lines 142-164).
>
> ---
> Due to space constraints, some responses refer to similar points addressed in other reviews, with references at the end of our response to Reviewer FUc7.
>
> We appreciate the Reviewer’s feedback and hope our clarifications are satisfactory. If so, we kindly ask you to consider raising the score. We’re happy to address any further concerns in the next discussion phase.

---

> > ### Comment · Reviewer_wEUN · 2025-04-07
> >
> > I am satisfied with the Author's comment, and ask them kindly to include the updates mentioned in their answer.
> > I have updated my score accordingly.

---

> > > ### Author Response · Authors · 2025-04-08
> > >
> > > Dear Reviewer wEUN,
> > >
> > > We sincerely appreciate the time and effort you invested in reviewing our submission. Your thoughtful and constructive feedback has been incredibly valuable in helping us improve the quality and clarity of our work.
> > >
> > > Best regards,
> > >
> > > Authors

---

### Official Review · Reviewer_FUc7 · 2025-03-19

**Overall Recommendation:** 3

**Summary:**

This paper introduces MAGEP-NFN (Monomial mAtrix Group Equivariant Polynomial Neural Functional Network), a novel neural functional network (NFN) designed to process neural networks as input data. Existing NFNs with permutation and scaling equivariance typically rely on either graph-based message-passing or parameter-sharing mechanisms. While parameter-sharing-based NFNs offer lower memory consumption and faster computation, they often suffer from limited expressivity due to the symmetry constraints of the input networks. MAGEP-NFN addresses this limitation by employing a nonlinear equivariant layer represented as a polynomial in the input weights. This design allows the model to capture more complex relationships between weights from different hidden layers while maintaining efficiency. The authors demonstrate through empirical evaluation that MAGEP-NFN achieves competitive performance and efficiency compared to existing methods.

**Claims And Evidence:**

Claim 1

The authors introduce a novel class of stable polynomial terms in the input weights that remain stable under permutation and scaling (sign-flipping) group actions.
	•	Evidence:
They conduct a comprehensive study on the linear independence of these stable polynomial terms, ensuring they form a sound basis for constructing equivariant and invariant layers. This approach addresses the challenges associated with identifying polynomial orbits under group actions.

Claim 2

The authors characterize all equivariant and invariant layers as linear combinations of these stable polynomial terms, with polynomial degree at most L + 1, where L is the number of layers of the input neural networks.
	•	Evidence:
By focusing on this restricted class of polynomials, the proposed layers are shown to be both computationally efficient and memory-friendly, overcoming the high computational costs of working with generic polynomials.

Claim 3

They design MAGEP-NFN, a family of neural functional networks (NFNs) that are permutation and scaling equivariant, and offer improved expressivity while maintaining low memory consumption and running time.

Evidence:
Built upon the parameter-sharing mechanism, MAGEP-NFN leverages the newly introduced nonlinear equivariant polynomial layers. Empirical evaluations on three tasks—predicting CNN generalization (Small CNN Zoo dataset), weight space style editing, and classifying implicit neural representations (INRs)—demonstrate that MAGEP-NFN achieves competitive performance and efficiency compared to existing baselines.

**Essential References Not Discussed:**

The related work section is thorough and clearly articulated. The authors provide a well-organized overview of prior studies on functional equivalence in neural networks, neural functional networks (NFNs), and equivariant NFNs. They carefully outline the strengths and limitations of existing methods, particularly regarding permutation and scaling symmetries. This contextualization effectively highlights the gap addressed by their proposed MAGEP-NFN framework, demonstrating both a solid understanding of the literature and the relevance of their contribution.

**Experimental Designs Or Analyses:**

The experiments are designed to evaluate the effectiveness, expressivity, and efficiency of the proposed MAGEP-NFNs across both invariant and equivariant tasks. The goal is to demonstrate that MAGEP-NFNs outperform or match baseline models while maintaining low memory consumption and computational efficiency. All experiments are conducted over five independent runs, and results are reported as the mean along with standard error or standard deviation where applicable.

Task 1: Classifying Implicit Neural Representations (INRs)
	•	Objective:
Predict the class label of pretrained Implicit Neural Representation (INR) weights, which encode images from different datasets.
	•	Datasets:
INR weights trained on MNIST, FashionMNIST, and CIFAR-10.
	•	Baselines:
MLP, NP (Zhou et al., 2024b), HNP (Zhou et al., 2024b), and Monomial-NFN (Tran et al., 2024).
	•	Evaluation Metric:
Classification accuracy (%) on test sets.
	•	Protocol:
MAGEP-NFNs are trained and evaluated on the INR datasets, with test accuracy compared to baselines.
	•	Key Variation:
Models are compared in terms of their ability to generalize across different types of image representations encoded in the INRs.

Task 2: Predicting CNN Generalization from Weights
	•	Objective:
Predict the generalization performance of pretrained CNNs based solely on their weights without using test data.
	•	Dataset:
Small CNN Zoo (Unterthiner et al., 2020), divided into subsets based on activation functions:
	•	ReLU networks (with group action M_{>0}^n)
	•	Tanh networks (with group action M_{\pm1}^n)
	•	Baselines:
STATNet, NP, HNP, and Monomial-NFN.
	•	Evaluation Metric:
Kendall’s τ correlation coefficient to measure ranking agreement between predicted and true generalization scores.
	•	Protocol:
	•	For ReLU CNNs, additional experiments are performed with scale augmentations, where diagonal scaling matrices D_{n, ii}^{>0} are randomly sampled from uniform distributions U[1, 10^i] for i = 1, 2, 3, 4.
	•	Permutation matrices P_n are also randomly applied to assess robustness under group actions.
	•	Models are evaluated both on the original and augmented datasets.
	•	Analysis:
Comparisons focus on MAGEP-NFN’s ability to maintain high Kendall’s τ scores under varying levels of input transformations.

Task 3: Weight Space Style Editing
	•	Objective:
Modify pretrained SIREN weights to alter the visual characteristics of the encoded images (contrast enhancement and dilation).
	•	Datasets:
Pretrained SIREN models encoding CIFAR-10 and MNIST images (Zhou et al., 2024b).
	•	Baselines:
NP and Monomial-NFN.
	•	Evaluation Metric:
Mean Squared Error (MSE) between images reconstructed from the modified weights and the target images (contrast-enhanced or dilated versions).
	•	Protocol:
MAGEP-NFNs are trained to perform weight space edits that correspond to desired image modifications. The quality of the edited model output is compared with baselines by calculating MSE.

Task 4: Ablation Study on Higher-Order Inter-Layer Terms
	•	Objective:
Evaluate the contribution of higher-order Inter-Layer terms [W], [WW], [bW], [Wb] to model performance.
	•	Task Context:
CNN generalization prediction (ReLU subset) from Task 2.
	•	Evaluation Metric:
Kendall’s τ correlation coefficient.
	•	Protocol:
Two configurations are compared:
	•	MAGEP-NFN with only Non-Inter-Layer terms
	•	MAGEP-NFN with both Non-Inter-Layer and Inter-Layer terms
	•	Analysis:
Performance improvements (from Kendall’s τ of 0.929 to 0.933) are used to demonstrate the positive impact of including higher-order interactions in the model.

**Methods And Evaluation Criteria:**

Method

The paper introduces MAGEP-NFN (Monomial mAtrix Group Equivariant Polynomial Neural Functional Network), a novel class of Neural Functional Networks (NFNs) designed to process neural networks as input data. The key innovation lies in constructing polynomial invariant and equivariant layers based on stable polynomial terms. These terms are specifically designed to remain stable under the action of the monomial matrix group G, ensuring permutation and scaling (sign-flipping) equivariance.

MAGEP-NFN follows the parameter-sharing mechanism described in (Tran et al., 2024), but extends it by using nonlinear polynomial layers rather than linear ones. This polynomial formulation allows MAGEP-NFN to model more complex inter-layer relationships in the input neural networks, thereby improving expressivity while maintaining low memory usage and computational efficiency.

The construction of the invariant and equivariant polynomial layers is based on linear combinations of stable polynomial terms, ensuring computational tractability. These layers are polynomials of degree at most L + 1, where L is the number of layers in the input neural networks.

The authors derive explicit forms of G-invariant layers by solving parameter-sharing constraints that ensure the network is equivariant or invariant under group actions. MAGEP-NFNs are implemented for tasks involving both equivariance (e.g., weight space style editing) and invariance (e.g., classification and generalization prediction from weights).

Evaluation Criteria

The proposed MAGEP-NFNs are evaluated on three distinct tasks, each designed to test different aspects of the model’s expressivity, efficiency, and equivariance/invariance properties.
	1.	Classification of Implicit Neural Representations (INRs)
	•	Objective: Classify which class each pretrained INR weight was trained on.
	•	Datasets: INR weights trained on MNIST, FashionMNIST, and CIFAR-10 datasets.
	•	Metric: Classification accuracy (%).
	•	Setup: Comparison against baseline models including NP, HNP, and Monomial-NFN over 5 runs. Results include standard error.
	2.	Predicting CNN Generalization from Weights
	•	Objective: Predict the generalization performance of CNNs directly from their weights, without test data evaluation.
	•	Dataset: Small CNN Zoo, divided into ReLU and Tanh activation subsets.
	•	Metrics: Kendall’s τ correlation coefficient, assessing the rank correlation between predicted and true generalization performance.
	•	Setup: Evaluation under varying group actions, including scaling augmentations sampled from different uniform distributions and permutations. Performance is reported as the mean Kendall’s τ across 5 runs with standard deviation.
	3.	Weight Space Style Editing
	•	Objective: Modify the weights of pretrained SIREN models to enhance contrast (CIFAR-10) or dilate images (MNIST).
	•	Datasets: Pretrained SIREN models encoding CIFAR-10 and MNIST images.
	•	Metric: Mean Squared Error (MSE) between the image produced by the modified network and the ground truth modified image.
	•	Setup: Comparisons are made with NP and Monomial-NFN models.
	4.	Ablation Study on Higher-Order Terms
	•	Objective: Evaluate the impact of including higher-order Inter-Layer terms ([W], [WW], [bW], [Wb]) in MAGEP-NFN.
	•	Metric: Kendall’s τ on CNN generalization prediction (ReLU subset).
	•	Setup: Performance comparison between models with and without Inter-Layer terms, showing improvements in expressivity.

**Other Comments Or Suggestions:**

The abstract is informative and covers the key contributions of the paper; however, it feels somewhat lengthy. Condensing the abstract by focusing on the most essential points and streamlining the description of the method and results would improve readability and make the key messages more impactful.

**Other Strengths And Weaknesses:**

Overall, the paper is well-written, and the proposed method is clearly presented and thoroughly evaluated. The integration of permutation and scaling equivariance into the NFN framework is a notable contribution, and the empirical results are convincing. However, one area for improvement is the theoretical grounding of the method. While the paper introduces an innovative equivariant polynomial layer, providing an additional strong theoretical result—such as a formal expressivity theorem or a rigorous analysis of the equivariant properties—would further strengthen the contribution and enhance its impact.

**Questions For Authors:**

In Theorem 3.5, you prove the G-invariance of the proposed equivariant polynomial layer. Could you clarify where the main technical challenges lie in establishing this invariance? Understanding the core difficulties in the proof would help appreciate the contribution more fully.

In Table 3, the performance of MAGEP-NFN appears quite similar to that of the Monomial-NFN across the evaluated tasks. Could you provide further insight into this result? For instance, are there specific scenarios or tasks where the advantages of MAGEP-NFN become more apparent, or are there other factors (e.g., efficiency, scalability) that distinguish your approach in practice?

**Relation To Broader Scientific Literature:**

The paper presents MAGEP-NFN, a neural function network that achieves equivariance to both permutations and scaling symmetries via an innovative parameter-sharing mechanism based on equivariant polynomial layers. This contribution is positioned within the growing body of work on symmetry-aware and equivariant neural networks.

The authors demonstrate an awareness of prior research on permutation-invariant and permutation-equivariant models, such as Deep Sets [Zaheer et al., 2017] and equivariant graph neural networks [Maron et al., 2018; Keriven and Peyré, 2019]. Additionally, their approach complements existing methods that enforce equivariance through parameter-sharing techniques [Ravanbakhsh et al., 2017]. What distinguishes this work is the incorporation of stable polynomial terms in the construction of equivariant layers—an angle not commonly explored in prior neural network architectures.

However, the paper could benefit from a more detailed discussion of how their use of stable polynomials relates to existing literature in both machine learning and the mathematical study of stable polynomials (e.g., Borcea and Brändén, 2009). Clarifying whether this connection offers theoretical guarantees (e.g., in terms of stability, robustness, or generalization) would further strengthen the positioning of the work within the broader scientific context.

Moreover, while the authors mention the potential applicability of their parameter-sharing approach to other architectures (e.g., with normalization layers or alternative activation functions), it would be helpful to reference existing frameworks that tackle such extensions, even if only to highlight distinctions or potential synergies.

In summary, the paper makes a meaningful contribution to the literature on equivariant neural networks, but a deeper engagement with relevant prior works—particularly regarding the role of stable polynomials—would enhance the discussion of its relation to broader scientific advances.

**Theoretical Claims:**

1. Stable Polynomial Terms Enable Efficient Construction of Equivariant and Invariant Layers
	•	Claim: Determining equivariant and invariant layers from generic polynomials over the input weights is computationally infeasible due to the complexity of identifying polynomial orbits under group actions and the high memory and computational cost.
	•	Contribution: The authors introduce a specialized class of polynomials called Stable Polynomial Terms. These terms are carefully designed to remain stable under the action of the monomial matrix group G, which consists of permutations and scaling/sign-flipping transformations.
	•	Impact: By restricting equivariant and invariant polynomial layers to linear combinations of these stable polynomial terms, the model ensures both computational efficiency and reduced memory consumption, making it scalable and practical for NFN tasks.

2. Stable Polynomial Terms Generalize Weight and Bias Entries
	•	Claim: Stable polynomial terms generalize the conventional entries of weight matrices and bias vectors.
	•	Proposition (Proposition 3.2):
For all  L \ge s > t > r \ge 0 , stable polynomial terms satisfy the following recursive relationships:

[W]^{(s,s-1)} = [W]^{(s)} \in \mathbb{R}^{d \times n_s \times n_{s-1}}

and

[W]^{(s,t)} \cdot [W]^{(t,r)} = [W]^{(s,r)} \in \mathbb{R}^{d \times n_s \times n_r}

Similar relationships hold for terms involving biases.
	•	Impact: These recursive definitions show that stable polynomial terms extend the concept of weights and biases, providing a richer structure that facilitates capturing more complex inter-layer dependencies in the input networks.

3. Stability under Group Actions Guarantees Equivariance
	•	Claim: The stable polynomial terms maintain compatibility with the group action of G, ensuring equivariance.
	•	Theorem (Theorem 3.3):
When the group G acts on an input U = ([W],[b]), the stable polynomial terms transform predictably under G:

[gW]^{(s,t)} = g^{(s)} \cdot [W]^{(s,t)}


[gWgb]^{(s,t)}(t) = g^{(s)} \cdot [Wb]^{(s,t)}(t) \cdot (g^{(t)})^{-1}

and similar rules apply to other stable terms.
	•	Impact: These transformation rules guarantee that any layer built from these terms will be equivariant by design, and allow for efficient computation of equivariant and invariant polynomial layers.

5. MAGEP-NFNs Polynomial Layers Include Linear Layers as a Special Case
	•	Claim: The polynomial map I(U), constructed from stable polynomial terms, has a maximum degree of L + 1 in terms of the input weights. This includes linear layers as a special case.
	•	Impact: This ensures that MAGEP-NFNs generalize existing linear methods (such as those in Tran et al., 2024) while offering greater expressivity through higher-order polynomial terms.

---

> ### Author Rebuttal · Authors · 2025-03-31
>
> **W1: The paper could benefit from a more detailed discussion of how their use of stable polynomials ... the broader scientific context.**
>
> **W3: In summary, the paper makes a meaningful contribution to the literature on equivariant neural networks ... would enhance the discussion of its relation to broader scientific advances.**
>
>
> **Answer W1+W3.** We refer to the proposed stable polynomial terms as "stable" because they are inherently equivariant under the considered group action and therefore remain unchanged during the equivariance-enforcing process—specifically, the parameter-sharing mechanism discussed in the paper (lines 160-167). The term "stable" does not pertain to empirical notions such as stability, robustness, or generalization.
>
> **W2: While the authors mention the potential applicability of their parameter-sharing ... highlight distinctions or potential synergies.**
>
> **Answer W2.** Existing frameworks that involve parameter sharing can be mentioned as follows:
>
> - [1]: NFNs for MLPs or CNNs with permutation equivariant.
>
> - [2]: NFNs for Transformer with permutation equivariant.
>
> - [3]: NFNs for MLPs or CNNs with monomial matrix group equivariant.
>
> - [4]: NFNs for Transformer with equivariance under the maximal symmetry group of Multihead Attention mechanism.
>
> **W4: However, one area for improvement is the theoretical grounding ... strengthen the contribution and enhance its impact.**
>
> **Answer W4.** The rigorous computations in Appendices C and D establish the equivariance and invariance of the proposed functional layers. Additionally, Theorem 3.5 provides an expressivity result, stating that any invariant layer computed via the general formulation (10) must take the form given in Eq. (13). An analogous theorem for equivariant layers can be formulated similarly, given that the computational process for these layers follows the same structure.
>
> It is worth noting that the proof of this theorem is non-trivial. It is derived through an analysis of linear independence, as detailed in Appendices B.3 and B.4, which we found to be mathematically challenging.
>
> **W5: The abstract is informative and covers the key contributions ... improve readability and make the key messages more impactful.**
>
> **Answer W5.** We sincerely appreciate the Reviewer’s feedback on the abstract and will make the appropriate edits to improve its clarity and accuracy.
>
> **Q1: In Theorem 3.5, you prove the G-invariance of the proposed equivariant polynomial layer. Could you clarify where the main technical challenges lie in establishing this invariance? Understanding the core difficulties in the proof would help appreciate the contribution more fully.**
>
> **Answer Q1.** While it is relatively straightforward to write down a specific formulation of a layer that is equivariant or invariant under the considered group action, it is significantly more challenging to characterize all possible formulations of such layers. The main technical difficulties arise from two aspects: first, the computations presented in Appendices C and D; and second, the theoretical results in Appendices B.3 and B.4 that underpin these computations, which are essential to ensure that no valid layer type is overlooked.
>
> **Q2: In Table 3, the performance of MAGEP-NFN appears ... (e.g., efficiency, scalability) that distinguish your approach in practice?**
>
> **Answer Q2.** We believe the task in Table 3 is comparatively less challenging than other benchmarks, which explains why Monomial-NFN already performs well and our model offers only a slight improvement. This is reflected in the high Kendall’s $\tau$ values, ranging from $0.913$ to $0.940$ across all models. The strengths of MAGEP-NFN become more evident on more challenging tasks. For instance, in the Classify INR-MNIST task, where the accuracy of other models widely ranges from $10.62\\%$ to $69.82\\%$, MAGEP-NFN achieves the accuracy of $77.55\\%$, surpassing the second-best model by a notable margin of $7.73\\%$.
>
> ---
> We thank the Reviewer for the constructive feedback. If the Reviewer finds our clarifications satisfactory, we kindly ask you to consider raising the score. We would be happy to address any further concerns during the next stage of the discussion.
>
> ---
>
> *References.*
>
> [1] Allan Zhou et al., Permutation Equivariant Neural Functionals, NeurIPS 2023.
>
> [2] Allan Zhou et al., Neural functional transformers, NeurIPS 2023.
>
> [3] Hoang Tran et al., Monomial matrix group equivariant neural functional networks, NeurIPS 2024.
>
> [4] Hoang Tran et al., Equivariant Neural Functional Networks for Transformers, ICLR 2025.
>
> [5] Derek Lim et al., Graph Metanetworks for Processing Diverse Neural Architectures, ICLR 2024.
>
> [6] Ioannis Kalogeropoulos et al., Scale Equivariant Graph Metanetworks, NeurIPS 2024 Oral.
>
> [7] Miltiadis Kofinas et al., Graph Neural Networks for Learning Equivariant Representations of Neural Networks, ICLR 2024 Oral.

---

### Official Review · Reviewer_HSsC · 2025-03-19

**Overall Recommendation:** 3

**Summary:**

The paper is an extension to Tran et al 2024, by presenting a neural functional network based on stable polynomials. The input to the network are weights of other networks, and the constructed network is equivariant to permutations of its neurons and scaling of the weights. The construction is based on defining certain stable polynomials, where the respective groups (permutations and scalings) are equivariant w.r.t these polynomials. Several experiments and an ablation study are given, beating other constructions on similar tasks.

###############################################################

I thank the authors for their response. I still lean toward acceptance, but the paper seems to require some rewriting and clarifications to make the claims clearer and stand-alone, not relying on reading previous works. For example, stating what are the trainable parameters of their model should be clearly stated in the main part, and not left to the appendices. Also, the models for CNN and MLP should be stated, as well as the training procedure, preferably in the main paper. In the current form, I will maintain my score.

**Claims And Evidence:**

I believe the main claim, that the proposed construction improves on previous ones, is well supported on both the theoretical and empirical sides. However, the paper's presentation has some problems, which I will detail later on.

On the theoretical side, the proposed polynomials are proved to be “stable” under the group action, and every other G-invariant map has a similar form.
On the experimental side, several experiments are given for predicting the model’s accuracy, classifying which image an INR trained on and several ablation studies are given.

**Essential References Not Discussed:**

Not that I am aware of

**Experimental Designs Or Analyses:**

See above

**Methods And Evaluation Criteria:**

I believe the evaluation criteria are good and follow the baseline of previous related works. One problem I’m not sure about, is that previous work (e.g. Zhou et al. 2023) also tested on 3d datasets, such as ShapeNet and ScanNet, while this work doesn’t. Is there a reason for that? Because I believe this can give a better case for the improvement of performance over previous works, since these are harder datasets than networks trained on MNIST and CIFAR-10.

**Other Comments Or Suggestions:**

See above

**Other Strengths And Weaknesses:**

There are some major issues with the presentation of the paper which I think can be improved with further work:

1) I think the main issue is that this paper is written as a follow-up to Tran et al. 2024. In this way, some details are missing that are difficult to understand without knowledge of the previous work, and make the reading confusing. Some examples below.

a) The weight space is defined in Section 2, but not how the model is constructed with those weights. I recommend writing the model explicitly.

b) Are the weight space should be some generalized form to include both MLPs and CNNs? In this case for MLPs the w_i’s should be 1 in Eq. (1)?

c) In Eq. (9), the \Psi are the trained parameters? They seem to be hidden inside the bracket notation, so it is not clear whether they are trained or some weights of the model.

d) In Eq. (10) the notations are not clear. For example, whether the \Phi are scalars? what does the (s,t):pq indexing means?

e) What is the dimension d’ in line 193 right side?

2) Another major issue is that the training of the model is not explained in detail. Is it the case that only the coefficients of the polynomials are trained? If so, is it similar to learning a polynomial kernel? If so I think it should be said explicitly. If not, it would be helpful to state the exact training procedure, since the training is different than training standard MLPs or other simple models.

3) Is there some motivation for why choosing the \Psi in this form as learnable parameters? For \Psi it makes sense since these are the coefficients of the polynomials. However, for \Psi it is not clear, since there seem to be many places (i.e. other intermediate layers) where these parameters could be placed. Why they were chosen to be in this form?

**Questions For Authors:**

I would be happy if the authors could respond to the remarks about the presentation above.

**Relation To Broader Scientific Literature:**

I am not very familiar with the neural functionals literature, however, it seems that this paper is a direct followup to this line of work.

**Theoretical Claims:**

From what I went over, the theoretical claims seem sound.

---

> ### Author Rebuttal · Authors · 2025-03-31
>
> **Q1. I think the main issue is that this paper is written as a follow-up ...**
>
> **Answer Q1.** We appreciate the suggestion from the Reviewer and will include explanations of relevant concepts from prior work in the revised Appendix. Below, we address each part of the question.
>
> **a), b)** In the case of MLPs, all $w_i$'s and $b_i$'s values are equal to 1, whereas in CNNs, the $w_i$'s and $b_i$'s values correspond to the sizes of the convolutional kernels.
>
> We present the weight space using arbitrary $w_i$'s and $b_i$'s values for an additional reason: within a functional layer, $w_i$'s and $b_i$'s also represent the number of hidden functional units. This generalization allows the framework to accommodate a broader range of architectures and internal structures.
>
> For MLP's, the way the model is written explicitly is:
>
> $f(\mathbf{x} ~ ; ~ U, \sigma) = W^{(L)} \cdot \sigma \left( \ldots \sigma \left(W^{(2)} \cdot \sigma \left (W^{(1)} \cdot \mathbf{x}+b^{(1)}\right ) +b^{(2)}\right) \ldots \right ) + b^{(L)}.$
>
> **c)** The $\Phi$ and $\Psi$ are trainable parameters. We mentioned in the appendices that, in constructed functional layers, $\Phi_{-}$'s and $\Psi_{-}$'s are trainable parameters (line 1573 for equivariant layers and line 2372 for invariant layers).
>
> **d)** We provide the following examples to clarify the rationale behind our choice of notation in the paper:
>
> $[W], [b]$: Square brackets are used to distinguish whether a term belongs to the weight space or represents learnable weights in functional models.
>
> $\Psi$: In the expression $[bW]^{(s)(L,t)} = [b]^{(s)} \cdot \Psi^{(s)(L,t)} \cdot [W]^{(L,t)}$, the index $(s)(L,t)$ for $\Psi$ is chosen to reflect that it connects two indexed components: $(s)$ and $(L,t)$. This way, simply by inspecting the indices of $\Psi$, one can infer the terms it links—namely, $[b]^{(s)}$ and $[W]^{(L,t)}$.
>
> $\Phi$: In equivariant layers, the notation $\Phi_{(s,t):pq}^{(i):jk}$ represents a scalar used to compute the output component $[E(W)]^{(i):jk}$ (indicated by the top index), corresponding to the input component $[W]\_{(s,t):pq}$ (indicated by the bottom index). In invariant layers, the output does not lie in the weight space, and thus no top index is used—for example, $\Phi_{(s,t):pq}$.
>
> While this indexing system might initially seem verbose, we find it brings clarity and consistency to the paper, particularly when describing the computation of functional layers. We appreciate the Reviewer’s question and will include a paragraph explaining the notation system more thoroughly.
>
> **e)** The dimension $d'$ refers to the output dimension of the invariant layer. It can be chosen arbitrarily, and we use $d'$ to distinguish it from $d$, which denotes dimensions associated with the input.
>
> **Q2. Another major issue is ... or other simple models.**
>
> **Q3. Is there some motivation for why choosing the \Psi ... they were chosen to be in this form?**
>
> **Answer Q2+Q3.** In our functionals, all the trainable parameters are $\Phi$'s and $\Psi$'s.
>
> The main motivation for introducing the stable polynomial term with parameters $\Psi$ in our work—compared to the equivariant functionals in Tran et al. (2024)—is to address limitations arising from parameter sharing during the construction of equivariant functionals. Specifically, enforcing equivariance through parameter sharing often results in many parameters being forced to zero or constrained to be equal, thereby reducing expressiveness. In contrast, the parameters $\Psi$ in our stable polynomial term are inherently equivariant under the considered group action, and thus remain unaffected during the equivariance-enforcing process. As a result, our functionals gain greater representational capacity.
>
> It is important to note that $\Psi$ cannot be placed arbitrarily within the polynomial expression, as doing so could break equivariance. Instead, we carefully position $\Psi$ between terms in a way that ensures the group actions on both sides cancel out, preserving the equivariance of the overall construction.
>
> ---
>
> **Methods And Evaluation Criteria.** Is that previous work (e.g. Zhou et al. 2023) ... on MNIST and CIFAR-10.
>
> **Answer.** We do not include experiments on 3D datasets such as ShapeNet or ScanNet because the dataset of pretrained weights used in Zhou et al. 2023, originally from De Luigi et al. 2023, has not been publicly released. Additionally, the implementation code for Zhou et al. 2023 on these datasets is also not publicly available. Given these unfortunate constraints, it is not feasible to include this baseline within the one-week rebuttal period.
>
> ---
>
> We thank the Reviewer for the constructive feedback. If the Reviewer finds our clarifications satisfactory, we kindly ask you to consider raising the score. We would be happy to address any further concerns during the next stage of the discussion.

---

### Decision · Program_Chairs · 2025-05-01

**Decision:**

Accept (poster)

**Comment:**

This paper contributes to the growing literature on neural functional networks by extending the framework of Tran et al. (2024) to include polynomial equivariant layers, enhancing formal expressivity under symmetry constraints.

Reviewers generally recognize the strength of the theoretical contribution, supported by sound empirical evaluation and a principled generalization of prior work. While some noted that the practical gains over linear layers appear incremental, there was consensus that the paper offers a rigorous and well-motivated theoretical advancement. The authors’ rebuttal effectively clarified key concerns, and with the promised revisions, the paper merits acceptance, primarily on the strength of its theoretical insights.

The authors are encouraged to incorporate the important feedback given by the knowledgeable reviewers, particularly regarding notational clarity and the broader discussion of practical implications relative to deeper linear models.